# Neural Proximal/Trust Region Policy Optimization Attains Globally Optimal Policy

**Boyi Liu**[*†]    **Qi Cai**[*‡]    **Zhuoran Yang**[§]    **Zhaoran Wang**[¶]

## Abstract

Proximal policy optimization and trust region policy optimization (PPO and TRPO) with actor and critic parametrized by neural networks achieve significant empirical success in deep reinforcement learning. However, due to nonconvexity, the global convergence of PPO and TRPO remains less understood, which separates theory from practice. In this paper, we prove that a variant of PPO and TRPO equipped with overparametrized neural networks converges to the globally optimal policy at a sublinear rate. The key to our analysis is the global convergence of infinite-dimensional mirror descent under a notion of one-point monotonicity, where the gradient and iterate are instantiated by neural networks. In particular, the desirable representation power and optimization geometry induced by the overparametrization of such neural networks allow them to accurately approximate the infinite-dimensional gradient and iterate.

## 1 Introduction

Policy optimization aims to find the optimal policy that maximizes the expected total reward through gradient-based updates. Coupled with neural networks, proximal policy optimization (PPO) [40] and trust region policy optimization (TRPO) [39] are among the most important workhorses behind the empirical success of deep reinforcement learning across applications such as games [34] and robotics [13]. However, the global convergence of policy optimization, including PPO and TRPO, remains less understood due to multiple sources of nonconvexity, including (i) the nonconvexity of the expected total reward over the infinite-dimensional policy space and (ii) the parametrization of both policy (actor) and action-value function (critic) using neural networks, which leads to nonconvexity in optimizing their parameters. As a result, PPO and TRPO are only guaranteed to monotonically improve the expected total reward over the infinite-dimensional policy space [23, 24, 39, 40], while the global optimality of the attained policy, the rate of convergence, as well as the impact of parametrizing policy and action-value function all remain unclear. Such a gap between theory and practice hinders us from better diagnosing the possible failure of deep reinforcement learning [37, 19, 21] and applying it to critical domains such as healthcare [28] and autonomous driving [38] in a more principled manner.

Closing such a theory-practice gap boils down to answering three key questions: (i) In the ideal case that allows for infinite-dimensional policy updates based on exact action-value functions, how do PPO and TRPO converge to the optimal policy? (ii) When the action-value function is parametrized by a neural network, how does temporal-difference learning (TD) [41] converge to an approximate action-value function with sufficient accuracy within each iteration of PPO and TRPO? (iii) When

---

[*]equal contribution

[†]Northwestern University; `boyiliu2018@u.northwestern.edu`

[‡]Northwestern University; `qicai2022@u.northwestern.edu`

[§]Princeton University; `zy6@princeton.edu`

[¶]Northwestern University; `zhaoranwang@gmail.com`

the policy is parametrized by another neural network, based on the approximate action-value function attained by TD, how does stochastic gradient descent (SGD) converge to an improved policy that accurately approximates its ideal version within each iteration of PPO and TRPO? However, these questions largely elude the classical optimization framework, as questions (i)-(iii) involve non-convexity, question (i) involves infinite-dimensionality, and question (ii) involves bias in stochastic (semi)gradients [44, 42]. Moreover, the policy evaluation error arising from question (ii) compounds with the policy improvement error arising from question (iii), and they together propagate through the iterations of PPO and TRPO, making the convergence analysis even more challenging.

**Contribution.** By answering questions (i)-(iii), we establish the first nonasymptotic global rate of convergence of a variant of PPO (and TRPO) equipped with neural networks. In detail, we prove that, with policy and action-value function parametrized by randomly initialized and overparametrized two-layer neural networks, PPO converges to the optimal policy at the rate of $O(1/\sqrt{K})$, where $K$ is the number of iterations. For solving the subproblems of policy evaluation and policy improvement within each iteration of PPO, we establish nonasymptotic upper bounds of the numbers of TD and SGD iterations, respectively. In particular, we prove that, to attain an $\epsilon$ accuracy of policy evaluation and policy improvement, which appears in the constant of the $O(1/\sqrt{K})$ rate of PPO, it suffices to take $O(1/\epsilon^2)$ TD and SGD iterations, respectively.

More specifically, to answer question (i), we cast the infinite-dimensional policy updates in the ideal case as mirror descent iterations. To circumvent the lack of convexity, we prove that the expected total reward satisfies a notation of one-point monotonicity [14], which ensures that the ideal policy sequence evolves towards the optimal policy. In particular, we show that, in the context of infinite-dimensional mirror descent, the exact action-value function plays the role of dual iterate, while the ideal policy plays the role of primal iterate [31, 32, 36]. Such a primal-dual perspective allows us to cast the policy evaluation error in question (ii) as the dual error and the policy improvement error in question (iii) as the primal error. More specifically, the dual and primal errors arise from using neural networks to approximate the exact action-value function and the ideal improved policy, respectively. To characterize such errors in questions (ii) and (iii), we unify the convergence analysis of TD for minimizing the mean squared Bellman error (MSBE) [7] and SGD for minimizing the mean squared error (MSE) [22, 27, 10, 3, 54, 8, 9, 26, 5], both over neural networks. In particular, we show that the desirable representation power and optimization geometry induced by the overparametrization of neural networks enable the global convergence of both the MSBE and MSE, which correspond to the dual and primal errors, at a sublinear rate to zero. By incorporating such errors into the analysis of infinite-dimensional mirror descent, we establish the global rate of convergence of PPO. As a side product, the proof techniques developed here for handling nonconvexity, infinite-dimensionality, semigradient bias, and overparametrization may be of independent interest to the analysis of more general deep reinforcement learning algorithms. In addition, it is worth mentioning that, when the activation functions of neural networks are linear, our results cover the classical setting with linear function approximation, which encompasses the classical tabular setting as a special case.

**More Related Work.** PPO [40] and TRPO [39] are proposed to improve the convergence of vanilla policy gradient [49, 43] in deep reinforcement learning. Related algorithms based on the idea of KL-regularization include natural policy gradient and actor-critic [23, 35], entropy-regularized policy gradient and actor-critic [29], primal-dual actor-critic [12, 11], soft Q-learning and actor-critic [17, 18], and dynamic policy programming [6]. Despite its empirical success, policy optimization generally lacks global convergence guarantees due to nonconvexity. One exception is the recent analysis by [33], which establishes the global convergence of TRPO to the optimal policy. However, [33] require infinite-dimensional policy updates based on exact action-value functions and do not provide the nonasymptotic rate of convergence. In contrast, we allow for the parametrization of both policy and action-value function using neural networks and provide the nonasymptotic rate of PPO as well as the iteration complexity of solving the subproblems of policy improvement and policy evaluation. In particular, based on the primal-dual perspective of reinforcement learning [36], we develop a concise convergence proof of PPO as infinite-dimensional mirror descent under one-point monotonicity, which is of independent interest. In addition, we refer to the closely related concurrent work [2] for the global convergence analysis of (natural) policy gradient for discrete state and action spaces as well as continuous state space with linear function approximation. See also the concurrent work [52], which studies continuous state space with general function approximation, but only es-

tablishes the convergence to a locally optimal policy. In addition, in our companion paper [48], we establish the global convergence of neural (natural) policy gradient.

## 2 Background

In this section, we briefly introduce the general setting of reinforcement learning as well as PPO and TRPO.

**Markov Decision Process.** We consider the Markov decision process $(\mathcal{S}, \mathcal{A}, \mathcal{P}, r, \gamma)$, where $\mathcal{S}$ is a compact state space, $\mathcal{A}$ is a finite action space, $\mathcal{P} : \mathcal{S} \times \mathcal{S} \times \mathcal{A} \to \mathbb{R}$ is the transition kernel, $r : \mathcal{S} \times \mathcal{A} \to \mathbb{R}$ is the reward function, and $\gamma \in (0, 1)$ is the discount factor. We track the performance of a policy $\pi : \mathcal{A} \times \mathcal{S} \to \mathbb{R}$ using its action-value function (Q-function) $Q^\pi : \mathcal{S} \times \mathcal{A} \to \mathbb{R}$, which is defined as

$$Q^\pi(s, a) = (1 - \gamma) \cdot \mathbb{E}\left[ \sum_{t=0}^\infty \gamma^t \cdot r(s_t, a_t) \,\bigg|\, s_0 = s,\ a_0 = a,\ a_t \sim \pi(\cdot \,|\, s_t),\ s_{t+1} \sim \mathcal{P}(\cdot \,|\, s_t, a_t) \right].$$

Correspondingly, the state-value function $V^\pi : \mathcal{S} \to \mathbb{R}$ of a policy $\pi$ is defined as

$$V^\pi(s) = (1 - \gamma) \cdot \mathbb{E}\left[ \sum_{t=0}^\infty \gamma^t \cdot r(s_t, a_t) \,\bigg|\, s_0 = s,\ a_t \sim \pi(\cdot \,|\, s_t),\ s_{t+1} \sim \mathcal{P}(\cdot \,|\, s_t,\ a_t) \right]. \tag{2.1}$$

The advantage function $A^\pi : \mathcal{S} \times \mathcal{A} \to \mathbb{R}$ of a policy $\pi$ is defined as $A^\pi(s, a) = Q^\pi(s, a) - V^\pi(s)$. We denote by $\nu_\pi(s)$ and $\sigma_\pi(s, a) = \pi(a \,|\, s) \cdot \nu_\pi(s)$ the stationary state distribution and the stationary state-action distribution associated with a policy $\pi$, respectively. Correspondingly, we denote by $\mathbb{E}_{\sigma_\pi}[\,\cdot\,]$ and $\mathbb{E}_{\nu_\pi}[\,\cdot\,]$ the expectations $\mathbb{E}_{(s,a)\sim\sigma_\pi}[\,\cdot\,] = \mathbb{E}_{a\sim\pi(\cdot\,|\,s), s\sim\nu_\pi(\cdot)}[\,\cdot\,]$ and $\mathbb{E}_{s\sim\nu_\pi}[\,\cdot\,]$, respectively. Meanwhile, we denote by $\langle\cdot,\cdot\rangle$ the inner product over $\mathcal{A}$, e.g., we have $V^\pi(s) = \mathbb{E}_{a\sim\pi(\cdot\,|\,s)}[Q^\pi(s,a)] = \langle Q^\pi(s,\cdot), \pi(\cdot\,|\,s)\rangle$.

**PPO and TRPO.** At the $k$-th iteration of PPO, the policy parameter $\theta$ is updated by

$$\theta_{k+1} \leftarrow \operatorname*{argmax}_\theta \widehat{\mathbb{E}}\left[ \frac{\pi_\theta(a \,|\, s)}{\pi_{\theta_k}(a \,|\, s)} \cdot A_k(s, a) - \beta_k \cdot \mathrm{KL}(\pi_\theta(\cdot \,|\, s) \,\|\, \pi_{\theta_k}(\cdot \,|\, s)) \right], \tag{2.2}$$

where $A_k$ is an estimator of $A^{\pi_{\theta_k}}$ and $\widehat{\mathbb{E}}[\,\cdot\,]$ is taken with respect to the empirical version of $\sigma_{\pi_{\theta_k}}$, that is, the empirical stationary state-action distribution associated with the current policy $\pi_{\theta_k}$. In practice, the penalty parameter $\beta_k$ is adjusted by line search.

At the $k$-th iteration of TRPO, the policy parameter $\theta$ is updated by

$$\theta_{k+1} \leftarrow \operatorname*{argmax}_\theta \widehat{\mathbb{E}}\left[ \frac{\pi_\theta(a \,|\, s)}{\pi_{\theta_k}(a \,|\, s)} \cdot A_k(s, a) \right], \quad \text{subject to } \mathrm{KL}(\pi_\theta(\cdot \,|\, s) \,\|\, \pi_{\theta_k}(\cdot \,|\, s)) \le \delta, \tag{2.3}$$

where $\delta$ is the radius of the trust region. The PPO update in (2.2) can be viewed as a Lagrangian relaxation of the TRPO update in (2.3) with Lagrangian multiplier $\beta_k$, which implies their updates are equivalent if $\beta_k$ is properly chosen. Without loss of generality, we focus on PPO hereafter.

It is worth mentioning that, compared with the original versions of PPO [40] and TRPO [39], the variants in (2.2) and (2.3) use $\mathrm{KL}(\pi_\theta(\cdot \,|\, s) \,\|\, \pi_{\theta_k}(\cdot \,|\, s))$ instead of $\mathrm{KL}(\pi_{\theta_k}(\cdot \,|\, s) \,\|\, \pi_\theta(\cdot \,|\, s))$. In Sections 3 and 4, we show that, as the original versions, such variants also allow us to approximately obtain the improved policy $\pi_{\theta_{k+1}}$ using SGD, and moreover, enjoy global convergence.

## 3 Neural PPO

We present more details of PPO with policy and action-value function parametrized by neural networks. For notational simplicity, we denote by $\nu_k$ and $\sigma_k$ the stationary state distribution $\nu_{\pi_{\theta_k}}$ and the stationary state-action distribution $\sigma_{\pi_{\theta_k}}$, respectively. Also, we define an auxiliary distribution $\widetilde{\sigma}_k$ over $\mathcal{S} \times \mathcal{A}$ as $\widetilde{\sigma}_k = \nu_k \pi_0$.

**Neural Network Parametrization.** Without loss of generality, we assume that $(s, a) \in \mathbb{R}^d$ for all $s \in \mathcal{S}$ and $a \in \mathcal{A}$. We parametrize a function $u : \mathcal{S} \times \mathcal{A} \to \mathbb{R}$, e.g., policy $\pi$ or action-value function

$Q^\pi$, by the following two-layer neural network, which is denoted by $\mathrm{NN}(\alpha; m)$,

$$u_\alpha(s,a) = \frac{1}{\sqrt{m}} \sum_{i=1}^{m} b_i \cdot \sigma([\alpha]_i^\top (s,a)). \tag{3.1}$$

Here $m$ is the width of the neural network, $b_i \in \{-1, 1\}$ $(i \in [m])$ are the output weights, $\sigma(\cdot)$ is the rectified linear unit (ReLU) activation, and $\alpha = ([\alpha]_1^\top, \ldots, [\alpha]_m^\top)^\top \in \mathbb{R}^{md}$ with $[\alpha]_i \in \mathbb{R}^d$ $(i \in [m])$ are the input weights. We consider the random initialization

$$b_i \overset{\text{i.i.d.}}{\sim} \mathrm{Unif}(\{-1,1\}), \quad [\alpha(0)]_i \overset{\text{i.i.d.}}{\sim} \mathcal{N}(0, I_d/d), \quad \text{for all } i \in [m]. \tag{3.2}$$

We restrict the input weights $\alpha$ to an $\ell_2$-ball centered at the initialization $\alpha(0)$ by the projection $\Pi_{\mathcal{B}^0(R_\alpha)}(\alpha') = \mathrm{argmin}_{\alpha \in \mathcal{B}^0(R_\alpha)}\{\|\alpha - \alpha'\|_2\}$, where $\mathcal{B}^0(R_\alpha) = \{\alpha : \|\alpha - \alpha(0)\|_2 \leq R_\alpha\}$. Throughout training, we only update $\alpha$, while keeping $b_i$ $(i \in [m])$ fixed at the initialization. Hence, we omit the dependency on $b_i$ $(i \in [m])$ in $\mathrm{NN}(\alpha; m)$ and $u_\alpha(s,a)$.

**Policy Improvement.** We consider the population version of the objective function in (2.2),

$$L(\theta) = \mathbb{E}_{\nu_k}\big[\langle Q_{\omega_k}(s,\cdot), \pi_\theta(\cdot \,|\, s)\rangle - \beta_k \cdot \mathrm{KL}(\pi_\theta(\cdot \,|\, s) \,\|\, \pi_{\theta_k}(\cdot \,|\, s))\big], \tag{3.3}$$

where $Q_{\omega_k}$ is an estimator of $Q^{\pi_{\theta_k}}$, that is, the exact action-value function of $\pi_{\theta_k}$. In the following, we convert the subproblem $\max_\theta L(\theta)$ of policy improvement into a least-squares subproblem. We consider the energy-based policy $\pi(a \,|\, s) \propto \exp\{\tau^{-1} f(s,a)\}$, which is abbreviated as $\pi \propto \exp\{\tau^{-1}f\}$. Here $f : \mathcal{S} \times \mathcal{A} \to \mathbb{R}$ is the energy function and $\tau > 0$ is the temperature parameter. We have the following closed form of the ideal infinite-dimensional policy update. See also, e.g., [1] for a Bayesian inference perspective.

**Proposition 3.1.** Let $\pi_{\theta_k} \propto \exp\{\tau_k^{-1} f_{\theta_k}\}$ be an energy-based policy. Given an estimator $Q_{\omega_k}$ of $Q^{\pi_{\theta_k}}$, the update $\widehat{\pi}_{k+1} \leftarrow \mathrm{argmax}_\pi\{\mathbb{E}_{\nu_k}[\langle Q_{\omega_k}(s,\cdot), \pi(\cdot \,|\, s)\rangle - \beta_k \cdot \mathrm{KL}(\pi(\cdot \,|\, s) \,\|\, \pi_{\theta_k}(\cdot \,|\, s))]\}$ gives

$$\widehat{\pi}_{k+1} \propto \exp\{\beta_k^{-1} Q_{\omega_k} + \tau_k^{-1} f_{\theta_k}\}. \tag{3.4}$$

*Proof.* See Appendix C for a detailed proof. □

Here we note that the closed form of ideal infinite-dimensional update in (3.4) holds state-wise. To represent the ideal improved policy $\widehat{\pi}_{k+1}$ in Proposition 3.1 using the energy-based policy $\pi_{\theta_{k+1}} \propto \exp\{\tau_{k+1}^{-1} f_{\theta_{k+1}}\}$, we solve the subproblem of minimizing the MSE,

$$\theta_{k+1} \leftarrow \underset{\theta \in \mathcal{B}^0(R_f)}{\mathrm{argmin}} \mathbb{E}_{\widetilde{\sigma}_k}\big[\big(f_\theta(s,a) - \tau_{k+1} \cdot (\beta_k^{-1} Q_{\omega_k}(s,a) + \tau_k^{-1} f_{\theta_k}(s,a))\big)^2\big], \tag{3.5}$$

which is justified in Appendix B as a majorization of $-L(\theta)$ defined in (3.3). Here we use the neural network parametrization $f_\theta = \mathrm{NN}(\theta; m_f)$ defined in (3.1), where $\theta$ denotes the input weights and $m_f$ is the width. It is worth mentioning that in (3.5) we sample the actions according to $\widetilde{\sigma}_k$ so that $\pi_{\theta_{k+1}}$ approximates the ideal infinite-dimensional policy update in (3.4) evenly well over all actions. Also note that the subproblem in (3.5) allows for off-policy sampling of both states and actions [1].

To solve (3.5), we use the SGD update

$$\theta(t + 1/2) \leftarrow \theta(t) - \eta \cdot \big(f_{\theta(t)}(s,a) - \tau_{k+1} \cdot (\beta_k^{-1} Q_{\omega_k}(s,a) + \tau_k^{-1} f_{\theta_k}(s,a))\big) \cdot \nabla_\theta f_{\theta(t)}(s,a), \tag{3.6}$$

where $(s,a) \sim \widetilde{\sigma}_k$ and $\theta(t + 1) \leftarrow \Pi_{\mathcal{B}^0(R_f)}(\theta(t + 1/2))$. Here $\eta$ is the stepsize. See Appendix A for a detailed algorithm.

**Policy Evaluation.** To obtain the estimator $Q_{\omega_k}$ of $Q^{\pi_{\theta_k}}$ in (3.3), we solve the subproblem of minimizing the MSBE,

$$\omega_k \leftarrow \underset{\omega \in \mathcal{B}^0(R_Q)}{\mathrm{argmin}} \mathbb{E}_{\sigma_k}[(Q_\omega(s,a) - [\mathcal{T}^{\pi_{\theta_k}} Q_\omega](s,a))^2]. \tag{3.7}$$

Here the Bellman evaluation operator $\mathcal{T}^\pi$ of a policy $\pi$ is defined as

$$[\mathcal{T}^\pi Q](s,a) = \mathbb{E}\big[(1 - \gamma) \cdot r(s,a) + \gamma \cdot Q(s',a') \,\big|\, s' \sim \mathcal{P}(\cdot \,|\, s,a), \, a' \sim \pi(\cdot \,|\, s')\big].$$

We use the neural network parametrization $Q_\omega = \mathrm{NN}(\omega; m_Q)$ defined in (3.1), where $\omega$ denotes the input weights and $m_Q$ is the width. To solve (3.7), we use the TD update

$$\omega(t + 1/2) \leftarrow \omega(t) - \eta \cdot \big(Q_{\omega(t)}(s,a) - (1 - \gamma) \cdot r(s,a) - \gamma \cdot Q_{\omega(t)}(s',a')\big) \cdot \nabla_\omega Q_{\omega(t)}(s,a), \tag{3.8}$$

where $(s,a) \sim \sigma_k$, $s' \sim \mathcal{P}(\cdot \,|\, s,a)$, $a' \sim \pi_{\theta_k}(\cdot \,|\, s')$, and $\omega(t + 1) = \Pi_{\mathcal{B}^0(R_Q)}(\omega(t + 1/2))$. Here $\eta$ is the stepsize. See Appendix A for a detailed algorithm.

**Neural PPO.** By assembling the subproblems of policy improvement and policy evaluation, we present neural PPO in Algorithm 1, which is characterized in Section 4.

---

**Algorithm 1** Neural PPO

---

**Require:** MDP $(\mathcal{S}, \mathcal{A}, \mathcal{P}, r, \gamma)$, penalty parameter $\beta$, widths $m_f$ and $m_Q$, number of SGD and TD iterations $T$, number of TRPO iterations $K$, and projection radii $R_f \geq R_Q$

1: Initialize with uniform policy: $\tau_0 \leftarrow 1$, $f_{\theta_0} \leftarrow 0$, $\pi_{\theta_0} \leftarrow \pi_0 \propto \exp\{\tau_0^{-1} f_{\theta_0}\}$
2: **for** $k = 0, \ldots, K - 1$ **do**
3:      Set temperature parameter $\tau_{k+1} \leftarrow \beta\sqrt{K}/(k+1)$ and penalty parameter $\beta_k \leftarrow \beta\sqrt{K}$
4:      Sample $\{(s_t, a_t, a_t^0, s_t', a_t')\}_{t=1}^T$ with $(s_t, a_t) \sim \sigma_k$, $a_t^0 \sim \pi_0(\cdot \,|\, s_t)$, $s_t' \sim \mathcal{P}(\cdot \,|\, s_t, a_t)$ and $a_t' \sim \pi_{\theta_k}(\cdot \,|\, s_t')$
5:      Solve for $Q_{\omega_k} = \text{NN}(\omega_k; m_Q)$ in (3.7) using the TD update in (3.8) (Algorithm 3)
6:      Solve for $f_{\theta_{k+1}} = \text{NN}(\theta_{k+1}; m_f)$ in (3.5) using the SGD update in (3.6) (Algorithm 2)
7:      Update policy: $\pi_{\theta_{k+1}} \propto \exp\{\tau_{k+1}^{-1} f_{\theta_{k+1}}\}$
8: **end for**

---

## 4 Main Results

In this section, we establish the global convergence of neural PPO in Algorithm 1 based on characterizing the errors arising from solving the subproblems of policy improvement and policy evaluation in (3.5) and (3.7), respectively.

Our analysis relies on the following regularity condition on the boundedness of reward.

**Assumption 4.1** (Bounded Reward). There exists a constant $R_{\max} > 0$ such that $R_{\max} = \sup_{(s,a) \in \mathcal{S} \times \mathcal{A}} |r(s,a)|$, which implies $|V^\pi(s)| \leq R_{\max}$ and $|Q^\pi(s,a)| \leq R_{\max}$ for any policy $\pi$.

To ensure the compatibility between the policy and the action-value function [25, 43, 23, 35, 46, 47], we set $m_f = m_Q$ and use the following random initialization. In Algorithm 1, we first generate according to (3.2) the random initialization $\alpha(0) = \theta(0) = \omega(0)$ and $b_i$ ($i \in [m]$), and then use it as the fixed initialization of both SGD and TD in Lines 6 and 5 of Algorithm 1 for all $k \in [K]$, respectively.

### 4.1 Errors of Policy Improvement and Policy Evaluation

We define the following function class, which characterizes the representation power of the neural network defined in (3.1).

**Definition 4.2.** For any constant $R > 0$, we define the function class

$$\mathcal{F}_{R,m} = \left\{ \frac{1}{\sqrt{m}} \sum_{i=1}^m b_i \cdot \mathbb{1}\{[\alpha(0)]_i^\top (s,a) > 0\} \cdot [\alpha]_i^\top (s,a) : \|\alpha - \alpha(0)\|_2 \leq R \right\},$$

where $[\alpha(0)]_i$ and $b_i$ ($i \in [m]$) are the random initialization defined in (3.2).

As $m \to \infty$, $\mathcal{F}_{R,m} - \text{NN}(\alpha(0); m)$ approximates a subset of the reproducing kernel Hilbert space (RKHS) induced by the kernel $K(x,y) = \mathbb{E}_{z \sim N(0, I_d/d)}[\mathbb{1}\{z^\top x > 0, z^\top y > 0\} x^\top y]$ [22, 27, 10, 3, 54, 8, 9, 26, 5, 7]. Such a subset is a ball with radius $R$ in the corresponding $\mathcal{H}$-norm, which is known to be a rich function class [20]. Correspondingly, for a sufficiently large width $m$ and radius $R$, $\mathcal{F}_{R,m}$ is also a sufficiently rich function class.

Based on Definition 4.2, we lay out the following regularity condition on the action-value function class.

**Assumption 4.3** (Action-Value Function Class). It holds that $Q^\pi(s,a) \in \mathcal{F}_{R_Q, m_Q}$ for any $\pi$.

Assumption 4.3 states that $\mathcal{F}_{R_Q, m_Q}$ is closed under the Bellman evaluation operator $\mathcal{T}^\pi$, as $Q^\pi$ is the fixed-point solution of the Bellman equation $\mathcal{T}^\pi Q^\pi = Q^\pi$. Such a regularity condition is commonly used in the literature [30, 4, 16, 15, 45, 51]. In particular, [50] define a class of Markov decision processes that satisfy such a regularity condition, which is sufficiently rich due to the representation power of $\mathcal{F}_{R_Q, m_Q}$.

In the sequel, we lay out another regularity condition on the stationary state-action distribution $\sigma_\pi$.

**Assumption 4.4** (Regularity of Stationary Distribution)**.** There exists a constant $c > 0$ such that for any vector $z \in \mathbb{R}^d$ and $\zeta > 0$, it holds almost surely that $\mathbb{E}_{\sigma_\pi}[\mathbb{1}\{|z^\top(s,a)| \le \zeta\} \,|\, z] \le c \cdot \zeta/\|z\|_2$ for any $\pi$.

Assumption 4.4 states that the density of $\sigma_\pi$ is sufficiently regular. Such a regularity condition holds as long as the stationary state distribution $\nu_\pi$ has upper bounded density.

We are now ready present bounds for errors induced by approximation via two-layer neural networks, with analysis generalizing those of [7, 5] included in Appendix D. First, we characterize the policy improvement error, which is induced by solving the subproblem in (3.5) using the SGD update in (3.6), in the following theorem. See Line 6 of Algorithm 1 and Algorithm 2 for a detailed algorithm.

**Theorem 4.5** (Policy Improvement Error)**.** Suppose that Assumptions 4.1, 4.3, and 4.4 hold. We set $T \ge 64$ and the stepsize to be $\eta = T^{-1/2}$. Within the $k$-th iteration of Algorithm 1, the output $f_{\bar{\theta}}$ of Algorithm 2 satisfies

$$\mathbb{E}_{\text{init},\widetilde{\sigma}_k}\Big[\big(f_{\bar{\theta}}(s,a) - \tau_{k+1} \cdot (\beta_k^{-1} Q_{\omega_k}(s,a) + \tau_k^{-1} f_{\theta_k}(s,a))\big)^2\Big]$$
$$= O(R_f^2 T^{-1/2} + R_f^{5/2} m_f^{-1/4} + R_f^3 m_f^{-1/2}).$$

*Proof.* See Appendix D for a detailed proof. $\qquad\square$

Similarly, we characterize the policy evaluation error, which is induced by solving the subproblem in (3.7) using the TD update in (3.8), in the following theorem. See Line 5 of Algorithm 1 and Algorithm 3 for a detailed algorithm.

**Theorem 4.6** (Policy Evaluation Error)**.** Suppose that Assumptions 4.1, 4.3, and 4.4 hold. We set $T \ge 64/(1-\gamma)^2$ and the stepsize to be $\eta = T^{-1/2}$. Within the $k$-th iteration of Algorithm 1, the output $Q_{\overline{\omega}}$ of Algorithm 3 satisfies

$$\mathbb{E}_{\text{init},\sigma_k}[(Q_{\overline{\omega}}(s,a) - Q^{\pi_{\theta_k}}(s,a))^2] = O(R_Q^2 T^{-1/2} + R_Q^{5/2} m_Q^{-1/4} + R_Q^3 m_Q^{-1/2}).$$

*Proof.* See Appendix D for a detailed proof. $\qquad\square$

As we show in Sections 4.3 and 5, Theorems 4.5 and 4.6 characterize the primal and dual errors of the infinite-dimensional mirror descent corresponding to neural PPO. In particular, such errors decay to zero at the rate of $1/\sqrt{T}$ when the width $m_f = m_Q$ is sufficiently large, where $T$ is the number of TD and SGD iterations in Algorithm 1. For notational simplicity, we omit the dependency on the random initialization in the expectations hereafter.

## 4.2 Error Propagation

We denote by $\pi^*$ the optimal policy with $\nu^*$ being its stationary state distribution and $\sigma^*$ being its stationary state-action distribution. Recall that, as defined in (3.4), $\widehat{\pi}_{k+1}$ is the ideal improved policy based on $Q_{\omega_k}$, which is an estimator of the exact action-value function $Q^{\pi_{\theta_k}}$. Correspondingly, we define the ideal improved policy based on $Q^{\pi_{\theta_k}}$ as

$$\pi_{k+1} = \underset{\pi}{\arg\max}\big\{\mathbb{E}_{\nu_k}\big[\langle Q^{\pi_{\theta_k}}(s,\cdot), \pi(\cdot,s)\rangle - \beta_k \cdot \text{KL}(\pi(\cdot\,|\,s)\,\|\,\pi_{\theta_k}(\cdot\,|\,s))\big]\big\}. \qquad (4.1)$$

By the same proof of Proposition 3.1, we have $\pi_{k+1} \propto \exp\{\beta_k^{-1} Q^{\pi_{\theta_k}} + \tau_k^{-1} f_{\theta_k}\}$, which is also an energy-based policy.

We define the following quantities related to density ratios between policies or stationary distributions,

$$\phi_k^* = \mathbb{E}_{\widetilde{\sigma}_k}[|\mathrm{d}\sigma^*/\mathrm{d}\widetilde{\sigma}_k - \mathrm{d}(\pi_{\theta_k}\nu^*)/\mathrm{d}\widetilde{\sigma}_k|^2]^{1/2}, \quad \psi_k^* = \mathbb{E}_{\sigma_k}[|\mathrm{d}\sigma^*/\mathrm{d}\sigma_k - \mathrm{d}\nu^*/\mathrm{d}\nu_k|^2]^{1/2}, \quad (4.2)$$

where $\mathrm{d}\sigma^*/\mathrm{d}\widetilde{\sigma}_k$, $\mathrm{d}(\pi_{\theta_k}\nu^*)/\mathrm{d}\widetilde{\sigma}_k$, $\mathrm{d}\sigma^*/\mathrm{d}\sigma_k$, and $\mathrm{d}\nu^*/\mathrm{d}\nu_k$ are the Radon-Nikodym derivatives. A closely related quantity known as the concentrability coefficient is commonly used in the literature [30, 4, 16, 45, 51]. In comparison, as our analysis is based on stationary distributions, our definitions of $\phi_k^*$ and $\psi_k^*$ are simpler in that they do not require unrolling the state-action sequence. Then we have the following lemma that quantifies how the errors of policy improvement and policy evaluation propagate into the infinite-dimensional policy space.

**Lemma 4.7** (Error Propagation). Suppose that the policy improvement error in Line 6 of Algorithm 1 satisfies

$$\mathbb{E}_{\widetilde{\sigma}_k}\big[\big(f_{\theta_{k+1}}(s,a) - \tau_{k+1} \cdot (\beta_k^{-1} Q_{\omega_k}(s,a) - \tau_k^{-1} f_{\theta_k}(s,a))\big)^2\big] \le \epsilon_{k+1}, \qquad (4.3)$$

and the policy evaluation error in Line 5 of Algorithm 1 satisfies

$$\mathbb{E}_{\sigma_k}[(Q_{\omega_k}(s,a) - Q^{\pi_{\theta_k}}(s,a))^2] \le \epsilon_k'. \qquad (4.4)$$

For $\pi_{k+1}$ defined in (4.1) and $\pi_{\theta_{k+1}}$ obtained in Line 7 of Algorithm 1, we have

$$\big|\mathbb{E}_{\nu^*}\big[\langle \log(\pi_{\theta_{k+1}}(\cdot\,|\,s)/\pi_{k+1}(\cdot\,|\,s)), \pi^*(\cdot\,|\,s) - \pi_{\theta_k}(\cdot\,|\,s)\rangle\big]\big| \le \varepsilon_k, \qquad (4.5)$$

where $\varepsilon_k = \tau_{k+1}^{-1}\epsilon_{k+1} \cdot \phi_{k+1}^* + \beta_k^{-1}\epsilon_k' \cdot \psi_k^*$.

*Proof.* See Appendix E for a detailed proof. $\qquad\square$

Lemma 4.7 quantifies the difference between the ideal case, where we use the infinite-dimensional policy update based on the exact action-value function, and the realistic case, where we use the neural networks defined in (3.1) to approximate the exact action-value function and the ideal improved policy.

The following lemma characterizes the difference between $f_{\theta_{k+1}}$ and $f_{\theta_k}$.

**Lemma 4.8** (Stepwise Energy Difference). Under the same conditions of Lemma 4.7, we have

$$\mathbb{E}_{\nu^*}[\|\tau_{k+1}^{-1} f_{\theta_{k+1}}(s,\cdot) - \tau_k^{-1} f_{\theta_k}(s,\cdot)\|_\infty^2] \le 2\varepsilon_k' + 2\beta_k^{-2} M,$$

where $\varepsilon_k' = |\mathcal{A}| \cdot \tau_{k+1}^{-2}\epsilon_{k+1}^2$ and $M = 2\mathbb{E}_{\nu^*}[\max_{a\in\mathcal{A}}(Q_{\omega_0}(s,a))^2] + 2R_f^2$.

*Proof.* See Appendix E for a detailed proof. $\qquad\square$

Intuitively, the bounded difference between $f_{\theta_{k+1}}$ and $f_{\theta_{k+1}}$ quantified in Lemma 4.8 is due to the KL-regularization in (3.3), which keeps the updated policy $\pi_{\theta_{k+1}}$ from being too far away from the current policy $\pi_{\theta_k}$.

The differences characterized in Lemmas 4.7 and 4.8 play key roles in establishing the global convergence of neural PPO.

## 4.3 Global Convergence of Neural PPO

We track the progress of neural PPO in Algorithm 1 using the expected total reward

$$\mathcal{L}(\pi) = \mathbb{E}_{\nu^*}[V^\pi(s)] = \mathbb{E}_{\nu^*}[\langle Q^\pi(s,\cdot), \pi(\cdot\,|\,s)\rangle], \qquad (4.6)$$

where $\nu^*$ is the stationary state distribution of the optimal policy $\pi^*$. The following theorem characterizes the global convergence of $\mathcal{L}(\pi_{\theta_k})$ towards $\mathcal{L}(\pi^*)$. Recall that $T_f$ and $T_Q$ are the numbers of SGD and TD iterations in Lines 6 and 5 of Algorithm 1, while $\phi_k^*$ and $\psi_k^*$ are defined in (4.2).

**Theorem 4.9** (Global Rate of Convergence of Neural PPO). Suppose that Assumptions 4.1, 4.3, and 4.4 hold. For the policy sequence $\{\pi_{\theta_k}\}_{k=1}^K$ attained by neural PPO in Algorithm 1, we have

$$\min_{0\le k\le K}\big\{\mathcal{L}(\pi^*) - \mathcal{L}(\pi_{\theta_k})\big\} \le \frac{\beta^2 \log|\mathcal{A}| + M + \beta^2 \sum_{k=0}^{K-1}(\varepsilon_k + \varepsilon_k')}{(1-\gamma)\cdot\beta\cdot\sqrt{K}}.$$

Here $\varepsilon_k = \tau_{k+1}^{-1}\epsilon_{k+1} \cdot \phi_k^* + \beta_k^{-1}\epsilon_k' \cdot \psi_k^*$ and $\varepsilon_k' = |\mathcal{A}| \cdot \tau_{k+1}^{-2}\epsilon_{k+1}^2$, where

$$\epsilon_{k+1} = O(R_f^2 T^{-1/2} + R_f^{5/2} m_f^{-1/4} + R_f^3 m_f^{-1/2}), \quad \epsilon_k' = O(R_Q^2 T^{-1/2} + R_Q^{5/2} m_Q^{-1/4} + R_Q^3 m_Q^{-1/2}).$$

Also, we have $M = 2\mathbb{E}_{\nu^*}[\max_{a\in\mathcal{A}}(Q_{\omega_0}(s,a))^2] + 2R_f^2$.

*Proof.* See Section 5 for a detailed proof of Theorem 4.9. The key to our proof is the global convergence of infinite-dimensional mirror descent with errors under one-point monotonicity, where the primal and dual errors are characterized by Theorems 4.5 and 4.6, respectively. $\qquad\square$

To understand Theorem 4.9, we consider the infinite-dimensional policy update based on the exact action-value function, that is, $\epsilon_{k+1} = \epsilon_k' = 0$ for any $k+1 \in [K]$. In such an ideal case, by Theorem 4.9, neural PPO globally converges to the optimal policy $\pi^*$ at the rate of

$$\min_{0\le k\le K}\big\{\mathcal{L}(\pi^*) - \mathcal{L}(\pi_{\theta_k})\big\} \le \frac{2\sqrt{M \log|\mathcal{A}|}}{(1-\gamma)\cdot\sqrt{K}},$$

with the optimal choice of the penalty parameter $\beta_k = \sqrt{MK/\log|\mathcal{A}|}$.

Note that Theorem 4.9 sheds light on the difficulty of choosing the optimal penalty coefficient in practice, which is observed by [40]. In particular, the optimal choice of $\beta$ in $\beta_k = \beta\sqrt{K}$ is given by

$$\beta = \frac{\sqrt{M}}{\sqrt{\log|\mathcal{A}| + \sum_{k=0}^{K-1}(\varepsilon_k + \varepsilon_k')}},$$

where $M$ and $\sum_{k=0}^{K-1}(\varepsilon_k + \varepsilon_k')$ may vary across different deep reinforcement learning problems. As a result, line search is often needed in practice.

To better understand Theorem 4.9, the following corollary quantifies the minimum width $m_f$ and $m_Q$ and the minimum number of SGD and TD iterations $T$ that ensure the $O(1/\sqrt{K})$ rate of convergence.

**Corollary 4.10** (Iteration Complexity of Subproblems and Minimum Widths of Neural Networks). Suppose that Assumptions 4.1, 4.3, and 4.4 hold. Let $m_f = \Omega(K^6 R_f^{10} \cdot \phi_k^{*4} + K^4 R_f^{10} \cdot |\mathcal{A}|^2)$, $m_Q = \Omega(K^2 R_Q^{10} \cdot \psi_k^{*4})$, and $T = \Omega(K^3 R_f^4 \cdot \phi_k^{*2} + K^2 R_f^4 \cdot |\mathcal{A}| + K R_Q^4 \cdot \psi_k^{*2})$ for any $0 \le k \le K$. We have

$$\min_{0 \le k \le K}\{\mathcal{L}(\pi^*) - \mathcal{L}(\pi_{\theta_k})\} \le \frac{\beta^2 \log|\mathcal{A}| + M + O(1)}{(1-\gamma)\beta \cdot \sqrt{K}}.$$

*Proof.* See Appendix F for a detailed proof. ∎

The difference between the requirements on the widths $m_f$ and $m_Q$ in Corollary 4.10 suggests that the errors of policy improvement and policy evaluation play distinct roles in the global convergence of neural PPO. In fact, Theorem 4.9 depends on the total error $\tau_{k+1}^{-1}\epsilon_{k+1} \cdot \phi_k^* + \beta_k^{-1}\epsilon_k' \cdot \psi_k^* + |\mathcal{A}| \cdot \tau_{k+1}^{-2}\epsilon_{k+1}^2$, where the weight $\tau_{k+1}^{-1}$ of the policy improvement error $\epsilon_{k+1}$ is much larger than the weight $\beta_k^{-1}$ of the policy evaluation error $\epsilon_k'$, and $|\mathcal{A}| \cdot \tau_{k+1}^{-2}\epsilon_{k+1}^2$ is a high-order term when $\epsilon_{k+1}$ is sufficiently small. In other words, the policy improvement error plays a more important role.

## 5  Proof Sketch

In this section, we sketch the proof of Theorem 4.9. In detail, we cast neural PPO in Algorithm 1 as infinite-dimensional mirror descent with primal and dual errors and exploit a notion of one-point monotonicity to establish its global convergence.

We first present the performance difference lemma of [24]. Recall that the expected total reward $\mathcal{L}(\pi)$ is defined in (4.6) and $\nu^*$ is the stationary state distribution of the optimal policy $\pi^*$.

**Lemma 5.1** (Performance Difference). For $\mathcal{L}(\pi)$ defined in (4.6), we have
$$\mathcal{L}(\pi) - \mathcal{L}(\pi^*) = (1-\gamma)^{-1} \cdot \mathbb{E}_{\nu^*}[\langle Q^\pi(s,\cdot), \pi(\cdot\,|\,s) - \pi^*(\cdot\,|\,s)\rangle].$$

*Proof.* See Appendix G for a detailed proof. ∎

Since the optimal policy $\pi^*$ maximizes the value function $V^\pi(s)$ with respect to $\pi$ for any $s \in \mathcal{S}$, we have $\mathcal{L}(\pi^*) = \mathbb{E}_{\nu^*}[V^{\pi^*}(s)] \ge \mathbb{E}_{\nu^*}[V^\pi(s)] = \mathcal{L}(\pi)$ for any $\pi$. As a result, we have
$$\mathbb{E}_{\nu^*}[\langle Q^\pi(s,\cdot), \pi(\cdot\,|\,s) - \pi^*(\cdot\,|\,s)\rangle] \le 0, \quad \text{for any } \pi. \tag{5.1}$$
Under the variational inequality framework [14], (5.1) corresponds to the monotonicity of the mapping $Q^\pi$ evaluated at $\pi^*$ and any $\pi$. Note that the classical notion of monotonicity requires the evaluation at any pair $\pi'$ and $\pi$, while we restrict $\pi'$ to $\pi^*$ in (5.1). Hence, we refer to (5.1) as one-point monotonicity. In the context of nonconvex optimization, the mapping $Q^\pi$ can be viewed as the gradient of $\mathcal{L}(\pi)$ at $\pi$, which lives in the dual space, while $\pi$ lives in the primal space. Another condition related to (5.1) in nonconvex optimization is known as dissipativity [53].

The following lemma establishes the one-step descent of the KL-divergence in the infinite-dimensional policy space, which follows from the analysis of mirror descent [31, 32] as well as the fact that given any $\nu_k$, the subproblem of policy improvement in (4.1) can be solved for each $s \in \mathcal{S}$ individually.

**Lemma 5.2** (One-Step Descent). For the ideal improved policy $\pi_{k+1}$ defined in (4.1) and the current policy $\pi_{\theta_k}$, we have that, for any $s \in \mathcal{S}$,

$$\mathrm{KL}(\pi^*(\cdot \,|\, s) \,\|\, \pi_{\theta_{k+1}}(\cdot \,|\, s)) - \mathrm{KL}(\pi^*(\cdot \,|\, s) \,\|\, \pi_{\theta_k}(\cdot \,|\, s))$$

$$\leq \langle \log(\pi_{\theta_{k+1}}(\cdot \,|\, s)/\pi_{k+1}(\cdot \,|\, s)), \pi_{\theta_k}(\cdot \,|\, s) - \pi^*(\cdot \,|\, s) \rangle - \beta_k^{-1} \cdot \langle Q^{\pi_{\theta_k}}(s, \cdot), \pi^*(\cdot \,|\, s) - \pi_{\theta_k}(\cdot \,|\, s) \rangle$$

$$- 1/2 \cdot \|\pi_{\theta_{k+1}}(\cdot \,|\, s) - \pi_{\theta_k}(\cdot \,|\, s)\|_1^2 - \langle \tau_{k+1}^{-1} f_{\theta_{k+1}}(s, \cdot) - \tau_k^{-1} f_{\theta_k}(s, \cdot), \pi_{\theta_k}(\cdot \,|\, s) - \pi_{\theta_{k+1}}(\cdot \,|\, s) \rangle.$$

*Proof.* See Appendix G for a detailed proof. $\qquad\square$

Based on Lemmas 5.1 and 5.2, we prove Theorem 4.9 by casting neural PPO as infinite-dimensional mirror descent with primal and dual errors, whose impact is characterized in Lemma 4.7. In particular, we employ the $\ell_1$-$\ell_\infty$ pair of primal-dual norms.

*Proof of Theorem 4.9.* Taking expectation with respect to $s \sim \nu^*$ and invoking Lemmas 4.7 and 5.2, we have

$$\mathbb{E}_{\nu^*}[\mathrm{KL}(\pi^*(\cdot \,|\, s) \,\|\, \pi_{\theta_{k+1}}(\cdot \,|\, s))] - \mathbb{E}_{\nu^*}[\mathrm{KL}(\pi^*(\cdot \,|\, s) \,\|\, \pi_{\theta_k}(\cdot \,|\, s))]$$

$$\leq \varepsilon_k - \beta_k^{-1} \cdot \mathbb{E}_{\nu^*}[\langle Q^{\pi_{\theta_k}}(s, \cdot), \pi^*(\cdot \,|\, s) - \pi_{\theta_k}(\cdot \,|\, s) \rangle] - 1/2 \cdot \mathbb{E}_{\nu^*}[\|\pi_{\theta_{k+1}}(\cdot \,|\, s) - \pi_{\theta_k}(\cdot \,|\, s)\|_1^2]$$

$$- \mathbb{E}_{\nu^*}[\langle \tau_{k+1}^{-1} f_{\theta_{k+1}}(s, \cdot) - \tau_k^{-1} f_{\theta_k}(s, \cdot), \pi_{\theta_k}(\cdot \,|\, s) - \pi_{\theta_{k+1}}(\cdot \,|\, s) \rangle].$$

By Lemma 5.1 and the Hölder's inequality, we further have

$$\mathbb{E}_{\nu^*}[\mathrm{KL}(\pi^*(\cdot \,|\, s) \,\|\, \pi_{\theta_{k+1}}(\cdot \,|\, s))] - \mathbb{E}_{\nu^*}[\mathrm{KL}(\pi^*(\cdot \,|\, s) \,\|\, \pi_{\theta_k}(\cdot \,|\, s))]$$

$$\leq \varepsilon_k - (1-\gamma)\beta_k^{-1} \cdot (\mathcal{L}(\pi^*) - \mathcal{L}(\pi_{\theta_k})) - 1/2 \cdot \mathbb{E}_{\nu^*}[\|\pi_{\theta_{k+1}}(\cdot \,|\, s) - \pi_{\theta_k}(\cdot \,|\, s)\|_1^2]$$

$$+ \mathbb{E}_{\nu^*}\big[\|\tau_{k+1}^{-1} f_{\theta_{k+1}}(s, \cdot) - \tau_k^{-1} f_{\theta_k}(s, \cdot)\|_\infty \cdot \|\pi_{\theta_k}(\cdot \,|\, s) - \pi_{\theta_{k+1}}(\cdot \,|\, s)\|_1\big]$$

$$\leq \varepsilon_k - (1-\gamma)\beta_k^{-1} \cdot (\mathcal{L}(\pi^*) - \mathcal{L}(\pi_{\theta_k})) + 1/2 \cdot \mathbb{E}_{\nu^*}[\|\tau_{k+1}^{-1} f_{\theta_{k+1}}(s, \cdot) - \tau_k^{-1} f_{\theta_k}(s, \cdot)\|_\infty^2]$$

$$\leq \varepsilon_k - (1-\gamma)\beta_k^{-1} \cdot (\mathcal{L}(\pi^*) - \mathcal{L}(\pi_{\theta_k})) + (\varepsilon_k' + \beta_k^{-2}M), \tag{5.2}$$

where in the second inequality we use $2xy - y^2 \leq x^2$ and in the last inequality we use Lemma 4.8. Rearranging the terms in (5.2), we have

$$(1-\gamma)\beta_k^{-1} \cdot (\mathcal{L}(\pi^*) - \mathcal{L}(\pi_{\theta_k})) \tag{5.3}$$

$$\leq \mathbb{E}_{\nu^*}[\mathrm{KL}(\pi^*(\cdot \,|\, s) \,\|\, \pi_{\theta_{k+1}}(\cdot \,|\, s))] - \mathbb{E}_{\nu^*}[\mathrm{KL}(\pi^*(\cdot \,|\, s) \,\|\, \pi_{\theta_k}(\cdot \,|\, s))] + \beta_k^{-2}M + \varepsilon_k + \varepsilon_k'.$$

Telescoping (5.3) for $k + 1 \in [K]$, we obtain

$$\sum_{k=0}^{K-1} (1-\gamma)\beta_k^{-1} \cdot (\mathcal{L}(\pi_{\theta_k}) - \mathcal{L}(\pi^*))$$

$$\leq \mathbb{E}_{\nu^*}[\mathrm{KL}(\pi^*(\cdot \,|\, s) \,\|\, \pi_{\theta_K}(\cdot \,|\, s))] - \mathbb{E}_{\nu^*}[\mathrm{KL}(\pi^*(\cdot \,|\, s) \,\|\, \pi_{\theta_0}(\cdot \,|\, s))]$$

$$+ M \sum_{k=0}^{K-1} \beta_k^{-2} + \sum_{k=0}^{K-1} (\varepsilon_k + \varepsilon_k').$$

Note that we have (i) $\sum_{k=0}^{K-1} \beta_k^{-1} \cdot (\mathcal{L}(\pi^*) - \mathcal{L}(\pi_{\theta_k})) \geq (\sum_{k=0}^{K-1} \beta_k^{-1}) \cdot \min_{0 \leq k \leq K}\{\mathcal{L}(\pi^*) - \mathcal{L}(\pi_{\theta_k})\}$, (ii) $\mathbb{E}_{\nu^*}[\mathrm{KL}(\pi^*(\cdot \,|\, s) \,\|\, \pi_{\theta_0}(\cdot \,|\, s))] \leq \log|\mathcal{A}|$ due to the uniform initialization of policy, and that (iii) the KL-divergence is nonnegative. Hence, we have

$$\min_{0 \leq k \leq K} \big\{\mathcal{L}(\pi^*) - \mathcal{L}(\pi_{\theta_k})\big\} \leq \frac{\log|\mathcal{A}| + M\sum_{k=0}^{K-1}\beta_k^{-2} + \sum_{k=0}^{K-1}(\varepsilon_k + \varepsilon_k')}{(1-\gamma)\sum_{k=0}^{K-1}\beta_k^{-1}}. \tag{5.4}$$

Setting the penalty parameter $\beta_k = \beta\sqrt{K}$, we have $\sum_{k=0}^{K-1}\beta_k^{-1} = \beta^{-1}\sqrt{K}$ and $\sum_{k=0}^{K-1}\beta_k^{-2} = \beta^{-2}$, which together with (5.4) concludes the proof of Theorem 4.9. $\qquad\square$

## Acknowledgement

The authors thank Jason D. Lee, Chi Jin, and Yu Bai for enlightening discussions throughout this project.

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
