[Supplementary Material 1]

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

2: Set stepsize $\eta \leftarrow T^{-1/2}$
3: **for** $t = 0, \ldots, T-1$ **do**
4: $\quad (s, a) \leftarrow (s_{t+1}, a_{t+1}^0)$
5: $\quad \theta(t+1/2) \leftarrow \theta(t) - \eta \cdot \left( f_{\theta(t)}(s, a) - \tau_{k+1} \cdot (\beta_k^{-1} Q_{\omega_k}(s, a) + \tau_k^{-1} f_{\theta_k}(s, a)) \right) \cdot \nabla_\theta f_{\theta(t)}(s, a)$
6: $\quad \theta(t+1) \leftarrow \operatorname{argmin}_{\theta \in \mathcal{B}^0(R_f)} \{\|\theta - \theta(t+1/2)\|_2\}$
7: **end for**
8: Average over path $\overline{\theta} \leftarrow 1/T \cdot \sum_{t=0}^{T-1} \theta(t)$
9: **Output:** $f_{\overline{\theta}}$

---

---

**Algorithm 3** Policy Evaluation via TD

---

1: **Require:** MDP $(\mathcal{S}, \mathcal{A}, \mathcal{P}, r, \gamma)$, initial weights $b_i$, $[\omega(0)]_i$ ($i \in [m_Q]$), number of iterations $T$, sample $\{(s_t, a_t, s_t', a_t')\}_{t=1}^T$
2: Set stepsize $\eta \leftarrow T^{-1/2}$
3: **for** $t = 0, \ldots, T-1$ **do**
4: $\quad (s, a, s', a') \leftarrow (s_{t+1}, a_{t+1}, s_{t+1}', a_{t+1}')$
5: $\quad \omega(t+1/2) \leftarrow \omega(t) - \eta \cdot \left( Q_{\omega(t)}(s, a) - (1-\gamma) \cdot r(s, a) - \gamma Q_{\omega(t)}(s', a') \right) \cdot \nabla_\omega Q_{\omega(t)}(s, a)$
6: $\quad \omega(t+1) \leftarrow \operatorname{argmin}_{\omega \in \mathcal{B}^0(R_Q)} \{\|\omega - \omega(t+1/2)\|_2\}$
7: **end for**
8: Average over path $\overline{\omega} \leftarrow 1/T \cdot \sum_{t=0}^{T-1} \omega(t)$
9: **Output:** $Q_{\overline{\omega}}$

---

# B  Supplementary Lemma in Section 3

The following lemma quantifies the policy improvement error in terms of the distance between polices, which is induced by solving (3.5).

**Lemma B.1.** Suppose that $\pi_{\theta_{k+1}} \propto \exp\{\tau_{k+1}^{-1} f_{\theta_{k+1}}\}$ satisfies

$$\mathbb{E}_{\widetilde{\sigma}_k} \left[ \left( f_{\theta_{k+1}}(s, a) - \tau_{k+1} \cdot (\beta_k^{-1} Q_{\omega_k}(s, a) + \tau_k^{-1} f_{\theta_k}(s, a)) \right)^2 \right] \leq \epsilon_{k+1}.$$

We have

$$\mathbb{E}_{\widetilde{\sigma}_k}[(\pi_{\theta_{k+1}}(a \mid s) - \widehat{\pi}_{k+1}(a \mid s))^2] \leq \tau_{k+1}^{-2} \epsilon_{k+1}/16,$$

where $\widehat{\pi}_{k+1}$ is defined in (3.4).

*Proof.* Let $\tau_{k+1}^{-1} \widehat{f}_{k+1} = \beta_k^{-1} Q_{\omega_k} + \tau_k^{-1} f_{\theta_k}$. Since an energy-based policy $\pi \propto \exp\{\tau^{-1} f\}$ is continuous with respect to $f$, by the mean value theorem, we have

$$|\pi_{\theta_{k+1}}(a \mid s) - \widehat{\pi}_{k+1}(a \mid s)| = \left| \frac{\exp\{\tau_{k+1}^{-1} f_{\theta_{k+1}}(s, a)\}}{\sum_{a' \in \mathcal{A}} \exp\{\tau_{k+1}^{-1} f_{\theta_{k+1}}(s, a')\}} - \frac{\exp\{\tau_{k+1}^{-1} \widehat{f}_{k+1}(s, a)\}}{\sum_{a' \in \mathcal{A}} \exp\{\tau_{k+1}^{-1} \widehat{f}_{k+1}(s, a')\}} \right|$$

$$= \left| \frac{\partial}{\partial f(s, a)} \left( \frac{\exp\{\tau_{k+1}^{-1} \widetilde{f}(s, a)\}}{\sum_{a' \in \mathcal{A}} \exp\{\tau_{k+1}^{-1} \widetilde{f}(s, a')\}} \right) \right| \cdot |f_{\theta_{k+1}}(s, a) - \widehat{f}_{k+1}(s, a)|,$$

where $\widetilde{f}$ is a function determined by $f_{\theta_{k+1}}$ and $\widehat{f}_{k+1}$. Furthermore, we have

$$\left| \frac{\partial}{\partial f(s, a)} \left( \frac{\exp\{\tau_{k+1}^{-1} f(s, a)\}}{\sum_{a' \in \mathcal{A}} \exp\{\tau_{k+1}^{-1} f(s, a')\}} \right) \right| = \tau_{k+1}^{-1} \cdot \pi(a \mid s) \cdot (1 - \pi(a \mid s)) \leq \tau_{k+1}^{-1}/4.$$

Therefore, we obtain

$$
(\pi_{\theta_{k+1}}(a \,|\, s) - \widehat{\pi}_{k+1}(a \,|\, s))^2
$$
$$
\leq \tau_{k+1}^{-2}/16 \cdot \big(f_{\theta_{k+1}}(s,a) - \tau_{k+1} \cdot (\beta_k^{-1} Q_{\omega_k}(s,a) + \tau_k^{-1} f_{\theta_k}(s,a))\big)^2. \tag{B.1}
$$

Taking expectation $\mathbb{E}_{\widetilde{\sigma}_k}[\,\cdot\,]$ on the both sides of (B.1), we finally obtain

$$
\mathbb{E}_{\widetilde{\sigma}_k}[(\pi_{\theta_{k+1}}(a \,|\, s) - \widehat{\pi}_{k+1}(a \,|\, s))^2]
$$
$$
\leq \tau_{k+1}^{-2}/16 \cdot \mathbb{E}_{\widetilde{\sigma}_k}\big[\big(f_{\theta_{k+1}}(s,a) - \tau_{k+1} \cdot (\beta_k^{-1} Q_{\omega_0}(s,a) + \tau_k^{-1} f_{\theta_k}(s,a))\big)^2\big] \leq \tau_{k+1}^{-2}\epsilon_{k+1}/16,
$$

which concludes the proof of Lemma B.1. □

Lemma B.1 ensures that if the policy improvement error $\epsilon_{k+1}$ is small, then the corresponding improved policy $\pi_{\theta_{k+1}}$ is close to the ideal improved policy $\widehat{\pi}_{k+1}$, which justifies solving the subproblem in (3.5) for policy improvement.

# C    Proof of Proposition 3.1

*Proof.* The subproblem of policy improvement for solving $\widehat{\pi}_{k+1}$ takes the form

$$
\max_{\pi} \; \mathbb{E}_{\nu_k}\big[\langle \pi(\cdot \,|\, s), Q_{\omega_k}(s, \cdot)\rangle - \beta_k \cdot \mathrm{KL}(\pi(\cdot \,|\, s) \,\|\, \pi_{\theta_k}(\cdot \,|\, s))\big]
$$
$$
\text{subject to } \sum_{a \in \mathcal{A}} \pi(a \,|\, s) = 1, \text{ for any } s \in \mathcal{S}.
$$

The Lagrangian of the above maximization problem takes the form

$$
\int_{s \in \mathcal{S}} \big[\langle \pi(\cdot \,|\, s), Q_{\omega_k}(s, \cdot)\rangle - \beta_k \cdot \mathrm{KL}(\pi(\cdot \,|\, s) \,\|\, \pi_{\theta_k}(\cdot \,|\, s))\big] \nu_k(\mathrm{d}s) + \int_{s \in \mathcal{S}} \bigg(\sum_{a \in \mathcal{A}} \pi(a \,|\, s) - 1\bigg)\lambda(\mathrm{d}s).
$$

Plugging in $\pi_{\theta_k}(s,a) = \exp\{\tau_k^{-1} f_{\theta_k}(s,a)\}/\sum_{a' \in \mathcal{A}} \exp\{\tau_k^{-1} f_{\theta_k}(s,a')\}$, we obtain the optimality condition

$$
Q_{\omega_k}(s,a) + \beta_k \tau_k^{-1} f_{\theta_k}(s,a) - \beta_k \cdot \bigg[\log\bigg(\sum_{a' \in \mathcal{A}} \exp\{\tau_k^{-1} f_{\theta_k}(s,a')\}\bigg) + \log \pi(a \,|s) + 1\bigg] + \frac{\lambda(s)}{\nu_k(s)} = 0,
$$

for any $a \in \mathcal{A}$ and $s \in \mathcal{S}$. Note that $\log(\sum_{a' \in \mathcal{A}} \exp\{\tau_k^{-1} f_{\theta_k}(s,a')\})$ is determined by the state $s$ only. Hence, we have $\widehat{\pi}_{k+1}(a \,|\, s) \propto \exp\{\beta_k^{-1} Q_{\omega_k}(s,a) + \tau_k^{-1} f_{\theta_k}(s,a)\}$ for any $a \in \mathcal{A}$ and $s \in \mathcal{S}$, which concludes the proof of Proposition 3.1. □

# D    Proofs for Section 4.1

The proofs in this section generalizes those of [7, 5] under a unified framework, which accounts for both SGD, and TD, which uses stochastic semi-gradient. In particular, we develop a unified global convergence analysis of a meta-algorithm with the following update,

$$
\alpha(t + 1/2) \leftarrow \alpha(t) - \eta \cdot (u_{\alpha(t)}(s,a) - v(s,a) - \mu \cdot u_{\alpha(t)}(s',a')) \cdot \nabla_\alpha u_{\alpha(t)}(s,a), \tag{D.1}
$$
$$
\alpha(t + 1) \leftarrow \Pi_{\mathcal{B}^0(R_u)}(\alpha(1 + 1/2)) = \underset{\alpha \in \mathcal{B}^0(R_u)}{\mathrm{argmin}} \|\alpha - \alpha(t + 1/2)\|_2, \tag{D.2}
$$

where $\mu \in [0, 1)$ is a constant, $(s,a,s',a')$ is sampled from a stationary distribution $\rho$, and $u_\alpha$ is parametrized by the two-layer neural network $\mathrm{NN}(\alpha; m)$ defined in (3.1). The random initialization of $u_\alpha$ is given in (3.2). We denote by $\mathbb{E}_{\mathrm{init}}[\,\cdot\,]$ the expectation over such random initialization and $\mathbb{E}_\rho[\,\cdot\,]$ the expectation over $(s,a)$ conditional on the random initialization.

Such a meta-algorithm recovers SGD for policy improvement in (3.5) when we set $\rho = \widetilde{\sigma}_k$, $u_\alpha = f_\theta$, $v = \tau_{k+1} \cdot (\beta_k^{-1} Q_{\omega_k} + \tau_k^{-1} f_{\theta_k})$, $\mu = 0$, and $R_u = R_f$, and recovers TD for policy evaluation in (3.8) when we set $\rho = \sigma_k$, $u_\alpha = Q_\omega$, $v = (1 - \gamma) \cdot r$, $\mu = \gamma$, and $R_u = R_Q$.

To unify our analysis for SGD and TD, we assume that $v$ in (D.1) satisfies

$$
\mathbb{E}_\rho[(v(s,a))^2] \leq \overline{v}_1 \cdot \mathbb{E}_\rho[(u_{\alpha(0)}(s,a))^2] + \overline{v}_2 \cdot R_u^2 + \overline{v}_3
$$

for constants $\overline{v}_1, \overline{v}_2, \overline{v}_3 \geq 0$. Also, without loss of generality, we assume that $\|(s,a)\|_2 \leq 1$ for any $s \in \mathcal{S}$ and $a \in \mathcal{A}$. In Section D.2, we set $\overline{v}_1 = 4$, $\overline{v}_2 = 4$, and $\overline{v}_3 = 0$ for SGD, and $\overline{v}_1 = 0$, $\overline{v}_2 = 0$, and $\overline{v}_3 = R_{\max}$ for TD, respectively.

For notational simplicity, we define the residual $\delta_\alpha(s, a, s', a') = u_\alpha(s, a) - v(s, a) - \mu \cdot u_\alpha(s', a')$. We denote by

$$g_{\alpha(t)}(s, a, s', a') = \delta_{\alpha(t)}(s, a, s', a') \cdot \nabla_\alpha u_{\alpha(t)}(s, a), \quad \bar{g}_{\alpha(t)} = \mathbb{E}_\rho[g_t(s, a, s', a')] \qquad \text{(D.3)}$$

the stochastic update vector at the $t$-th iteration and its population mean, respectively. For SGD, $g_{\alpha(t)}(s, a, s', a')$ corresponds to the stochastic gradient, while for TD, $g_{\alpha(t)}(s, a, s', a')$ corresponds to the stochastic semigradient.

Note that the gradient of $u_\alpha(s, a)$ with respect to $\alpha$ takes the form

$$\nabla_\alpha u_\alpha(s, a) = 1/\sqrt{m} \cdot \left(b_1 \cdot \mathbb{1}\{[\alpha]_1^\top(s, a) > 0\} \cdot (s, a)^\top, \dots, b_m \cdot \mathbb{1}\{[\alpha]_m^\top(s, a) > 0\} \cdot (s, a)^\top\right)^\top \in \mathbb{R}^{md}$$

almost everywhere, which yields

$$\|\nabla_\alpha u_\alpha(s, a)\|_2^2 = \frac{1}{m} \sum_{i=1}^m \mathbb{1}\{[\alpha]_i^\top(s, a) > 0\} \cdot \|(s, a)\|_2^2 \leq 1.$$

Therefore, $u_\alpha(s, a)$ is 1-Lipschitz continuous with respect to $\alpha$.

In the following, we first show in Section D.1 that the overparametrization of $u_\alpha$ ensures that it behaves similarly as its local linearization at the random initialization $\alpha(0)$ defined in (3.2). Then in Section D.2, we establish the global convergence of the meta-algorithm defined in (D.1) and (D.2), which implies the global convergence of SGD and TD.

## D.1 Local Linearization

In this section, we first define a local linearization of the two-layer neural network $u_\alpha$ at its random initialization and then characterize the error induced by local linearization. We define

$$u_\alpha^0(s, a) = \frac{1}{\sqrt{m}} \sum_{i=1}^m b_i \cdot \mathbb{1}\{[\alpha(0)]_i^\top(s, a) > 0\} \cdot [\alpha]_i^\top(s, a). \qquad \text{(D.4)}$$

The linearity of $u_\alpha^0$ with respect to $\alpha$ yields

$$\langle \nabla_\alpha u_\alpha^0(s, a), \alpha \rangle = u_\alpha^0(s, a). \qquad \text{(D.5)}$$

The following lemma characterizes how far $u_{\alpha(t)}^0$ deviates from $u_{\alpha(t)}$ for $\alpha(t) \in \mathcal{B}^0(R_u)$.

**Lemma D.1.** For any $\alpha' \in \mathcal{B}^0(R_u)$, we have

$$\mathbb{E}_{\text{init}, \rho}[(u_{\alpha'}(s, a) - u_{\alpha'}^0(s, a))^2] = O(R_u^3 m^{-1/2}).$$

*Proof.* By the definition of $u_\alpha$ in (3.1), we have

$$|u_{\alpha'}(s, a) - u_{\alpha'}^0(s, a)| \qquad \text{(D.6)}$$

$$\leq \frac{1}{\sqrt{m}} \left| \sum_{i=1}^m b_i \cdot \left(\mathbb{1}\{[\alpha(0)]_i^\top(s, a) > 0\} - \mathbb{1}\{[\alpha(0)]_i^\top(s, a) > 0\}\right) \cdot \left(|[\alpha(0)]_i^\top(s, a)| + \|[\alpha']_i - [\alpha(0)]_i\|_2\right) \right|$$

$$\leq \frac{1}{\sqrt{m}} \sum_{i=1}^m \mathbb{1}\{[\alpha(0)]_i^\top(s, a) \leq \|[\alpha']_i - [\alpha(0)]_i\|_2\} \cdot \left(|[\alpha(0)]_i^\top(s, a)| + \|[\alpha']_i - [\alpha(0)]_i\|_2\right),$$

where the second inequality follows from $|b_i| = 1$ and the fact that

$$\mathbb{1}\{[\alpha(t)]_i^\top(s, a) > 0\} \neq \mathbb{1}\{[\alpha(0)]_i^\top(s, a) > 0\}$$

implies

$$|[\alpha(0)]_i^\top(s, a)| \leq |[\alpha(t)]_i^\top(s, a) - [\alpha(0)]_i^\top(s, a)| \leq \|[\alpha(0)]_i - [\alpha(t)]_i\|_2.$$

Next, applying the inequality $\mathbb{1}\{|z| \leq y\}|z| \leq \mathbb{1}\{|z| \leq y\}y$ to the right-hand side of (D.6), we obtain

$$|u_{\alpha'}(s, a) - u_{\alpha'}^0(s, a)|$$

$$\leq \frac{2}{\sqrt{m}} \sum_{i=1}^m \mathbb{1}\{[\alpha(0)]_i^\top(s, a) \leq \|[\alpha']_i - [\alpha(0)]_i\|_2\} \cdot \|[\alpha']_i - [\alpha(0)]_i\|_2. \qquad \text{(D.7)}$$

Further applying the Cauchy-Schwarz inequality to (D.7) and invoking the upper bound $\|\alpha' - \alpha(0)\|_2 \leq R_u$, we obtain

$$|u_{\alpha'}(s, a) - u_{\alpha'}^0(s, a)|^2 \leq \frac{4R_u^2}{m} \sum_{i=1}^m \mathbb{1}\{[\alpha(0)]_i^\top(s, a) \leq \|[\alpha']_i - [\alpha(0)]_i\|_2\}. \qquad \text{(D.8)}$$

Taking expectation on the both sides and invoking Assumption 4.4, we obtain

$$\mathbb{E}_{\text{init},\rho}[(u_{\alpha'}(s,a) - u^0_{\alpha'}(s,a))^2] \leq \frac{4cR_u^2}{m} \cdot \mathbb{E}_{\text{init}}\left[\sum_{i=1}^m \|[\alpha']_i - [\alpha(0)]_i\|_2 / \|[\alpha(0)]_i\|_2\right]. \qquad \text{(D.9)}$$

By the Cauchy-Schwartz inequality, we have

$$\mathbb{E}_{\text{init}}\left[\sum_{i=1}^m \|[\alpha']_i - [\alpha(0)]_i\|_2 / \|[\alpha(0)]_i\|_2\right] \leq \mathbb{E}_{\text{init}}\left[\sum_{i=1}^m \|[\alpha']_i - [\alpha(0)]_i\|_2^2\right]^{1/2} \cdot \mathbb{E}_{\text{init}}\left[\sum_{i=1}^m \|[\alpha(0)]_i\|_2^{-2}\right]^{1/2}$$

$$\leq R_u \cdot \mathbb{E}_{\text{init}}\left[\sum_{i=1}^m \|[\alpha(0)]_i\|_2^{-2}\right]^{1/2},$$

where the second inequality follows from $\sum_{i=1}^m \|[\alpha']_i - [\alpha(0)]_i\|_2^2 = \|\alpha' - \alpha(0)\|_2^2 \leq R_u^2$. Therefore, we have that the right-hand side of (D.9) is $O(R_u^3 m^{-1/2})$. Thus, we obtain

$$\mathbb{E}_{\text{init},\rho}[(u_{\alpha'}(s,a) - u^0_{\alpha'}(s,a))^2] = O(R_u^3 m^{-1/2}),$$

which concludes the proof of Lemma D.1. $\qquad\qquad\qquad\qquad\qquad\qquad\qquad\qquad\qquad\qquad\square$

Corresponding to $u^0_\alpha$ defined in (D.4), let $\delta^0_\alpha(s,a,s',a') = u^0_\alpha(s,a) - v(s,a) - \mu \cdot u^0_\alpha(s',a')$. We define the local linearization of $\bar{g}_{\alpha(t)}$, which is defined in (D.3), as

$$\bar{g}^0_{\alpha(t)} = \mathbb{E}_\rho[\delta^0_{\alpha(t)}(s,a,s',a') \cdot \nabla_\alpha u^0_{\alpha(t)}(s,a)]. \qquad \text{(D.10)}$$

The following lemma characterizes the difference between $\bar{g}^0_{\alpha(t)}$ and $\bar{g}_{\alpha(t)}$.

**Lemma D.2.** For any $t \in [T]$, we have

$$\mathbb{E}_{\text{init}}[\|\bar{g}_{\alpha(t)} - \bar{g}^0_{\alpha(t)}\|_2^2] = O(R_u^3 m^{-1/2}).$$

*Proof.* By the definition of $\bar{g}^0_{\alpha(t)}$ and $\bar{g}_{\alpha(t)}$ in (D.10) and (D.3), we have

$$\|\bar{g}_{\alpha(t)} - \bar{g}^0_{\alpha(t)}\|_2^2 = \|\mathbb{E}_\rho[\delta_{\alpha(t)}(s,a,s',a') \cdot \nabla_\alpha u_{\alpha(t)}(s,a) - \delta^0_{\alpha(t)}(s,a,s',a') \cdot \nabla_\alpha u^0_{\alpha(t)}(s,a)]\|_2^2$$

$$\leq 2\underbrace{\mathbb{E}_\rho\big[|\delta_{\alpha(t)}(s,a,s',a') - \delta^0_{\alpha(t)}(s,a,s',a')|^2 \cdot \|\nabla_\alpha u_{\alpha(t)}(s,a)\|_2^2\big]}_{\text{(i)}} \qquad \text{(D.11)}$$

$$+ 2\underbrace{\mathbb{E}_\rho\big[|\delta^0_{\alpha(t)}(s,a,s',a')| \cdot \|\nabla_\alpha u_{\alpha(t)}(s,a) - \nabla_\alpha u^0_{\alpha(t)}(s,a)\|_2\big]^2}_{\text{(ii)}}.$$

**Upper Bounding (i):** We have $\|\nabla_\alpha u_{\alpha(t)}(s,a)\|_2 \leq 1$ as $\|(s,a)\|_2 \leq 1$. Note that the difference between $\delta_{\alpha(t)}$ and $\delta^0_{\alpha(t)}$ takes the form

$$\delta_{\alpha(t)}(s,a,s',a') - \delta^0_{\alpha(t)}(s,a,s',a') = (u_{\alpha(t)}(s,a) - u^0_{\alpha(t)}(s,a)) - \mu \cdot (u_{\alpha(t)}(s',a') - u^0_{\alpha(t)}(s',a')).$$

Taking expectation on the both sides, we obtain

$$\mathbb{E}_{\text{init},\rho}[|\delta_{\alpha(t)}(s,a,s',a') - \delta^0_{\alpha(t)}(s,a,s',a')|^2]$$

$$\leq 2\mathbb{E}_{\text{init},\rho}[(u_{\alpha(t)}(s,a) - u^0_{\alpha(t)}(s,a))^2] + 2\mu^2 \cdot \mathbb{E}_{\text{init},\rho}[(u_{\alpha(t)}(s',a') - u^0_{\alpha(t)}(s',a'))^2]$$

$$= 4\mathbb{E}_{\text{init},\rho}[(u_{\alpha(t)}(s,a) - u^0_{\alpha(t)}(s,a))^2],$$

where the equality follows from $|\mu| \leq 1$ and the fact that $(s,a)$ and $(s',a')$ have the same marginal distribution. Thus, by Lemma D.1, we have that (i) in (D.11) is $O(R_u^3 m^{-1/2})$.

**Upper Bounding (ii):** First, by the Hölder's inequality, we have

$$\mathbb{E}_\rho\big[|\delta^0_{\alpha(t)}(s,a,s',a')| \cdot \|\nabla_\alpha u_{\alpha(t)}(s,a) - \nabla_\alpha u^0_{\alpha(t)}(s,a)\|_2\big]^2$$

$$\leq \mathbb{E}_\rho[|\delta^0_{\alpha(t)}(s,a,s',a')|^2] \cdot \mathbb{E}_\rho[\|\nabla_\alpha u_{\alpha(t)}(s,a) - \nabla_\alpha u^0_{\alpha(t)}(s,a)\|_2^2].$$

We use $|u^0_{\alpha(t)}(s,a) - u^0_{\alpha(0)}(s,a)| \leq \|\alpha(t) - \alpha(0)\|_2 \leq R_u$ to obtain

$$|\delta^0_{\alpha(t)}(s,a,s',a')|^2 = (u^0_{\alpha(t)}(s,a) - v(s,a) - \mu \cdot u^0_{\alpha(t)}(s',a'))^2$$

$$\leq 3\big((u^0_{\alpha(t)}(s,a))^2 + (v(s,a))^2 + \mu^2 \cdot (u^0_{\alpha(t)}(s',a'))^2\big)$$

$$\leq 3(u^0_{\alpha(0)}(s,a))^2 + 3(u^0_{\alpha(0)}(s',a'))^2 + 6R_u^2 + 3(v(s,a))^2. \qquad \text{(D.12)}$$

Next we characterize $\|\nabla_\alpha u_{\alpha(t)}(s,a) - \nabla_\alpha u^0_{\alpha(t)}(s,a)\|_2$ in (ii). Recall that

$$\nabla_\alpha u_\alpha(s,a) = 1/\sqrt{m} \cdot \left(b_1 \cdot \mathbb{1}\{[\alpha]_1^\top(s,a) > 0\} \cdot (s,a)^\top, \ldots, b_m \cdot \mathbb{1}\{[\alpha]_m^\top(s,a) > 0\} \cdot (s,a)^\top\right)^\top,$$

and

$$\nabla_\alpha u^0_\alpha(s,a) = 1/\sqrt{m} \cdot \left(b_1 \cdot \mathbb{1}\{[\alpha(0)]_1^\top(s,a) > 0\} \cdot (s,a)^\top, \ldots, b_m \cdot \mathbb{1}\{[\alpha(0)]_m^\top(s,a) > 0\} \cdot (s,a)^\top\right)^\top.$$

We have

$$\|\nabla_\alpha u_{\alpha(t)}(s,a) - \nabla_\alpha u^0_{\alpha(t)}(s,a)\|_2^2 = \frac{1}{m}\sum_{i=1}^m \left(\mathbb{1}\{[\alpha(t)]_i^\top(s,a) > 0\} - \mathbb{1}\{[\alpha(0)]_i^\top(s,a) > 0\}\right)^2 \cdot \|(s,a)\|_2^2$$

$$\leq \frac{1}{m}\sum_{i=1}^m \mathbb{1}\{[\alpha(0)]_i^\top(s,a) \leq \|[\alpha(t)]_i - [\alpha(0)]_i\|_2\}, \quad \text{(D.13)}$$

where the inequality follows from the same arguments used to derive (D.6). Plugging (D.12) and (D.13) into (ii) and recalling that

$$\mathbb{E}_\rho[(v(s,a))^2] \leq \overline{v}_1 \cdot \mathbb{E}_\rho[(u_{\alpha(0)}(s,a))^2] + \overline{v}_2 \cdot R_u^2 + \overline{v}_3,$$

we find that it remains to upper bound the following two terms

$$\mathbb{E}_{\text{init},\rho}\left[\frac{1}{m}\sum_{i=1}^m \mathbb{1}\{[\alpha(0)]_i^\top(s,a) \leq \|[\alpha(t)]_i - [\alpha(0)]_i\|_2\}\right], \quad \text{(D.14)}$$

and

$$\mathbb{E}_{\text{init}}\left[\mathbb{E}_\rho[(u^0_{\alpha(0)}(s,a))^2] \cdot \mathbb{E}_\rho\left[\frac{1}{m}\sum_{i=1}^m \mathbb{1}\{[\alpha(0)]_i^\top(s,a) \leq \|[\alpha(t)]_i - [\alpha(0)]_i\|_2\}\right]\right]. \quad \text{(D.15)}$$

We already show in the proof of Lemma D.1 that (D.14) is $O(R_u m^{-1/2})$. We characterize (D.15) in the following. For the random initialization of $u_\alpha(s,a)$ in (3.2), we have

$$\mathbb{E}_\rho[(u^0_{\alpha(0)}(s,a))^2] = \frac{1}{m} \cdot \mathbb{E}_\rho\left[\sum_{i=1}^m \sigma([\alpha(0)]_i^\top(s,a))^2 + \sum_{1 \leq i \neq j \leq m} b_i b_j \cdot \sigma([\alpha(0)]_i^\top(s,a)) \cdot \sigma([\alpha(0)]_j^\top(s,a))\right],$$

plugging which into (D.15) gives

$$\mathbb{E}_{\text{init}}\left[\mathbb{E}_\rho[(u^0_{\alpha(0)}(s,a))^2] \cdot \mathbb{E}_\rho\left[\frac{1}{m}\sum_{i=1}^m \mathbb{1}\{[\alpha(0)]_i^\top(s,a) \leq \|\alpha(t) - \alpha(0)\|_2\}\right]\right]$$

$$\leq \mathbb{E}_{\text{init}}\left[\frac{1}{m} \cdot \mathbb{E}_\rho\left[\sum_{i=1}^m \sigma([\alpha(0)]_i^\top(s,a))^2 + \sum_{1 \leq i \neq j \leq m} b_i b_j \cdot \sigma([\alpha(0)]_i^\top(s,a)) \cdot \sigma([\alpha(0)]_j^\top(s,a))\right]\right.$$

$$\left. \cdot \frac{c}{m} \cdot \left(\sum_{i=1}^m \|[\alpha(t)]_i - [\alpha(0)]_i\|_2^2\right)^{1/2} \cdot \left(\sum_{i=1}^m \frac{1}{\|[\alpha(0)]_i\|_2^2}\right)^{1/2}\right],$$

where we use the same arguments applied to (D.8) in the proof of Lemma D.1. Note that $b_i, b_j$ are independent of $\alpha(0)$, $\mathbb{E}_{\text{init}}[b_i b_j] = 0$, and $\sum_{i=1}^m \|[\alpha(t)]_i - [\alpha(0)]_i\|_2^2 = \|\alpha(t) - \alpha(0)\|_2^2 \leq R_u^2$. We further obtain

$$\mathbb{E}_{\text{init}}\left[\mathbb{E}_\rho[(u^0_{\alpha(0)}(s,a))^2] \cdot \mathbb{E}_\rho\left[\frac{1}{m}\sum_{i=1}^m \mathbb{1}\{[\alpha(0)]_i^\top(s,a) \leq \|[\alpha(t)]_i - [\alpha(0)]_i\|_2\}\right]\right]$$

$$\leq \frac{cR_u}{m^2} \cdot \mathbb{E}_{\text{init}}\left[\mathbb{E}_\rho\left[\sum_{i=1}^m \sigma([\alpha(0)]_i^\top(s,a))^2\right] \cdot \left(\sum_{i=1}^m \frac{1}{\|[\alpha(0)]_i\|_2^2}\right)^{1/2}\right]$$

$$\leq \frac{cR_u}{m^2} \cdot \mathbb{E}_{\text{init}}\left[\left(\sum_{i=1}^m \|[\alpha(0)]_i\|_2^2\right) \cdot \left(\sum_{i=1}^m \frac{1}{\|[\alpha(0)]_i\|_2^2}\right)^{1/2}\right].$$

Finally, by the Cauchy-Schwarz inequality, we have

$$\mathbb{E}_{\text{init}}\left[\left(\sum_{i=1}^m \|[\alpha(0)]_i\|_2^2\right) \cdot \left(\sum_{i=1}^m \frac{1}{\|[\alpha(0)]_i\|_2^2}\right)^{1/2}\right]$$

$$\leq \mathbb{E}_{\text{init}}\left[\left(\sum_{i=1}^m \|[\alpha(0)]_i\|_2^2\right)^2\right]^{1/2} \cdot \mathbb{E}_{\text{init}}\left[\sum_{i=1}^m \frac{1}{\|[\alpha(0)]_i\|_2^2}\right]^{1/2},$$

whose right-hand side is $O(m^{3/2})$. Thus, we obtain that (D.15) is $O(R_u m^{-1/2})$ and (ii) in (D.11) is $O(R_u^3 m^{-1/2})$, which concludes the proof of Lemma D.2. $\qquad\square$

## D.2 Global Convergence

In this section, we establish the global convergence of the meta-algorithm defined in (D.1) and (D.2). We first present the following lemma for characterizing the variance of the stochastic update vector $g_{\alpha(t)}(s, a, s', a')$ defined in (D.3), which later allows us to focus on tracking its mean in the global convergence analysis.

**Lemma D.3** (Variance of the Stochastic Update Vector)**.** There exists a constant $\xi_g^2 = O(R_u^2)$ independent of $t$, such that for any $t \leq T$, it holds that
$$\mathbb{E}_{\text{init},\rho}[\|g_{\alpha(t)}(s, a, s', a') - \bar{g}_{\alpha(t)}\|_2^2] \leq \xi_g^2.$$

*Proof.* Since we have
$$\mathbb{E}_{\text{init},\rho}[\|g_{\alpha(t)}(s, a, s', a') - \bar{g}_{\alpha(t)}\|_2^2] = \mathbb{E}_{\text{init}}\left[\mathbb{E}_\rho[\|g_{\alpha(t)}(s, a, s', a') - \bar{g}_{\alpha(t)}\|_2^2]\right]$$
$$\leq \mathbb{E}_{\text{init}}\left[\mathbb{E}_\rho[\|g_{\alpha(t)}(s, a, s', a')\|_2^2]\right] = \mathbb{E}_{\text{init},\rho}[\|g_{\alpha(t)}(s, a, s', a')\|_2^2],$$
it suffices to prove that $\mathbb{E}[\|g_{\alpha(t)}(s, a, s', a')\|_2^2] = O(R_u^2)$. By the definition of $\mathbb{E}_\rho[\|g_{\alpha(t)}(s, a, s', a')\|_2^2]$ in (D.3), using $\|\nabla_{\alpha(t)} u_{\alpha(t)}(s, a)\|_2^2 \leq 1$, we obtain
$$\mathbb{E}_\rho[\|g_{\alpha(t)}(s, a, s', a')\|_2^2] = \mathbb{E}_\rho[\|\delta_{\alpha(t)}(s, a, s', s') \cdot \nabla_\alpha u_{\alpha(t)}(s, a)\|_2^2]$$
$$\leq \mathbb{E}_\rho[|\delta_{\alpha(t)}(s, a, s', s')|^2]. \tag{D.16}$$
Then, by similar arguments used in the derivation of (D.12), we obtain
$$\mathbb{E}_{\text{init},\rho}[|\delta_{\alpha(t)}(s, a, s', s')|^2] \leq 6\mathbb{E}_{\text{init},\rho}[(u_{\alpha(0)}(s, a))^2] + 6R_u^2 + 3\mathbb{E}_{\text{init},\rho}[(v(s, a))^2]$$
$$\leq (6 + 3\bar{v}_1) \cdot \mathbb{E}_{\text{init},\rho}[(u_{\alpha(0)}(s, a))^2] + (6 + \bar{v}_2)R_u^2 + 3\bar{v}_3^2. \tag{D.17}$$
Note that by $\|(s, a)\|_2 \leq 1$, we have
$$\mathbb{E}_{\text{init},\rho}[(u_{\alpha(0)}(s, a))^2] = \mathbb{E}_{z \sim \mathcal{N}(0, I_d/d), \rho}[\sigma(z^\top (s, a))^2] \leq \mathbb{E}_{z \sim \mathcal{N}(0, I_d/d)}[\|z\|_2^2] = 1,$$
which together with (D.16) and (D.17) implies $\mathbb{E}_{\text{init},\rho}[\|g_{\alpha(t)}(s, a, s', a')\|_2^2] = O(R_u^2)$. Thus, we complete the proof of Lemma D.3. □

Before presenting the global convergence result of the meta-algorithm defined in (D.1), we first define $u_{\alpha^*}^0$, which later become the exact learning target of the meta-algorithm defined in (D.1) and (D.2). In specific, we define the approximate stationary point as $\alpha^* \in \mathcal{B}^0(R_u)$ such that
$$\alpha^* = \Pi_{\mathcal{B}^0(R_u)}(\alpha^* - \eta \cdot \bar{g}_{\alpha^*}^0), \tag{D.18}$$
which is equivalent to the condition
$$\langle \bar{g}_{\alpha^*}^0, \alpha - \alpha^* \rangle \geq 0, \quad \text{for any } \alpha \in \mathcal{B}^0(R_u). \tag{D.19}$$
Then we establish the uniqueness and existence of $u_{\alpha^*}^0$ with $\alpha^*$ defined in D.18. We first define the operator
$$\mathcal{T}u(s, a) = \mathbb{E}[v(s, a) + \mu \cdot u(s', a') \mid s' \sim \mathcal{P}(\cdot \mid s, a), a \sim \pi(\cdot \mid s')]. \tag{D.20}$$
Then using the definition of $\mathcal{T}$ in (D.20) and plugging the definition of $\bar{g}_{\alpha^*}^0$ in (D.4) into (D.19), we obtain
$$\langle u_{\alpha^*}^0 - \mathcal{T}u_{\alpha^*}^0, u_\alpha^0 - u_{\alpha^*}^0 \rangle_\rho \geq 0, \quad \text{for any } u_\alpha^0 \in \mathcal{F}_{B,m},$$
which is equivalent to $u_{\alpha^*}^0 = \Pi_{\mathcal{F}_{B,m}} \mathcal{T}u_{\alpha^*}^0$. Here the projection $\Pi_{\mathcal{F}_{B,m}}$ is defined with respect to the $\ell_2$-distance under measure $\rho$. Finally, as we have the following contraction inequality
$$\mathbb{E}_\rho[(\Pi_{\mathcal{F}_{B,m}} \mathcal{T}u_\alpha^0(s, a) - \Pi_{\mathcal{F}_{B,m}} \mathcal{T}u_{\alpha'}^0(s, a))^2]$$
$$\leq \mathbb{E}_\rho[(\mathcal{T}u_\alpha^0(s, a) - \mathcal{T}u_{\alpha'}^0(s, a))^2]$$
$$= \mu^2 \cdot \mathbb{E}_\rho\left[\left(\mathbb{E}[u_\alpha^0(s', a') \mid s' \sim \mathcal{P}(\cdot \mid s, a), a' \sim \pi(\cdot \mid s')] - \mathbb{E}[u_{\alpha'}^0(s', a') \mid s' \sim \mathcal{P}(\cdot \mid s, a), a' \sim \pi(\cdot \mid s')]\right)^2\right]$$
$$\leq \mu^2 \cdot \mathbb{E}_\rho[(u_\alpha^0(s, a) - u_{\alpha'}^0(s, a))^2],$$
we know that such fixed-point solution $u_{\alpha^*}^0$ uniquely exists.

Now, with a well-defined learning target $u_{\alpha^*}^0$, we are ready to prove the the global convergence of the meta-algorithm defined in (D.1) and (D.2) with two-layer neural network approximation.

**Theorem D.4.** Suppose that we run $T \geq 64/(1-\mu)^2$ iterations of the meta-algorithm defined in (D.1) and (D.2). Setting the stepsize $\eta = T^{-1/2}$, we have
$$\mathbb{E}_{\text{init},\rho}[(u_{\bar{\alpha}}(s, a) - u_{\alpha^*}^0(s, a))^2] = O(R_u^2 T^{-1/2} + R_u^{5/2} m^{-1/4} + R_u^3 m^{-1/2}),$$
where $\bar{\alpha} = 1/T \cdot \sum_{t=0}^{T-1} \alpha(t)$ and $\alpha^*$ is the approximate stationary point defined in (D.18).

*Proof.* The proof of the theorem consists of two parts. We first analyze the progress of each step. Then based on such one-step analysis, we establish the error bound of the approximation via two-layer neural network $u_\alpha$.

**One-Step Analysis:** For any $t < T$, using the stationarity condition in (D.18) and the convexity of $\mathcal{B}^0(R_u)$, we obtain

$$\mathbb{E}_\rho[\|\alpha(t+1) - \alpha^*\|_2^2 \,|\, \alpha(t)] \tag{D.21}$$

$$= \mathbb{E}_\rho\big[\big\|\Pi_{\mathcal{B}^0(R_u)}(\alpha(t) - \eta \cdot g_{\alpha(t)}(s, a, s', a')) - \Pi_{\mathcal{B}^0(R_u)}(\alpha^* - \eta \bar{g}_{\alpha^*}^0)\big\|_2^2 \,\big|\, \alpha(t)\big]$$

$$\leq \mathbb{E}_\rho\big[\big\|(\alpha(t) - \alpha^*) - \eta \cdot (g_{\alpha(t)}(s, a, s', a') - \bar{g}_{\alpha^*}^0)\big\|_2^2 \,\big|\, \alpha(t)\big]$$

$$= \|\alpha(t) - \alpha^*\|_2^2 - 2\eta \cdot \langle \bar{g}_{\alpha(t)} - \bar{g}_{\alpha^*}^0, \alpha(t) - \alpha^* \rangle + \eta^2 \cdot \mathbb{E}_\rho[\|g_{\alpha(t)}(s, a, s', a') - \bar{g}_{\alpha^*}^0\|_2^2 \,|\, \alpha(t)].$$

In the following, we upper bound the last two terms in (D.21). First, to upper bound $\mathbb{E}_\rho[\|g_{\alpha(t)}(s, a, s', a') - \bar{g}_{\alpha^*}^0\|_2^2 \,|\, \alpha(t)]$, by the Cauchy-Schwarz inequality we have

$$\mathbb{E}_\rho[\|g_{\alpha(t)}(s, a, s', a') - \bar{g}_{\alpha^*}^0\|_2^2 \,|\, \alpha(t)]$$

$$\leq 2\mathbb{E}_\rho[\|g_{\alpha(t)}(s, a, s', a') - \bar{g}_{\alpha(t)}\|_2^2 \,|\, \alpha(t)] + 2\|\bar{g}_{\alpha(t)} - \bar{g}_{\alpha^*}^0\|_2^2$$

$$\leq 2\mathbb{E}_\rho[\|g_{\alpha(t)}(s, a, s', a') - \bar{g}_{\alpha(t)}\|_2^2 \,|\, \alpha(t)] + 4\|\bar{g}_{\alpha(t)} - \bar{g}_{\alpha(t)}^0\|_2^2 + 4\|\bar{g}_{\alpha(t)}^0 - \bar{g}_{\alpha^*}^0\|_2^2, \tag{D.22}$$

where the total expectation on the first two terms on the right-hand side are characterized in Lemmas D.3 and D.2, respectively. To characterize $\|\bar{g}_{\alpha(t)}^0 - \bar{g}_{\alpha^*}^0\|_2^2$, again using $\|(s, a)\|_2 \leq 1$, we have

$$\|\bar{g}_{\alpha(t)}^0 - \bar{g}_{\alpha^*}^0\|_2^2 = \mathbb{E}_\rho\big[(\delta_{\alpha(t)}(s, a, s', a') - \delta_{\alpha^*}(s, a, s', a'))^2 \cdot \|\nabla_\alpha u_{\alpha(t)}^0(s, a)\|_2^2\big]$$

$$\leq \mathbb{E}_\rho\big[\big((u_{\alpha(t)}^0(s, a) - u_{\alpha^*}^0(s, a)) - \mu \cdot (u_{\alpha(t)}^0(s', a') - u_{\alpha^*}^0(s', a'))\big)^2\big]. \tag{D.23}$$

For the right-hand side of (D.23), we use the Cauchy-Schwarz inequality on the interaction term and obtain

$$\mathbb{E}_\rho\big[(u_{\alpha(t)}^0(s', a') - u_{\alpha^*}^0(s', a')) \cdot (u_{\alpha(t)}^0(s, a) - u_{\alpha^*}^0(s, a))\big]$$

$$\leq \mathbb{E}_\rho[(u_{\alpha(t)}^0(s', a') - u_{\alpha^*}^0(s', a'))^2]^{1/2} \cdot \mathbb{E}_\rho[(u_{\alpha(t)}^0(s, a) - u_{\alpha^*}^0(s, a))^2]^{1/2}$$

$$= \mathbb{E}_\rho[(u_{\alpha(t)}^0(s, a) - u_{\alpha^*}^0(s, a))^2], \tag{D.24}$$

where in the last line we use the fact that $(s, a)$ and $(s', a')$ have the same marginal distribution. Thus, we obtain

$$\|\bar{g}_{\alpha(t)}^0 - \bar{g}_{\alpha^*}^0\|_2^2 \leq 4\mathbb{E}_\rho[(u_{\alpha(t)}^0(s, a) - u_{\alpha^*}^0(s, a))^2]. \tag{D.25}$$

Next, to upper bound $\langle \bar{g}_{\alpha(t)} - \bar{g}_{\alpha^*}^0, \alpha(t) - \alpha^* \rangle$, we use the Hölder's inequality to obtain

$$\langle \bar{g}_{\alpha(t)} - \bar{g}_{\alpha^*}^0, \alpha(t) - \alpha^* \rangle = \langle \bar{g}_{\alpha(t)} - \bar{g}_{\alpha(t)}^0, \alpha(t) - \alpha^* \rangle + \langle \bar{g}_{\alpha(t)}^0 - \bar{g}_{\alpha^*}^0, \alpha(t) - \alpha^* \rangle$$

$$\geq -\|\bar{g}_{\alpha(t)} - \bar{g}_{\alpha(t)}^0\|_2 \cdot \|\alpha(t) - \alpha^*\|_2 + \langle \bar{g}_{\alpha(t)}^0 - \bar{g}_{\alpha^*}^0, \alpha(t) - \alpha^* \rangle$$

$$\geq -R_u\|\bar{g}_{\alpha(t)} - \bar{g}_{\alpha(t)}^0\|_2 + \langle \bar{g}_{\alpha(t)}^0 - \bar{g}_{\alpha^*}^0, \alpha(t) - \alpha^* \rangle, \tag{D.26}$$

where the second inequality follows from $\|\alpha(t) - \alpha^*\|_2 \leq R_u$. For the term $\langle \bar{g}_{\alpha(t)}^0 - \bar{g}_{\alpha^*}^0, \alpha(t) - \alpha^* \rangle$ on the right-hand side of (D.26), we have

$$\langle \bar{g}_{\alpha(t)}^0 - \bar{g}_{\alpha^*}^0, \alpha(t) - \alpha^* \rangle$$

$$= \mathbb{E}_\rho\Big[\big((u_{\alpha(t)}^0(s, a) - u_{\alpha^*}^0(s, a)) - \mu \cdot (u_{\alpha(t)}^0(s', a') - u_{\alpha^*}^0(s', a'))\big) \cdot \langle \nabla_\alpha u_{\alpha(t)}^0(s, a), \alpha(t) - \alpha^* \rangle\Big]$$

$$= \mathbb{E}_\rho\Big[\big((u_{\alpha(t)}^0(s, a) - u_{\alpha^*}^0(s, a)) - \mu \cdot (u_{\alpha(t)}^0(s', a') - u_{\alpha^*}^0(s', a'))\big) \cdot (u_{\alpha(t)}^0(s, a) - u_{\alpha^*}^0(s, a))\Big]$$

$$\geq \mathbb{E}_\rho[(u_{\alpha(t)}^0(s, a) - u_{\alpha^*}^0(s, a))^2] - \mu \cdot \mathbb{E}_\rho[(u_{\alpha(t)}^0(s, a) - u_{\alpha^*}^0(s, a))]^2$$

$$\geq (1 - \mu) \cdot \mathbb{E}_\rho[(u_{\alpha(t)}^0(s, a) - u_{\alpha^*}^0(s, a))^2], \tag{D.27}$$

where the second equality and the first inequality follow from (D.5) and (D.24), respectively.

Therefore, combining (D.21) with (D.22), (E.4), (D.26), and (D.27), we obtain

$$\mathbb{E}_\rho[\|\alpha(t+1) - \alpha^*\|_2^2 \,|\, \alpha(t)]$$

$$\leq \|\alpha(t) - \alpha^*\|_2^2 - \big(2\eta(1 - \gamma) - 8\eta^2\big) \cdot \mathbb{E}_\rho[(u_{\alpha(t)}^0(s, a) - u_{\alpha^*}^0(s, a))^2 \,|\, \alpha(t)] \tag{D.28}$$

$$+ 2\eta^2\|\bar{g}_{\alpha(t)} - \bar{g}_{\alpha(t)}^0\|_2^2 + 2\eta R_u\|\bar{g}_{\alpha(t)} - \bar{g}_{\alpha(t)}^0\|_2 + \eta^2 \cdot \mathbb{E}_\rho[\|g_{\alpha(t)}(s, a, s', a') - \bar{g}_{\alpha(t)}\|_2^2 \,|\, \alpha(t)].$$

**Error Bound:** Rearranging (D.28), we obtain

$$\mathbb{E}_\rho[(u_{\alpha(t)}(s,a) - u_{\alpha^*}^0(s,a))^2 \mid \alpha(t)]$$

$$\leq \mathbb{E}_\rho\big[2(u_{\alpha(t)}(s,a) - u_{\alpha(t)}^0(s,a))^2 + 2(u_{\alpha(t)}^0(s,a) - u_{\alpha^*}^0(s,a))^2 \mid \alpha(t)\big]$$

$$\leq \big(\eta(1-\gamma) - 4\eta^2\big)^{-1} \cdot \big(\|\alpha(t) - \alpha^*\|_2^2 - \mathbb{E}_\rho[\|\alpha(t+1) - \alpha^*\|_2^2 \mid \alpha(t)] + \xi_\alpha^2\eta^2\big) \qquad \text{(D.29)}$$

$$+ O(R_u^{5/2}m^{-1/4} + R_u^3 m^{-1/2}).$$

Taking total expectation on both sides of (D.29) and telescoping for $t+1 \in [T]$, we further obtain

$$\mathbb{E}_{\text{init},\rho}[(u_{\overline{\alpha}}(s,a) - u_{\alpha^*}^0(s,a))^2] \leq \frac{1}{T}\sum_{t=0}^{T-1}\mathbb{E}_{\text{init},\rho}[(u_{\alpha(t)}(s,a) - u_{\alpha^*}(s,a))^2] \qquad \text{(D.30)}$$

$$\leq T^{-1} \cdot \big(\eta(1-\gamma) - 4\eta^2\big)^{-1} \cdot \big(\mathbb{E}_{\text{init}}[\|\alpha(0) - \alpha^*\|_2^2] + T\xi_\alpha^2\eta^2\big)$$

$$+ O(R_u^{5/2}m^{-1/4} + R_u^3 m^{-1/2}).$$

Let $T \geq 64/(1-\mu)^2$ and $\eta = T^{-1/2}$, it holds that $T^{-1/2} \cdot (\eta(1-\gamma) - 4\eta^2)^{-1} \leq 16(1-\gamma)^{-1/2}$ and $T\eta^2 \leq 1$, which together with (D.30) implies

$$\mathbb{E}_{\text{init},\rho}[(u_{\alpha(t)}(s,a) - u_{\alpha^*}^0(s,a))^2 \mid \alpha(t)]$$

$$\leq \frac{16}{(1-\mu)^2\sqrt{T}} \cdot \big(\mathbb{E}_{\text{init}}[\|\alpha(0) - \alpha^*\|_2^2] + \xi_\alpha^2\big) + O(R_u^{5/2}m^{-1/4} + R_u^3 m^{-1/2})$$

$$\leq \frac{16(R_\alpha^2 + \xi_\alpha^2)}{(1-\mu)^2\sqrt{T}} + O(R_u^{5/2}m^{-1/4} + R_u^3 m^{-1/2}) = O(R_u^2 T^{-1/2} + R_u^{5/2}m^{-1/4} + R_u^3 m^{-1/2}),$$

where in the second inequality we use $\|\alpha(0) - \alpha^*\|_2 \leq R_u$ and in the equality we use Lemma D.3. Thus, we conclude the proof of Theorem D.4. $\qquad\square$

Following the definition of $u_\alpha^0$ in (D.4), we define the local linearization of $Q_\omega$ at the initialization as

$$Q_\omega^0(s,a) = \frac{1}{\sqrt{m_Q}}\sum_{i=1}^{m_Q} b_i \cdot \mathbb{1}\big\{[\omega(0)]_i^\top(s,a) \geq 0\big\} \cdot [\omega]_i^\top(s,a).$$

Similarly, for $f_\theta$ we define

$$f_\theta^0(s,a) = \frac{1}{\sqrt{m_f}}\sum_{i=1}^{m_f} b_i \cdot \mathbb{1}\big\{[\theta(0)]_i^\top(s,a) \geq 0\big\} \cdot [\theta]_i^\top(s,a).$$

In the sequel, we show that Theorem D.4 implies both Theorems 4.5 and 4.6.

To obtain Theorem 4.5, we set $\rho = \widetilde{\sigma}_k$, $u_\alpha = f_\theta$, $v = \tau_{k+1} \cdot (\beta_k^{-1}Q_{\omega_k} + \tau_k^{-1}f_{\theta_k})$, $\mu = 0$, and $R_u = R_f$. Using $\tau_{k+1}, \tau_k$, and $\beta_k$ specified in Algorithm 1, we have

$$\mathbb{E}_{\widetilde{\sigma}_k}[(v(s,a))^2] \leq 2\tau_{k+1}^2 \cdot \big(\beta_k^{-2} \cdot \mathbb{E}_{\widetilde{\sigma}_k}[(Q_{\omega_k}(s,a))^2] + \tau_k^{-2} \cdot \mathbb{E}_{\widetilde{\sigma}_k}[(f_{\theta_k}(s,a))^2]\big)$$

$$\leq 4\mathbb{E}_{\widetilde{\sigma}_k}[(f_{\theta(0)}(s,a))^2] + 4R_f^2,$$

where in the second inequality we use $\tau_{k+1}^2\beta_k^{-2} + \tau_{k+1}^2\tau_k^{-2} \leq 1$ and the fact that $(Q_{\omega_k}(s,a))^2 \leq 2(Q_{\omega(0)}(s,a))^2 + 2R_Q^2$ and $(f_{\theta_k}(s,a))^2 \leq 2(f_{\theta(0)}(s,a))^2 + 2R_f^2$, which is a consequence of the 1-Lipschitz continuity of the neural network with respect to the weights. Also note that $Q_{\omega(0)}(s,a) = f_{\theta(0)}(s,a)$ due to the fact that $Q_{\omega_k}$ and $f_{\theta_k}$ share the same initialization. Thus, we have $\overline{v}_1 = 4$, $\overline{v}_2 = 4$, and $\overline{v}_3 = 0$. Moreover, by $f_{\theta^*}^0 = \Pi_{\mathcal{F}_{R_f,m_f}}\mathcal{T}f_{\theta^*}^0 = \Pi_{\mathcal{F}_{R_f,m}}(\tau_{k+1} \cdot (\beta_k^{-1}Q_{\omega_k} + \tau_k^{-1}f_{\theta_k}))$, we have

$$f_{\theta^*}^0 = \operatorname*{argmin}_{f \in \mathcal{F}_{R_f,m_f}} \big\{\big\|f - \tau_{k+1} \cdot (\beta_k^{-1}Q_{\omega_k} + \tau_k^{-1}f_{\theta_k})\big\|_{2,\widetilde{\sigma}_k}\big\},$$

which together with the fact that $\tau_{k+1} \cdot (\beta_k^{-1}Q_{\omega_k}^0(s,a) + \tau_k^{-1}f_{\theta_k}^0(s,a)) \in \mathcal{F}_{R_f,m_f}$ implies

$$\mathbb{E}_{\text{init},\widetilde{\sigma}_k}\big[\big(f_{\theta^*}^0(s,a) - \tau_{k+1} \cdot (\beta_k^{-1}Q_{\omega_k}(s,a) + \tau_k^{-1}f_{\theta_k}(s,a))\big)^2\big]$$

$$\leq \mathbb{E}_{\text{init},\widetilde{\sigma}_k}\big[\big(\tau_{k+1} \cdot (\beta_k^{-1}Q_{\omega_k}^0(s,a) + \tau_k^{-1}f_{\theta_k}^0(s,a)) - \tau_{k+1} \cdot (\beta_k^{-1}Q_{\omega_k}(s,a) + \tau_k^{-1}f_{\theta_k}(s,a))\big)^2\big]$$

$$\leq \tau_{k+1}^2\beta_k^{-2} \cdot \mathbb{E}_{\text{init},\widetilde{\sigma}_k}[(Q_{\omega_k}^0(s,a) - Q_{\omega_k}(s,a))^2] + \tau_{k+1}^2\tau_k^{-2} \cdot \mathbb{E}_{\text{init},\widetilde{\sigma}_k}[(f_{\theta_k}^0(s,a) - f_{\theta_k}(s,a))^2]$$

$$= O(R_f^3 m_f^{-1/2}). \qquad \text{(D.31)}$$

Finally, plugging (D.31) into Theorem D.4 for $f_\theta$, we obtain

$$\mathbb{E}_{\text{init},\widetilde{\sigma}_k}\left[\left(f_{\overline{\theta}}(s,a) - \tau_{k+1} \cdot (\beta_k^{-1} Q_{\omega_k}(s,a) + \tau_k^{-1} f_{\theta_k}(s,a))\right)^2\right]$$

$$\leq 2\mathbb{E}_{\text{init},\widetilde{\sigma}_k}[(f_{\overline{\theta}}(s,a) - f_{\theta^*}^0(s,a))^2] + 2\mathbb{E}_{\text{init},\widetilde{\sigma}_k}\left[\left(f_{\theta^*}^0(s,a) - \tau_{k+1} \cdot (\beta_k^{-1} Q_{\omega_k}(s,a) + \tau_k^{-1} f_{\theta_k}(s,a))\right)^2\right]$$

$$= O(R_f^2 T^{-1/2} + R_f^{5/2} m_f^{-1/4} + R_f^3 m_f^{-1/2})$$

which gives Theorem 4.5.

To obtain Theorem 4.6, we set $\rho = \sigma_k$, $u_\alpha = Q_\omega$, $v = (1-\gamma) \cdot r$, $\mu = \gamma$ and $R_u = R_Q$. Correspondingly, we have $\overline{v}_1 = 0, \overline{v}_2 = 0, \overline{v}_3 = R_{\max}^2$ and $u_{\alpha^*}^0 = Q_{\omega^*}^0$. Moreover, by the definition of the operator $\mathcal{T}$ in (D.20), we have $\mathcal{T} = \mathcal{T}^{\pi_{\theta_k}}$, which implies $Q^{\pi_{\theta_k}} = \mathcal{T}Q^{\pi_{\theta_k}}$. Meanwhile, by Assumption 4.3, we have $Q^{\pi_{\theta_k}} \in \mathcal{F}_{R_Q,m_Q}$, which implies $Q^{\pi_{\theta_k}} = \Pi_{\mathcal{F}_{R_Q,m_Q}} Q^{\pi_{\theta_k}} = \Pi_{\mathcal{F}_{R_Q,m_Q}} \mathcal{T}Q^{\pi_{\theta_k}}$. Since we already show that $Q_{\omega^*}^0$ is the unique solution to the equation $Q = \Pi_{\mathcal{F}_{R_Q,m_Q}} \mathcal{T}Q$, we obtain $Q_{\alpha^*}^0 = Q^{\pi_{\theta_k}}$. Therefore, we can substitute $Q_{\alpha^*}^0$ with $Q^{\pi_{\theta_k}}$ in Theorem D.4 to obtain Theorem 4.6.

# E    Proofs for Section 4.2

*Proof of Lemma 4.7.* We first have

$$\pi_{k+1}(a\,|\,s) = \exp\{\beta_k^{-1} Q^{\pi_{\theta_k}}(s,a) + \tau_k^{-1} f_{\theta_k}(s,a)\}/Z_{k+1}(s),$$

and

$$\pi_{\theta_{k+1}}(a\,|\,s) = \exp\{\tau_{k+1}^{-1} f_{\theta_{k+1}}(s,a)\}/Z_{\theta_{k+1}}(s).$$

Here $Z_{k+1}(s), Z_{\theta_{k+1}}(s) \in \mathbb{R}$ are normalization factors, which are defined as

$$Z_{k+1}(s) = \sum_{a' \in \mathcal{A}} \exp\{\beta_k^{-1} Q^{\pi_{\theta_k}}(s,a') + \tau_k^{-1} f_{\theta_k}(s,a')\},$$

$$Z_{\theta_{k+1}}(s) = \sum_{a' \in \mathcal{A}} \exp\{\tau_{k+1}^{-1} f_{\theta_{k+1}}(s,a')\}, \tag{E.1}$$

respectively. Thus, we reformulate the inner product in (4.5) as

$$\langle \log \pi_{\theta_{k+1}}(\cdot\,|\,s) - \log \pi_{k+1}(\cdot\,|\,s), \pi^*(\cdot\,|\,s) - \pi_{\theta_k}(\cdot\,|\,s)\rangle$$

$$= \langle \tau_{k+1}^{-1} f_{\theta_{k+1}}(s,\cdot) - (\beta_k^{-1} Q^{\pi_{\theta_k}}(s,\cdot) + \tau_k^{-1} f_{\theta_k}(s,\cdot)), \pi^*(\cdot\,|\,s) - \pi_{\theta_k}(\cdot\,|\,s)\rangle, \tag{E.2}$$

where we use the fact that

$$\langle \log Z_{k+1}(s) - \log Z_{\theta_{k+1}}(s), \pi^*(\cdot\,|\,s) - \pi_{\theta_k}(\cdot\,|\,s)\rangle$$

$$= (\log Z_{k+1}(s) - \log Z_{\theta_{k+1}}(s)) \sum_{a' \in \mathcal{A}} (\pi^*(a'\,|\,s) - \pi_{\theta_k}(a'\,|\,s)) = 0.$$

Thus, it remains to upper bound the right-hand side of (E.2). We first decompose it to two terms, namely the error from learning the Q-function and the error from fitting the improved policy, that is,

$$\langle \tau_{k+1}^{-1} f_{\theta_{k+1}}(s,\cdot) - (\beta_k^{-1} Q^{\pi_{\theta_k}}(s,\cdot) + \tau_k^{-1} f_{\theta_k}(s,\cdot)), \pi^*(\cdot\,|\,s) - \pi_{\theta_k}(\cdot\,|\,s)\rangle$$

$$= \underbrace{\langle \tau_{k+1}^{-1} f_{\theta_{k+1}}(s,\cdot) - (\beta_k^{-1} Q_{\omega_k}(s,\cdot) + \tau_k^{-1} f_{\theta_k}(s,\cdot)), \pi^*(\cdot\,|\,s) - \pi_{\theta_k}(\cdot\,|\,s)\rangle}_{\text{(i)}}$$

$$+ \underbrace{\langle \beta_k^{-1} Q_{\omega_k}(s,\cdot) - \beta_k^{-1} Q^{\pi_{\theta_k}}(s,\cdot), \pi^*(\cdot\,|\,s) - \pi_{\theta_k}(\cdot\,|\,s)\rangle}_{\text{(ii)}}. \tag{E.3}$$

**Upper Bounding (i):** We have

$$\langle \tau_{k+1}^{-1} f_{\theta_{k+1}}(s,\cdot) - (\beta_k^{-1} Q_{\omega_k}(s,\cdot) + \tau_k^{-1} f_{\theta_k}(s,\cdot)), \pi^*(\cdot\,|\,s) - \pi_{\theta_k}(\cdot\,|\,s)\rangle \tag{E.4}$$

$$= \left\langle \tau_{k+1}^{-1} f_{\theta_{k+1}}(s,\cdot) - (\beta_k^{-1} Q_{\omega_k}(s,\cdot) + \tau_k^{-1} f_{\theta_k}(s,\cdot)), \pi_0(\cdot\,|\,s) \cdot \left(\frac{\pi^*(\cdot\,|\,s)}{\pi_0(\cdot\,|\,s)} - \frac{\pi_{\theta_k}(\cdot\,|\,s)}{\pi_0(\cdot\,|\,s)}\right)\right\rangle.$$

Taking expectation with respect to $s \sim \nu^*$ on the both sides of (E.4) and using the Cauchy-Schwarz inequality, we obatin

$$\left| \mathbb{E}_{\nu^*} \left[ \langle \tau_{k+1}^{-1} f_{\theta_{k+1}}(s, \cdot) - (\beta_k^{-1} Q_{\omega_k}(s, \cdot) + \tau_k^{-1} f_{\theta_k}(s, \cdot)), \pi^*(\cdot \mid s) - \pi_{\theta_k}(\cdot \mid s) \rangle \right] \right|$$

$$= \left| \int_{\mathcal{S}} \left\langle \tau_{k+1}^{-1} f_{\theta_{k+1}}(s, \cdot) - (\beta_k^{-1} Q_{\omega_k}(s, \cdot) + \tau_k^{-1} f_{\theta_k}(s, \cdot)), \pi_0(\cdot \mid s) \cdot \nu_k(s) \cdot \left( \frac{\pi^*(\cdot \mid s)}{\pi_0(\cdot \mid s)} - \frac{\pi_{\theta_k}(\cdot \mid s)}{\pi_0(\cdot \mid s)} \right) \right\rangle \cdot \frac{\nu^*(s)}{\nu_k(s)} \mathrm{d}s \right|$$

$$= \left| \int_{\mathcal{S} \times \mathcal{A}} \left( \tau_{k+1}^{-1} f_{\theta_{k+1}}(s, a) - (\beta_k^{-1} Q_{\omega_k}(s, a) + \tau_k^{-1} f_{\theta_k}(s, a)) \right) \cdot \left( \frac{\sigma^*(a \mid s)}{\widetilde{\sigma}_k(a \mid s)} - \frac{\pi_{\theta_k}(a \mid s) \cdot \nu^*(s)}{\widetilde{\sigma}_k(a \mid s)} \right) \mathrm{d}\widetilde{\sigma}_k(s, a) \right|$$

$$\leq \mathbb{E}_{\widetilde{\sigma}_k} \left[ \left( \tau_{k+1}^{-1} f_{\theta_{k+1}}(s, a) - (\beta_k^{-1} Q_{\omega_k}(s, a) + \tau_k^{-1} f_{\theta_k}(s, a)) \right)^2 \right]^{1/2} \cdot \mathbb{E}_{\widetilde{\sigma}_k} \left[ \left| \frac{\mathrm{d}\sigma^*}{\mathrm{d}\widetilde{\sigma}_k} - \frac{\mathrm{d}(\pi_{\theta_k} \nu^*)}{\mathrm{d}\widetilde{\sigma}_k} \right|^2 \right]^{1/2}$$

$$\leq \tau_{k+1}^{-1} \epsilon_{k+1} \cdot \phi_k^*, \tag{E.5}$$

where in the last inequality we use the error bound in (4.3) and the definition of $\phi_k^*$ in (4.2).

**Upper Bounding (ii):** By the Cauchy-Schwartz inequality, we have

$$\left| \mathbb{E}_{\nu^*} \left[ \langle \beta_k^{-1} Q_{\omega_k}(s, \cdot) - \beta_k^{-1} Q^{\pi_{\theta_k}}(s, \cdot), \pi^*(\cdot \mid s) - \pi_{\theta_k}(\cdot \mid s) \rangle \right] \right|$$

$$= \left| \int_{\mathcal{S} \times \mathcal{A}} (\beta_k^{-1} Q_{\omega_k}(s, a) - \beta_k^{-1} Q^{\pi_{\theta_k}}(s, a)) \cdot \left( \frac{\pi^*(a \mid s)}{\pi_{\theta_k}(a \mid s)} - \frac{\pi_{\theta_k}(a \mid s)}{\pi_{\theta_k}(a \mid s)} \right) \cdot \frac{\nu^*(s)}{\nu_k(s)} \mathrm{d}\sigma_k(s, a) \right|$$

$$\leq \mathbb{E}_{\sigma_k} [(\beta_k^{-1} Q_{\omega_k}(s, a) - \beta_k^{-1} Q^{\pi_{\theta_k}}(s, a))^2]^{1/2} \cdot \mathbb{E}_{\sigma_k} \left[ \left| \frac{\mathrm{d}\sigma^*}{\mathrm{d}\sigma_k} - \frac{\mathrm{d}\nu^*}{\mathrm{d}\nu_k} \right|^2 \right]^{1/2}$$

$$\leq \beta_k^{-1} \epsilon_k' \cdot \psi_k^*, \tag{E.6}$$

where in the last inequality we use the error bound in (4.4) and the definition of $\psi_k^*$ in (4.2). Finally, combining (E.2), (E.3), (E.5), and (E.6), we have

$$\left| \mathbb{E}_{\nu^*} \left[ \langle \log \pi_{\theta_{k+1}}(\cdot \mid s) - \log \pi_{k+1}(\cdot \mid s), \pi^*(\cdot \mid s) - \pi_{\theta_k}(\cdot \mid s) \rangle \right] \right|$$

$$\leq \tau_{k+1}^{-1} \epsilon_{k+1} \cdot \phi_k^* + \beta_k^{-1} \epsilon_k' \cdot \psi_k^*,$$

which concludes the proof of Lemma 4.7. $\qquad \square$

*Proof of Lemma 4.8.* By the triangle inequality, we have

$$\| \tau_{k+1}^{-1} f_{\theta_{k+1}}(s, \cdot) - \tau_k^{-1} f_{\theta_k}(s, \cdot) \|_\infty^2$$

$$\leq 2 \| \tau_{k+1}^{-1} f_{\theta_{k+1}}(s, \cdot) - \tau_k^{-1} f_{\theta_k}(s, \cdot) - \beta_k^{-1} Q_{\omega_k}(s, \cdot) \|_\infty^2 + 2 \| \beta_k^{-1} Q_{\omega_k}(s, \cdot) \|_\infty^2. \tag{E.7}$$

For the first term on the right-hand side of (E.7), we have

$$\mathbb{E}_{\nu^*} [ \| \tau_{k+1}^{-1} f_{\theta_{k+1}}(s, \cdot) - \tau_k^{-1} f_{\theta_k}(s, \cdot) - \beta_k^{-1} Q_{\omega_k}(s, \cdot) \|_\infty^2 ] \leq |\mathcal{A}| \cdot \tau_{k+1}^{-2} \epsilon_{k+1}^2. \tag{E.8}$$

For the second term on the right-hand side of (E.7), we have

$$\mathbb{E}_{\nu^*} [ \| \beta_k^{-1} Q_{\omega_k}(s, \cdot) \|_\infty^2 ] \leq \beta_k^{-2} \cdot \mathbb{E}_{\nu^*} \left[ \max_{a \in \mathcal{A}} 2(Q_{\omega_0}(s, a))^2 + 2R_f^2 \right] = \beta_k^{-2} M, \tag{E.9}$$

where we use the 1-Lipschitz continuity of $Q_\omega$ in $\omega$ and the constraint $\| \omega_k - \omega_0 \|_2 \leq R_\omega$. Then, taking expectation with respect to $s \sim \nu^*$ on the both sides of (E.7) and plugging in (E.8) and (E.9), we finish the proof of Lemma 4.8. $\qquad \square$

# F   Proof of Corollary 4.10

*Proof.* By Theorems 4.5 and 4.6, we have $\epsilon_{k+1} = O(R_f^2 T^{-1/2} + R_f^{5/2} m_f^{-1/4} + R_f^3 m_f^{-1/2})$ and $\epsilon_k' = O(R_Q^2 T^{-1/2} + R_Q^{5/2} m_Q^{-1/4} + R_Q^3 m_Q^{-1/2})$, which gives

$$\tau_{k+1}^{-1} \epsilon_{k+1} \cdot \phi_{k+1}^* = O\big( kK^{-1/2} \cdot \phi_k^* \cdot (R_f^2 T^{-1/2} + R_f^{5/2} m_f^{-1/4}) \big),$$

$$|\mathcal{A}| \cdot \tau_{k+1}^{-2} \epsilon_{k+1}^2 = O\big( k^2 K^{-1} \cdot |\mathcal{A}| \cdot (R_f^2 T^{-1/2} + R_f^{5/2} m_f^{-1/4})^2 \big),$$

$$\beta_k^{-1} \epsilon_k' \cdot \psi_k^* = O\big( K^{-1/2} \cdot \psi_k^* \cdot (R_Q^2 T^{-1/2} + R_Q^{5/2} m_Q^{-1/4}) \big),$$

when $m_f = \Omega(R_f^2)$ and $m_Q = \Omega(R_Q^2)$.

Next, setting $m_f = R_f^{10} \cdot \Omega(K^6 \cdot \phi_k^{*4} + K^4 \cdot |\mathcal{A}|^2)$, $m_Q = \Omega(K^2 R_Q^{10} \cdot \psi_k^{*4})$ and $T = \Omega(K^3 R_f^4 \cdot \phi_k^{*2} + K R_Q^4 \cdot \psi_k^{*2})$, we further have

$$\varepsilon_k = \tau_{k+1}^{-1} \epsilon_{k+1} \cdot \phi_k^* + \beta_k^{-1} \epsilon_k' \cdot \psi_k^* = O(K^{-1}). \tag{F.1}$$

Meanwhile, setting $m_f = \Omega(K^4 R_f^{10} \cdot |\mathcal{A}|^2)$ and $T = \Omega(K^2 R_f^4 \cdot |\mathcal{A}|)$, we have

$$\varepsilon_k' = |\mathcal{A}| \cdot \tau_{k+1}^{-2} \epsilon_{k+1}^2 = O(K^{-1}). \tag{F.2}$$

Summing up (F.1) and (F.2) for $k + 1 \in [K]$ and plugging it into Theorem 4.9, we obtain

$$\min_{0 \le k \le K} \left\{ \mathcal{L}(\pi^*) - \mathcal{L}(\pi_{\theta_k}) \right\} \le \frac{\beta^2 \log |\mathcal{A}| + M + O(1)}{(1 - \gamma)\beta \cdot \sqrt{K}},$$

which completes the proof of Corollary 4.10. $\qquad \square$

# G  Proofs of Section 5

*Proof of Lemma 5.1.* The proof follows that of Lemma 6.1 in [24]. By the definition of $V^\pi(s)$ in (2.1), we have

$$\mathbb{E}_{\nu^*}[V^{\pi^*}(s)] = \sum_{t=0}^\infty \gamma^t \cdot \mathbb{E}_{a_t \sim \pi^*(\cdot \mid s_t), s_t \sim (\mathcal{P}^{\pi^*})^t \nu^*} \left[ (1 - \gamma) \cdot r(s_t, a_t) \right] \tag{G.1}$$

$$= \sum_{t=0}^\infty \gamma^t \cdot \mathbb{E}_{a_t \sim \pi^*(\cdot \mid s_t), s_t \sim (\mathcal{P}^{\pi^*})^t \nu^*} \left[ (1 - \gamma) \cdot r(s_t, a_t) + V^\pi(s_t) - V^\pi(s_t) \right]$$

$$= \sum_{t=0}^\infty \gamma^t \cdot \mathbb{E}_{s_{t+1} \sim \mathcal{P}(\cdot \mid s_t, a_t), a_t \sim \pi^*(\cdot \mid s_t), s_t \sim (\mathcal{P}^{\pi^*})^t \nu^*} \left[ (1 - \gamma) \cdot r(s_t, a_t) + \gamma \cdot V^\pi(s_{t+1}) - V^\pi(s_t) \right]$$

$$+ \mathbb{E}_{\nu^*}[V^\pi(s)],$$

where the third inequality is obtained by taking $\mathbb{E}_{\nu^*}[V^\pi(s_0)] = \mathbb{E}_{\nu^*}[V^\pi(s)]$ out and, correspondingly, delaying $V^\pi(s_t)$ by one time step to $V^\pi(s_{t+1})$ in each term of the summation. Note that for the advantage function, by definition of the action-value function, we have

$$A^\pi(s, a) = Q^\pi(s, a) - V^\pi(s) = (1 - \gamma) \cdot r(s, a) + \gamma \cdot \mathbb{E}_{s' \sim \mathcal{P}(\cdot \mid s, a)}[V^\pi(s')] - V^\pi(s),$$

which together with (G.1) implies

$$\mathbb{E}_{\nu^*}[V^{\pi^*}(s)] = \sum_{t=0}^\infty \gamma^t \cdot \mathbb{E}_{a_t \sim \pi^*(\cdot \mid s_t), s_t \sim (\mathcal{P}^{\pi^*})^t \nu^*}[A^\pi(s_t, a_t)] + \mathbb{E}_{\nu^*}[V^\pi(s)]$$

$$= (1 - \gamma)^{-1} \cdot \mathbb{E}_{\sigma^*}[A^\pi(s, a)] + \mathbb{E}_{\nu^*}[V^\pi(s)]. \tag{G.2}$$

Here the second equality follows from $(\mathcal{P}^{\pi^*})^t \nu^* = \nu^*$ for any $t \ge 0$ and $\sigma^* = \pi^* \nu^*$. Finally, note that for any given $s \in \mathcal{S}$,

$$\mathbb{E}_{\pi^*}[A^\pi(s, a)] = \mathbb{E}_{\pi^*}[Q^\pi(s, a) - V^\pi(s)] = \langle Q^\pi(s, \cdot), \pi^*(\cdot \mid s) \rangle - \langle Q^\pi(s, \cdot), \pi(\cdot \mid s) \rangle$$

$$= \langle Q^\pi(s, \cdot), \pi^*(\cdot \mid s) - \pi(\cdot \mid s) \rangle. \tag{G.3}$$

Plugging (G.3) into (G.2) and recalling the definition of $\mathcal{L}(\pi)$ in (4.6), we finish the proof of Lemma 5.1. $\qquad \square$

*Proof of Lemma 5.2.* First, we have

$$\mathrm{KL}(\pi^*(\cdot \mid s) \,\|\, \pi_{\theta_k}(\cdot \mid s)) - \mathrm{KL}(\pi^*(\cdot \mid s) \,\|\, \pi_{\theta_{k+1}}(\cdot \mid s))$$

$$= \left\langle \log(\pi_{\theta_{k+1}}(\cdot \mid s)/\pi_{\theta_k}(\cdot \mid s)), \pi^*(\cdot \mid s) \right\rangle$$

$$= \left\langle \log(\pi_{\theta_{k+1}}(\cdot \mid s)/\pi_{\theta_k}(\cdot \mid s)), \pi^*(\cdot \mid s) - \pi_{\theta_{k+1}}(\cdot \mid s) \right\rangle + \mathrm{KL}(\pi_{\theta_{k+1}}(\cdot \mid s) \,\|\, \pi_{\theta_k}(\cdot \mid s))$$

$$= \left\langle \log(\pi_{\theta_{k+1}}(\cdot \mid s)/\pi_{\theta_k}(\cdot \mid s)) - \beta_k^{-1} Q^{\pi_{\theta_k}}(s, \cdot), \pi^*(\cdot \mid s) - \pi_{\theta_k}(\cdot \mid s) \right\rangle$$

$$+ \beta_k^{-1} \cdot \left\langle Q^{\pi_{\theta_k}}(s, \cdot), \pi^*(\cdot \mid s) - \pi_{\theta_k}(\cdot \mid s) \right\rangle + \mathrm{KL}(\pi_{\theta_{k+1}}(\cdot \mid s) \,\|\, \pi_{\theta_k}(\cdot \mid s))$$

$$+ \left\langle \log(\pi_{\theta_{k+1}}(\cdot \mid s)/\pi_{\theta_k}(\cdot \mid s)), \pi_{\theta_k}(\cdot \mid s) - \pi_{\theta_{k+1}}(\cdot \mid s) \right\rangle. \tag{G.4}$$

Recall that $\pi_{k+1} \propto \exp\{\tau_k^{-1} f_{\theta_k} + \beta_k^{-1} Q^{\pi_{\theta_k}}\}$ and $Z_{k+1}(s)$ and $Z_{\theta_k}(s)$ are defined in (E.1). Also recall that we have $\langle \log Z_{\theta_k}(s), \pi(\cdot \mid s) - \pi'(\cdot \mid s) \rangle = \langle \log Z_k(s), \pi(\cdot \mid s) - \pi'(\cdot \mid s) \rangle = 0$ for all $k$,

$\pi$, and $\pi'$, which implies that, on the right-hand-side of (G.4),

$$\langle \log \pi_{\theta_k}(\cdot \mid s) + \beta_k^{-1} Q^{\pi_{\theta_k}}(s, \cdot), \pi^*(\cdot \mid s) - \pi_{\theta_k}(\cdot \mid s) \rangle$$

$$= \langle \tau_k^{-1} f_{\theta_k}(s, \cdot) + \beta_k^{-1} Q^{\pi_{\theta_k}}(s, \cdot), \pi^*(\cdot \mid s) - \pi_{\theta_k}(\cdot \mid s) \rangle - \langle \log Z_{\theta_k}(s), \pi^*(\cdot \mid s) - \pi_{\theta_k}(\cdot \mid s) \rangle$$

$$= \langle \tau_k^{-1} f_{\theta_k}(s, \cdot) + \beta_k^{-1} Q^{\pi_{\theta_k}}(s, \cdot), \pi^*(\cdot \mid s) - \pi_{\theta_k}(\cdot \mid s) \rangle - \langle \log Z_{k+1}(s), \pi^*(\cdot \mid s) - \pi_{\theta_k}(\cdot \mid s) \rangle$$

$$= \langle \log \pi_{k+1}(\cdot \mid s), \pi^*(\cdot \mid s) - \pi_{\theta_k}(\cdot \mid s) \rangle, \tag{G.5}$$

and

$$\langle \log(\pi_{\theta_{k+1}}(\cdot \mid s)/\pi_{\theta_k}(\cdot \mid s)), \pi_{\theta_k}(\cdot \mid s) - \pi_{\theta_{k+1}}(\cdot \mid s) \rangle$$

$$= \langle \tau_{k+1}^{-1} f_{\theta_{k+1}}(s, \cdot) - \tau_k^{-1} f_{\theta_k}(s, \cdot), \pi_{\theta_k}(\cdot \mid s) - \pi_{\theta_{k+1}}(\cdot \mid s) \rangle$$

$$\quad - \langle \log Z_{\theta_{k+1}}(s), \pi_{\theta_k}(\cdot \mid s) - \pi_{\theta_{k+1}}(\cdot \mid s) \rangle + \langle \log Z_{\theta_k}(s), \pi_{\theta_k}(\cdot \mid s) - \pi_{\theta_{k+1}}(\cdot \mid s) \rangle$$

$$= \langle \tau_{k+1}^{-1} f_{\theta_{k+1}}(s, \cdot) - \tau_k^{-1} f_{\theta_k}(s, \cdot), \pi_{\theta_k}(\cdot \mid s) - \pi_{\theta_{k+1}}(\cdot \mid s) \rangle. \tag{G.6}$$

Plugging (G.5) and (G.6) into (G.4), we obtain

$$\mathrm{KL}(\pi^*(\cdot \mid s) \,\|\, \pi_{\theta_k}(\cdot \mid s)) - \mathrm{KL}(\pi^*(\cdot \mid s) \,\|\, \pi_{\theta_{k+1}}(\cdot \mid s)) \tag{G.7}$$

$$= \langle \log(\pi_{\theta_{k+1}}(\cdot \mid s)/\pi_{k+1}(\cdot \mid s)), \pi^*(\cdot \mid s) - \pi_{\theta_k}(\cdot \mid s) \rangle + \beta_k^{-1} \cdot \langle Q^{\pi_{\theta_k}}(s, \cdot), \pi^*(\cdot \mid s) - \pi_{\theta_k}(\cdot \mid s) \rangle$$

$$\quad + \langle \tau_{k+1}^{-1} f_{\theta_{k+1}}(s, \cdot) - \tau_k^{-1} f_{\theta_k}(s, \cdot), \pi_{\theta_k}(\cdot \mid s) - \pi_{\theta_{k+1}}(\cdot \mid s) \rangle + \mathrm{KL}(\pi_{\theta_{k+1}}(\cdot \mid s) \,\|\, \pi_{\theta_k}(\cdot \mid s))$$

$$\geq \langle \log(\pi_{\theta_{k+1}}(\cdot \mid s)/\pi_{k+1}(\cdot \mid s)), \pi^*(\cdot \mid s) - \pi_{\theta_k}(\cdot \mid s) \rangle + \beta_k^{-1} \cdot \langle Q^{\pi_{\theta_k}}(s, \cdot), \pi^*(\cdot \mid s) - \pi_{\theta_k}(\cdot \mid s) \rangle$$

$$\quad + \langle \tau_{k+1}^{-1} f_{\theta_{k+1}}(s, \cdot) - \tau_k^{-1} f_{\theta_k}(s, \cdot), \pi_{\theta_k}(\cdot \mid s) - \pi_{\theta_{k+1}}(\cdot \mid s) \rangle + 1/2 \cdot \|\pi_{\theta_{k+1}}(\cdot \mid s) - \pi_{\theta_k}(\cdot \mid s)\|_1^2,$$

where in the last inequality we use the Pinsker's inequality. Rearranging the terms in (G.7), we finish the proof of Lemma 5.2. $\qquad\square$

[Supplementary Material 2 · NeuralPPO_camera_appendix.pdf]

# A   Algorithms in Section 3

We present the algorithms for solving the subproblems of policy improvement and policy evaluation in Section 3.

---

**Algorithm 2** Policy Improvement via SGD

---

1: **Require:** MDP $(\mathcal{S}, \mathcal{A}, \mathcal{P}, r, \gamma)$, current energy function $f_{\theta_k}$, initial weights $b_i$, $[\theta(0)]_i$ ($i \in [m_f]$), number of iterations $T$, sample $\{(s_t, a_t^0)\}_{t=1}^T$
2: Set stepsize $\eta \leftarrow T^{-1/2}$
3: **for** $t = 0, \ldots, T - 1$ **do**
4:    $(s, a) \leftarrow (s_{t+1}, a_{t+1}^0)$
5:    $\theta(t + 1/2) \leftarrow \theta(t) - \eta \cdot \left( f_{\theta(t)}(s, a) - \tau_{k+1} \cdot (\beta_k^{-1} Q_{\omega_k}(s, a) + \tau_k^{-1} f_{\theta_k}(s, a)) \right) \cdot \nabla_\theta f_{\theta(t)}(s, a)$
6:    $\theta(t + 1) \leftarrow \operatorname{argmin}_{\theta \in \mathcal{B}^0(R_f)} \{ \| \theta - \theta(t + 1/2) \|_2 \}$
7: **end for**
8: Average over path $\overline{\theta} \leftarrow 1/T \cdot \sum_{t=0}^{T-1} \theta(t)$
9: **Output:** $f_{\overline{\theta}}$

---

---

**Algorithm 3** Policy Evaluation via TD

---

1: **Require:** MDP $(\mathcal{S}, \mathcal{A}, \mathcal{P}, r, \gamma)$, initial weights $b_i$, $[\omega(0)]_i$ ($i \in [m_Q]$), number of iterations $T$, sample $\{(s_t, a_t, s_t', a_t')\}_{t=1}^T$
2: Set stepsize $\eta \leftarrow T^{-1/2}$
3: **for** $t = 0, \ldots, T - 1$ **do**
4:    $(s, a, s', a') \leftarrow (s_{t+1}, a_{t+1}, s_{t+1}', a_{t+1}')$
5:    $\omega(t + 1/2) \leftarrow \omega(t) - \eta \cdot \left( Q_{\omega(t)}(s, a) - (1 - \gamma) \cdot r(s, a) - \gamma Q_{\omega(t)}(s', a') \right) \cdot \nabla_\omega Q_{\omega(t)}(s, a)$
6:    $\omega(t + 1) \leftarrow \operatorname{argmin}_{\omega \in \mathcal{B}^0(R_Q)} \{ \| \omega - \omega(t + 1/2) \|_2 \}$
7: **end for**
8: Average over path $\overline{\omega} \leftarrow 1/T \cdot \sum_{t=0}^{T-1} \omega(t)$
9: **Output:** $Q_{\overline{\omega}}$

---

# B   Supplementary Lemma in Section 3

The following lemma quantifies the policy improvement error in terms of the distance between polices, which is induced by solving (3.5).

**Lemma B.1.** Suppose that $\pi_{\theta_{k+1}} \propto \exp\{\tau_{k+1}^{-1} f_{\theta_{k+1}}\}$ satisfies

$$\mathbb{E}_{\widetilde{\sigma}_k} \left[ \left( f_{\theta_{k+1}}(s, a) - \tau_{k+1} \cdot (\beta_k^{-1} Q_{\omega_k}(s, a) + \tau_k^{-1} f_{\theta_k}(s, a)) \right)^2 \right] \leq \epsilon_{k+1}.$$

We have

$$\mathbb{E}_{\widetilde{\sigma}_k}[(\pi_{\theta_{k+1}}(a \,|\, s) - \widehat{\pi}_{k+1}(a \,|\, s))^2] \leq \tau_{k+1}^{-2} \epsilon_{k+1}/16,$$

where $\widehat{\pi}_{k+1}$ is defined in (3.4).

*Proof.* Let $\tau_{k+1}^{-1} \widehat{f}_{k+1} = \beta_k^{-1} Q_{\omega_k} + \tau_k^{-1} f_{\theta_k}$. Since an energy-based policy $\pi \propto \exp\{\tau^{-1} f\}$ is continuous with respect to $f$, by the mean value theorem, we have

$$|\pi_{\theta_{k+1}}(a \,|\, s) - \widehat{\pi}_{k+1}(a \,|\, s)| = \left| \frac{\exp\{\tau_{k+1}^{-1} f_{\theta_{k+1}}(s, a)\}}{\sum_{a' \in \mathcal{A}} \exp\{\tau_{k+1}^{-1} f_{\theta_{k+1}}(s, a')\}} - \frac{\exp\{\tau_{k+1}^{-1} \widehat{f}_{k+1}(s, a)\}}{\sum_{a' \in \mathcal{A}} \exp\{\tau_{k+1}^{-1} \widehat{f}_{k+1}(s, a')\}} \right|$$

$$= \left| \frac{\partial}{\partial f(s, a)} \left( \frac{\exp\{\tau_{k+1}^{-1} \widetilde{f}(s, a)\}}{\sum_{a' \in \mathcal{A}} \exp\{\tau_{k+1}^{-1} \widetilde{f}(s, a')\}} \right) \right| \cdot |f_{\theta_{k+1}}(s, a) - \widehat{f}_{k+1}(s, a)|,$$

where $\widetilde{f}$ is a function determined by $f_{\theta_{k+1}}$ and $\widehat{f}_{k+1}$. Furthermore, we have

$$\left| \frac{\partial}{\partial f(s, a)} \left( \frac{\exp\{\tau_{k+1}^{-1} f(s, a)\}}{\sum_{a' \in \mathcal{A}} \exp\{\tau_{k+1}^{-1} f(s, a')\}} \right) \right| = \tau_{k+1}^{-1} \cdot \pi(a \,|\, s) \cdot (1 - \pi(a \,|\, s)) \leq \tau_{k+1}^{-1}/4.$$

Therefore, we obtain

$$(\pi_{\theta_{k+1}}(a\,|\,s) - \widehat{\pi}_{k+1}(a\,|\,s))^2$$
$$\leq \tau_{k+1}^{-2}/16 \cdot \big(f_{\theta_{k+1}}(s,a) - \tau_{k+1} \cdot (\beta_k^{-1} Q_{\omega_k}(s,a) + \tau_k^{-1} f_{\theta_k}(s,a))\big)^2. \tag{B.1}$$

Taking expectation $\mathbb{E}_{\widetilde{\sigma}_k}[\,\cdot\,]$ on the both sides of (B.1), we finally obtain

$$\mathbb{E}_{\widetilde{\sigma}_k}[(\pi_{\theta_{k+1}}(a\,|\,s) - \widehat{\pi}_{k+1}(a\,|\,s))^2]$$
$$\leq \tau_{k+1}^{-2}/16 \cdot \mathbb{E}_{\widetilde{\sigma}_k}\big[\big(f_{\theta_{k+1}}(s,a) - \tau_{k+1} \cdot (\beta_k^{-1} Q_{\omega_0}(s,a) + \tau_k^{-1} f_{\theta_k}(s,a))\big)^2\big] \leq \tau_{k+1}^{-2} \epsilon_{k+1}/16,$$

which concludes the proof of Lemma B.1. $\qquad\square$

Lemma B.1 ensures that if the policy improvement error $\epsilon_{k+1}$ is small, then the corresponding improved policy $\pi_{\theta_{k+1}}$ is close to the ideal improved policy $\widehat{\pi}_{k+1}$, which justifies solving the subproblem in (3.5) for policy improvement.

## C  Proof of Proposition 3.1

*Proof.* The subproblem of policy improvement for solving $\widehat{\pi}_{k+1}$ takes the form

$$\max_{\pi} \; \mathbb{E}_{\nu_k}\big[\langle \pi(\cdot\,|\,s), Q_{\omega_k}(s, \cdot)\rangle - \beta_k \cdot \mathrm{KL}(\pi(\cdot\,|\,s)\,\|\,\pi_{\theta_k}(\cdot\,|\,s))\big]$$

$$\text{subject to } \sum_{a\in\mathcal{A}} \pi(a\,|\,s) = 1, \;\text{ for any } s \in \mathcal{S}.$$

The Lagrangian of the above maximization problem takes the form

$$\int_{s\in\mathcal{S}} \big[\langle \pi(\cdot\,|\,s), Q_{\omega_k}(s, \cdot)\rangle - \beta_k \cdot \mathrm{KL}(\pi(\cdot\,|\,s)\,\|\,\pi_{\theta_k}(\cdot\,|\,s))\big]\nu_k(\mathrm{d}s) + \int_{s\in\mathcal{S}}\bigg(\sum_{a\in\mathcal{A}} \pi(a\,|\,s) - 1\bigg)\lambda(\mathrm{d}s).$$

Plugging in $\pi_{\theta_k}(s,a) = \exp\{\tau_k^{-1} f_{\theta_k}(s,a)\}/\sum_{a'\in\mathcal{A}} \exp\{\tau_k^{-1} f_{\theta_k}(s,a')\}$, we obtain the optimality condition

$$Q_{\omega_k}(s,a) + \beta_k\tau_k^{-1} f_{\theta_k}(s,a) - \beta_k \cdot \bigg[\log\bigg(\sum_{a'\in\mathcal{A}} \exp\{\tau_k^{-1} f_{\theta_k}(s,a')\}\bigg) + \log\pi(a\,|s) + 1\bigg] + \frac{\lambda(s)}{\nu_k(s)} = 0,$$

for any $a \in \mathcal{A}$ and $s \in \mathcal{S}$. Note that $\log(\sum_{a'\in\mathcal{A}} \exp\{\tau_k^{-1} f_{\theta_k}(s,a')\})$ is determined by the state $s$ only. Hence, we have $\widehat{\pi}_{k+1}(a\,|\,s) \propto \exp\{\beta_k^{-1} Q_{\omega_k}(s,a) + \tau_k^{-1} f_{\theta_k}(s,a)\}$ for any $a \in \mathcal{A}$ and $s \in \mathcal{S}$, which concludes the proof of Proposition 3.1. $\qquad\square$

## D  Proofs for Section 4.1

The proofs in this section generalizes those of [7, 5] under a unified framework, which accounts for both SGD, and TD, which uses stochastic semi-gradient. In particular, we develop a unified global convergence analysis of a meta-algorithm with the following update,

$$\alpha(t + 1/2) \leftarrow \alpha(t) - \eta \cdot (u_{\alpha(t)}(s,a) - v(s,a) - \mu \cdot u_{\alpha(t)}(s',a')) \cdot \nabla_\alpha u_{\alpha(t)}(s,a), \tag{D.1}$$

$$\alpha(t + 1) \leftarrow \Pi_{\mathcal{B}^0(R_u)}(\alpha(1 + 1/2)) = \operatorname*{argmin}_{\alpha\in\mathcal{B}^0(R_u)} \|\alpha - \alpha(t + 1/2)\|_2, \tag{D.2}$$

where $\mu \in [0, 1)$ is a constant, $(s, a, s', a')$ is sampled from a stationary distribution $\rho$, and $u_\alpha$ is parametrized by the two-layer neural network $\mathrm{NN}(\alpha; m)$ defined in (3.1). The random initialization of $u_\alpha$ is given in (3.2). We denote by $\mathbb{E}_{\mathrm{init}}[\,\cdot\,]$ the expectation over such random initialization and $\mathbb{E}_\rho[\,\cdot\,]$ the expectation over $(s, a)$ conditional on the random initialization.

Such a meta-algorithm recovers SGD for policy improvement in (3.5) when we set $\rho = \widetilde{\sigma}_k$, $u_\alpha = f_\theta$, $v = \tau_{k+1} \cdot (\beta_k^{-1} Q_{\omega_k} + \tau_k^{-1} f_{\theta_k})$, $\mu = 0$, and $R_u = R_f$, and recovers TD for policy evaluation in (3.8) when we set $\rho = \sigma_k$, $u_\alpha = Q_\omega$, $v = (1 - \gamma) \cdot r$, $\mu = \gamma$, and $R_u = R_Q$.

To unify our analysis for SGD and TD, we assume that $v$ in (D.1) satisfies

$$\mathbb{E}_\rho[(v(s,a))^2] \leq \overline{v}_1 \cdot \mathbb{E}_\rho[(u_{\alpha(0)}(s,a))^2] + \overline{v}_2 \cdot R_u^2 + \overline{v}_3$$

for constants $\overline{v}_1, \overline{v}_2, \overline{v}_3 \geq 0$. Also, without loss of generality, we assume that $\|(s,a)\|_2 \leq 1$ for any $s \in \mathcal{S}$ and $a \in \mathcal{A}$. In Section D.2, we set $\overline{v}_1 = 4$, $\overline{v}_2 = 4$, and $\overline{v}_3 = 0$ for SGD, and $\overline{v}_1 = 0$, $\overline{v}_2 = 0$, and $\overline{v}_3 = R_{\max}$ for TD, respectively.

For notational simplicity, we define the residual $\delta_\alpha(s, a, s', a') = u_\alpha(s, a) - v(s, a) - \mu \cdot u_\alpha(s', a')$. We denote by

$$g_{\alpha(t)}(s, a, s', a') = \delta_{\alpha(t)}(s, a, s', a') \cdot \nabla_\alpha u_{\alpha(t)}(s, a), \quad \bar{g}_{\alpha(t)} = \mathbb{E}_\rho[g_t(s, a, s', a')] \qquad \text{(D.3)}$$

the stochastic update vector at the $t$-th iteration and its population mean, respectively. For SGD, $g_{\alpha(t)}(s, a, s', a')$ corresponds to the stochastic gradient, while for TD, $g_{\alpha(t)}(s, a, s', a')$ corresponds to the stochastic semigradient.

Note that the gradient of $u_\alpha(s, a)$ with respect to $\alpha$ takes the form

$$\nabla_\alpha u_\alpha(s, a) = 1/\sqrt{m} \cdot \left(b_1 \cdot \mathbb{1}\{[\alpha]_1^\top(s, a) > 0\} \cdot (s, a)^\top, \ldots, b_m \cdot \mathbb{1}\{[\alpha]_m^\top(s, a) > 0\} \cdot (s, a)^\top\right)^\top \in \mathbb{R}^{md}$$

almost everywhere, which yields

$$\|\nabla_\alpha u_\alpha(s, a)\|_2^2 = \frac{1}{m} \sum_{i=1}^m \mathbb{1}\{[\alpha]_i^\top(s, a) > 0\} \cdot \|(s, a)\|_2^2 \leq 1.$$

Therefore, $u_\alpha(s, a)$ is 1-Lipschitz continuous with respect to $\alpha$.

In the following, we first show in Section D.1 that the overparametrization of $u_\alpha$ ensures that it behaves similarly as its local linearization at the random initialization $\alpha(0)$ defined in (3.2). Then in Section D.2, we establish the global convergence of the meta-algorithm defined in (D.1) and (D.2), which implies the global convergence of SGD and TD.

## D.1   Local Linearization

In this section, we first define a local linearization of the two-layer neural network $u_\alpha$ at its random initialization and then characterize the error induced by local linearization. We define

$$u_\alpha^0(s, a) = \frac{1}{\sqrt{m}} \sum_{i=1}^m b_i \cdot \mathbb{1}\{[\alpha(0)]_i^\top(s, a) > 0\} \cdot [\alpha]_i^\top(s, a). \qquad \text{(D.4)}$$

The linearity of $u_\alpha^0$ with respect to $\alpha$ yields

$$\langle \nabla_\alpha u_\alpha^0(s, a), \alpha \rangle = u_\alpha^0(s, a). \qquad \text{(D.5)}$$

The following lemma characterizes how far $u_{\alpha(t)}^0$ deviates from $u_{\alpha(t)}$ for $\alpha(t) \in \mathcal{B}^0(R_u)$.

**Lemma D.1.** For any $\alpha' \in \mathcal{B}^0(R_u)$, we have

$$\mathbb{E}_{\text{init}, \rho}[(u_{\alpha'}(s, a) - u_{\alpha'}^0(s, a))^2] = O(R_u^3 m^{-1/2}).$$

*Proof.* By the definition of $u_\alpha$ in (3.1), we have

$$|u_{\alpha'}(s, a) - u_{\alpha'}^0(s, a)| \qquad \text{(D.6)}$$

$$\leq \frac{1}{\sqrt{m}} \left| \sum_{i=1}^m b_i \cdot \left(\mathbb{1}\{[\alpha(0)]_i^\top(s, a) > 0\} - \mathbb{1}\{[\alpha(0)]_i^\top(s, a) > 0\}\right) \cdot \left(|[\alpha(0)]_i^\top(s, a)| + \|[\alpha']_i - [\alpha(0)]_i\|_2\right) \right|$$

$$\leq \frac{1}{\sqrt{m}} \sum_{i=1}^m \mathbb{1}\{[\alpha(0)]_i^\top(s, a) \leq \|[\alpha']_i - [\alpha(0)]_i\|_2\} \cdot \left(|[\alpha(0)]_i^\top(s, a)| + \|[\alpha']_i - [\alpha(0)]_i\|_2\right),$$

where the second inequality follows from $|b_i| = 1$ and the fact that

$$\mathbb{1}\{[\alpha(t)]_i^\top(s, a) > 0\} \neq \mathbb{1}\{[\alpha(0)]_i^\top(s, a) > 0\}$$

implies

$$|[\alpha(0)]_i^\top(s, a)| \leq |[\alpha(t)]_i^\top(s, a) - [\alpha(0)]_i^\top(s, a)| \leq \|[\alpha(0)]_i - [\alpha(t)]_i\|_2.$$

Next, applying the inequality $\mathbb{1}\{|z| \leq y\}|z| \leq \mathbb{1}\{|z| \leq y\}y$ to the right-hand side of (D.6), we obtain

$$|u_{\alpha'}(s, a) - u_{\alpha'}^0(s, a)|$$

$$\leq \frac{2}{\sqrt{m}} \sum_{i=1}^m \mathbb{1}\{[\alpha(0)]_i^\top(s, a) \leq \|[\alpha']_i - [\alpha(0)]_i\|_2\} \cdot \|[\alpha']_i - [\alpha(0)]_i\|_2. \qquad \text{(D.7)}$$

Further applying the Cauchy-Schwarz inequality to (D.7) and invoking the upper bound $\|\alpha' - \alpha(0)\|_2 \leq R_u$, we obtain

$$|u_{\alpha'}(s, a) - u_{\alpha'}^0(s, a)|^2 \leq \frac{4R_u^2}{m} \sum_{i=1}^m \mathbb{1}\{[\alpha(0)]_i^\top(s, a) \leq \|[\alpha']_i - [\alpha(0)]_i\|_2\}. \qquad \text{(D.8)}$$

Taking expectation on the both sides and invoking Assumption 4.4, we obtain

$$\mathbb{E}_{\text{init},\rho}[(u_{\alpha'}(s,a) - u_{\alpha'}^0(s,a))^2] \leq \frac{4cR_u^2}{m} \cdot \mathbb{E}_{\text{init}}\left[\sum_{i=1}^{m} \|[\alpha']_i - [\alpha(0)]_i\|_2 / \|[\alpha(0)]_i\|_2\right]. \quad \text{(D.9)}$$

By the Cauchy-Schwartz inequality, we have

$$\mathbb{E}_{\text{init}}\left[\sum_{i=1}^{m} \|[\alpha']_i - [\alpha(0)]_i\|_2 / \|[\alpha(0)]_i\|_2\right] \leq \mathbb{E}_{\text{init}}\left[\sum_{i=1}^{m} \|[\alpha']_i - [\alpha(0)]_i\|_2^2\right]^{1/2} \cdot \mathbb{E}_{\text{init}}\left[\sum_{i=1}^{m} \|[\alpha(0)]_i\|_2^{-2}\right]^{1/2}$$

$$\leq R_u \cdot \mathbb{E}_{\text{init}}\left[\sum_{i=1}^{m} \|[\alpha(0)]_i\|_2^{-2}\right]^{1/2},$$

where the second inequality follows from $\sum_{i=1}^{m} \|[\alpha']_i - [\alpha(0)]_i\|_2^2 = \|\alpha' - \alpha(0)\|_2^2 \leq R_u^2$. Therefore, we have that the right-hand side of (D.9) is $O(R_u^3 m^{-1/2})$. Thus, we obtain

$$\mathbb{E}_{\text{init},\rho}[(u_{\alpha'}(s,a) - u_{\alpha'}^0(s,a))^2] = O(R_u^3 m^{-1/2}),$$

which concludes the proof of Lemma D.1. □

Corresponding to $u_\alpha^0$ defined in (D.4), let $\delta_\alpha^0(s,a,s',a') = u_\alpha^0(s,a) - v(s,a) - \mu \cdot u_\alpha^0(s',a')$. We define the local linearization of $\bar{g}_{\alpha(t)}$, which is defined in (D.3), as

$$\bar{g}_{\alpha(t)}^0 = \mathbb{E}_\rho[\delta_{\alpha(t)}^0(s,a,s',a') \cdot \nabla_\alpha u_{\alpha(t)}^0(s,a)]. \quad \text{(D.10)}$$

The following lemma characterizes the difference between $\bar{g}_{\alpha(t)}^0$ and $\bar{g}_{\alpha(t)}$.

**Lemma D.2.** For any $t \in [T]$, we have

$$\mathbb{E}_{\text{init}}[\|\bar{g}_{\alpha(t)} - \bar{g}_{\alpha(t)}^0\|_2^2] = O(R_u^3 m^{-1/2}).$$

*Proof.* By the definition of $\bar{g}_{\alpha(t)}^0$ and $\bar{g}_{\alpha(t)}$ in (D.10) and (D.3), we have

$$\|\bar{g}_{\alpha(t)} - \bar{g}_{\alpha(t)}^0\|_2^2 = \|\mathbb{E}_\rho[\delta_{\alpha(t)}(s,a,s',a') \cdot \nabla_\alpha u_{\alpha(t)}(s,a) - \delta_{\alpha(t)}^0(s,a,s',a') \cdot \nabla_\alpha u_{\alpha(t)}^0(s,a)]\|_2^2$$

$$\leq 2\underbrace{\mathbb{E}_\rho\big[|\delta_{\alpha(t)}(s,a,s',a') - \delta_{\alpha(t)}^0(s,a,s',a')|^2 \cdot \|\nabla_\alpha u_{\alpha(t)}(s,a)\|_2^2\big]}_{(i)} \quad \text{(D.11)}$$

$$+ 2\underbrace{\mathbb{E}_\rho\big[|\delta_{\alpha(t)}^0(s,a,s',a')| \cdot \|\nabla_\alpha u_{\alpha(t)}(s,a) - \nabla_\alpha u_{\alpha(t)}^0(s,a)\|_2\big]^2}_{(ii)}.$$

**Upper Bounding (i):** We have $\|\nabla_\alpha u_{\alpha(t)}(s,a)\|_2 \leq 1$ as $\|(s,a)\|_2 \leq 1$. Note that the difference between $\delta_{\alpha(t)}$ and $\delta_{\alpha(t)}^0$ takes the form

$$\delta_{\alpha(t)}(s,a,s',a') - \delta_{\alpha(t)}^0(s,a,s',a') = (u_{\alpha(t)}(s,a) - u_{\alpha(t)}^0(s,a)) - \mu \cdot (u_{\alpha(t)}(s',a') - u_{\alpha(t)}^0(s',a')).$$

Taking expectation on the both sides, we obtain

$$\mathbb{E}_{\text{init},\rho}[|\delta_{\alpha(t)}(s,a,s',a') - \delta_{\alpha(t)}^0(s,a,s',a')|^2]$$

$$\leq 2\mathbb{E}_{\text{init},\rho}[(u_{\alpha(t)}(s,a) - u_{\alpha(t)}^0(s,a))^2] + 2\mu^2 \cdot \mathbb{E}_{\text{init},\rho}[(u_{\alpha(t)}(s',a') - u_{\alpha(t)}^0(s',a'))^2]$$

$$= 4\mathbb{E}_{\text{init},\rho}[(u_{\alpha(t)}(s,a) - u_{\alpha(t)}^0(s,a))^2],$$

where the equality follows from $|\mu| \leq 1$ and the fact that $(s,a)$ and $(s',a')$ have the same marginal distribution. Thus, by Lemma D.1, we have that (i) in (D.11) is $O(R_u^3 m^{-1/2})$.

**Upper Bounding (ii):** First, by the Hölder's inequality, we have

$$\mathbb{E}_\rho\big[|\delta_{\alpha(t)}^0(s,a,s',a')| \cdot \|\nabla_\alpha u_{\alpha(t)}(s,a) - \nabla_\alpha u_{\alpha(t)}^0(s,a)\|_2\big]^2$$

$$\leq \mathbb{E}_\rho[|\delta_{\alpha(t)}^0(s,a,s',a')|^2] \cdot \mathbb{E}_\rho[\|\nabla_\alpha u_{\alpha(t)}(s,a) - \nabla_\alpha u_{\alpha(t)}^0(s,a)\|_2^2].$$

We use $|u_{\alpha(t)}^0(s,a) - u_{\alpha(0)}^0(s,a)| \leq \|\alpha(t) - \alpha(0)\|_2 \leq R_u$ to obtain

$$|\delta_{\alpha(t)}^0(s,a,s',a')|^2 = (u_{\alpha(t)}^0(s,a) - v(s,a) - \mu \cdot u_{\alpha(t)}^0(s',a'))^2$$

$$\leq 3\big((u_{\alpha(t)}^0(s,a))^2 + (v(s,a))^2 + \mu^2 \cdot (u_{\alpha(t)}^0(s',a'))^2\big)$$

$$\leq 3(u_{\alpha(0)}^0(s,a))^2 + 3(u_{\alpha(0)}^0(s',a'))^2 + 6R_u^2 + 3(v(s,a))^2. \quad \text{(D.12)}$$

Next we characterize $\|\nabla_\alpha u_{\alpha(t)}(s,a) - \nabla_\alpha u^0_{\alpha(t)}(s,a)\|_2$ in (ii). Recall that

$$\nabla_\alpha u_\alpha(s,a) = 1/\sqrt{m} \cdot \left(b_1 \cdot \mathbb{1}\big\{[\alpha]_1^\top(s,a) > 0\big\} \cdot (s,a)^\top, \ldots, b_m \cdot \mathbb{1}\big\{[\alpha]_m^\top(s,a) > 0\big\} \cdot (s,a)^\top\right)^\top,$$

and

$$\nabla_\alpha u^0_\alpha(s,a) = 1/\sqrt{m} \cdot \left(b_1 \cdot \mathbb{1}\big\{[\alpha(0)]_1^\top(s,a) > 0\big\} \cdot (s,a)^\top, \ldots, b_m \cdot \mathbb{1}\big\{[\alpha(0)]_m^\top(s,a) > 0\big\} \cdot (s,a)^\top\right)^\top.$$

We have

$$\|\nabla_\alpha u_{\alpha(t)}(s,a) - \nabla_\alpha u^0_{\alpha(t)}(s,a)\|_2^2 = \frac{1}{m}\sum_{i=1}^m \big(\mathbb{1}\big\{[\alpha(t)]_i^\top(s,a) > 0\big\} - \mathbb{1}\big\{[\alpha(0)]_i^\top(s,a) > 0\big\}\big)^2 \cdot \|(s,a)\|_2^2$$

$$\leq \frac{1}{m}\sum_{i=1}^m \mathbb{1}\big\{[\alpha(0)]_i^\top(s,a) \leq \|[\alpha(t)]_i - [\alpha(0)]_i\|_2\big\}, \quad \text{(D.13)}$$

where the inequality follows from the same arguments used to derive (D.6). Plugging (D.12) and (D.13) into (ii) and recalling that

$$\mathbb{E}_\rho[(v(s,a))^2] \leq \overline{v}_1 \cdot \mathbb{E}_\rho[(u_{\alpha(0)}(s,a))^2] + \overline{v}_2 \cdot R_u^2 + \overline{v}_3,$$

we find that it remains to upper bound the following two terms

$$\mathbb{E}_{\text{init},\rho}\left[\frac{1}{m}\sum_{i=1}^m \mathbb{1}\big\{[\alpha(0)]_i^\top(s,a) \leq \|[\alpha(t)]_i - [\alpha(0)]_i\|_2\big\}\right], \quad \text{(D.14)}$$

and

$$\mathbb{E}_{\text{init}}\left[\mathbb{E}_\rho[(u^0_{\alpha(0)}(s,a))^2] \cdot \mathbb{E}_\rho\left[\frac{1}{m}\sum_{i=1}^m \mathbb{1}\big\{[\alpha(0)]_i^\top(s,a) \leq \|[\alpha(t)]_i - [\alpha(0)]_i\|_2\big\}\right]\right]. \quad \text{(D.15)}$$

We already show in the proof of Lemma D.1 that (D.14) is $O(R_u m^{-1/2})$. We characterize (D.15) in the following. For the random initialization of $u_\alpha(s,a)$ in (3.2), we have

$$\mathbb{E}_\rho[(u^0_{\alpha(0)}(s,a))^2] = \frac{1}{m} \cdot \mathbb{E}_\rho\left[\sum_{i=1}^m \sigma([\alpha(0)]_i^\top(s,a))^2 + \sum_{1\leq i\neq j\leq m} b_i b_j \cdot \sigma([\alpha(0)]_i^\top(s,a)) \cdot \sigma([\alpha(0)]_j^\top(s,a))\right],$$

plugging which into (D.15) gives

$$\mathbb{E}_{\text{init}}\left[\mathbb{E}_\rho[(u^0_{\alpha(0)}(s,a))^2] \cdot \mathbb{E}_\rho\left[\frac{1}{m}\sum_{i=1}^m \mathbb{1}\big\{[\alpha(0)]_i^\top(s,a) \leq \|\alpha(t) - \alpha(0)\|_2\big\}\right]\right]$$

$$\leq \mathbb{E}_{\text{init}}\left[\frac{1}{m} \cdot \mathbb{E}_\rho\left[\sum_{i=1}^m \sigma([\alpha(0)]_i^\top(s,a))^2 + \sum_{1\leq i\neq j\leq m} b_i b_j \cdot \sigma([\alpha(0)]_i^\top(s,a)) \cdot \sigma([\alpha(0)]_j^\top(s,a))\right]\right.$$

$$\left. \cdot \frac{c}{m} \cdot \left(\sum_{i=1}^m \|[\alpha(t)]_i - [\alpha(0)]_i\|_2^2\right)^{1/2} \cdot \left(\sum_{i=1}^m \frac{1}{\|[\alpha(0)]_i\|_2^2}\right)^{1/2}\right],$$

where we use the same arguments applied to (D.8) in the proof of Lemma D.1. Note that $b_i, b_j$ are independent of $\alpha(0)$, $\mathbb{E}_{\text{init}}[b_i b_j] = 0$, and $\sum_{i=1}^m \|[\alpha(t)]_i - [\alpha(0)]_i\|_2^2 = \|\alpha(t) - \alpha(0)\|_2^2 \leq R_u^2$. We further obtain

$$\mathbb{E}_{\text{init}}\left[\mathbb{E}_\rho[(u^0_{\alpha(0)}(s,a))^2] \cdot \mathbb{E}_\rho\left[\frac{1}{m}\sum_{i=1}^m \mathbb{1}\big\{[\alpha(0)]_i^\top(s,a) \leq \|[\alpha(t)]_i - [\alpha(0)]_i\|_2\big\}\right]\right]$$

$$\leq \frac{cR_u}{m^2} \cdot \mathbb{E}_{\text{init}}\left[\mathbb{E}_\rho\left[\sum_{i=1}^m \sigma\big([\alpha(0)]_i^\top(s,a)\big)^2\right] \cdot \left(\sum_{i=1}^m \frac{1}{\|[\alpha(0)]_i\|_2^2}\right)^{1/2}\right]$$

$$\leq \frac{cR_u}{m^2} \cdot \mathbb{E}_{\text{init}}\left[\left(\sum_{i=1}^m \|[\alpha(0)]_i\|_2^2\right) \cdot \left(\sum_{i=1}^m \frac{1}{\|[\alpha(0)]_i\|_2^2}\right)^{1/2}\right].$$

Finally, by the Cauchy-Schwarz inequality, we have

$$\mathbb{E}_{\text{init}}\left[\left(\sum_{i=1}^m \|[\alpha(0)]_i\|_2^2\right) \cdot \left(\sum_{i=1}^m \frac{1}{\|[\alpha(0)]_i\|_2^2}\right)^{1/2}\right]$$

$$\leq \mathbb{E}_{\text{init}}\left[\left(\sum_{i=1}^m \|[\alpha(0)]_i\|_2^2\right)^2\right]^{1/2} \cdot \mathbb{E}_{\text{init}}\left[\sum_{i=1}^m \frac{1}{\|[\alpha(0)]_i\|_2^2}\right]^{1/2},$$

whose right-hand side is $O(m^{3/2})$. Thus, we obtain that (D.15) is $O(R_u m^{-1/2})$ and (ii) in (D.11) is $O(R_u^3 m^{-1/2})$, which concludes the proof of Lemma D.2. $\qquad\square$

## D.2 Global Convergence

In this section, we establish the global convergence of the meta-algorithm defined in (D.1) and (D.2). We first present the following lemma for characterizing the variance of the stochastic update vector $g_{\alpha(t)}(s, a, s', a')$ defined in (D.3), which later allows us to focus on tracking its mean in the global convergence analysis.

**Lemma D.3** (Variance of the Stochastic Update Vector). *There exists a constant $\xi_g^2 = O(R_u^2)$ independent of $t$, such that for any $t \leq T$, it holds that*
$$\mathbb{E}_{\text{init},\rho}[\|g_{\alpha(t)}(s, a, s', a') - \bar{g}_{\alpha(t)}\|_2^2] \leq \xi_g^2.$$

*Proof.* Since we have
$$\mathbb{E}_{\text{init},\rho}[\|g_{\alpha(t)}(s, a, s', a') - \bar{g}_{\alpha(t)}\|_2^2] = \mathbb{E}_{\text{init}}[\mathbb{E}_\rho[\|g_{\alpha(t)}(s, a, s', a') - \bar{g}_{\alpha(t)}\|_2^2]]$$
$$\leq \mathbb{E}_{\text{init}}[\mathbb{E}_\rho[\|g_{\alpha(t)}(s, a, s', a')\|_2^2]] = \mathbb{E}_{\text{init},\rho}[\|g_{\alpha(t)}(s, a, s', a')\|_2^2],$$
it suffices to prove that $\mathbb{E}[\|g_{\alpha(t)}(s, a, s', a')\|_2^2] = O(R_u^2)$. By the definition of $\mathbb{E}_\rho[\|g_{\alpha(t)}(s, a, s', a')\|_2^2]$ in (D.3), using $\|\nabla_{\alpha(t)} u_{\alpha(t)}(s, a)\|_2^2 \leq 1$, we obtain
$$\mathbb{E}_\rho[\|g_{\alpha(t)}(s, a, s', a')\|_2^2] = \mathbb{E}_\rho[\|\delta_{\alpha(t)}(s, a, s', s') \cdot \nabla_\alpha u_{\alpha(t)}(s, a)\|_2^2]$$
$$\leq \mathbb{E}_\rho[|\delta_{\alpha(t)}(s, a, s', s')|^2]. \tag{D.16}$$
Then, by similar arguments used in the derivation of (D.12), we obtain
$$\mathbb{E}_{\text{init},\rho}[|\delta_{\alpha(t)}(s, a, s', s')|^2] \leq 6\mathbb{E}_{\text{init},\rho}[(u_{\alpha(0)}(s, a))^2] + 6R_u^2 + 3\mathbb{E}_{\text{init},\rho}[(v(s, a))^2]$$
$$\leq (6 + 3\overline{v}_1) \cdot \mathbb{E}_{\text{init},\rho}[(u_{\alpha(0)}(s, a))^2] + (6 + \overline{v}_2)R_u^2 + 3\overline{v_3}^2. \tag{D.17}$$
Note that by $\|(s, a)\|_2 \leq 1$, we have
$$\mathbb{E}_{\text{init},\rho}[(u_{\alpha(0)}(s, a))^2] = \mathbb{E}_{z \sim \mathcal{N}(0, I_d/d), \rho}[\sigma(z^\top(s, a))^2] \leq \mathbb{E}_{z \sim \mathcal{N}(0, I_d/d)}[\|z\|_2^2] = 1,$$
which together with (D.16) and (D.17) implies $\mathbb{E}_{\text{init},\rho}[\|g_{\alpha(t)}(s, a, s', a')\|_2^2] = O(R_u^2)$. Thus, we complete the proof of Lemma D.3. $\qquad\square$

Before presenting the global convergence result of the meta-algorithm defined in (D.1), we first define $u_{\alpha^*}^0$, which later become the exact learning target of the meta-algorithm defined in (D.1) and (D.2). In specific, we define the approximate stationary point as $\alpha^* \in \mathcal{B}^0(R_u)$ such that
$$\alpha^* = \Pi_{\mathcal{B}^0(R_u)}(\alpha^* - \eta \cdot \bar{g}_{\alpha^*}^0), \tag{D.18}$$
which is equivalent to the condition
$$\langle \bar{g}_{\alpha^*}^0, \alpha - \alpha^* \rangle \geq 0, \quad \text{for any } \alpha \in \mathcal{B}^0(R_u). \tag{D.19}$$
Then we establish the uniqueness and existence of $u_{\alpha^*}^0$ with $\alpha^*$ defined in D.18. We first define the operator
$$\mathcal{T}u(s, a) = \mathbb{E}[v(s, a) + \mu \cdot u(s', a') \,|\, s' \sim \mathcal{P}(\cdot \,|\, s, a), a \sim \pi(\cdot \,|\, s')]. \tag{D.20}$$
Then using the definition of $\mathcal{T}$ in (D.20) and plugging the definition of $\bar{g}_{\alpha^*}^0$ in (D.4) into (D.19), we obtain
$$\langle u_{\alpha^*}^0 - \mathcal{T}u_{\alpha^*}^0, u_\alpha^0 - u_{\alpha^*}^0 \rangle_\rho \geq 0, \quad \text{for any } u_\alpha^0 \in \mathcal{F}_{B,m},$$
which is equivalent to $u_{\alpha^*}^0 = \Pi_{\mathcal{F}_{B,m}} \mathcal{T}u_{\alpha^*}^0$. Here the projection $\Pi_{\mathcal{F}_{B,m}}$ is defined with respect to the $\ell_2$-distance under measure $\rho$. Finally, as we have the following contraction inequality
$$\mathbb{E}_\rho[(\Pi_{\mathcal{F}_{B,m}} \mathcal{T}u_\alpha^0(s, a) - \Pi_{\mathcal{F}_{B,m}} \mathcal{T}u_{\alpha'}^0(s, a))^2]$$
$$\leq \mathbb{E}_\rho[(\mathcal{T}u_\alpha^0(s, a) - \mathcal{T}u_{\alpha'}^0(s, a))^2]$$
$$= \mu^2 \cdot \mathbb{E}_\rho[(\mathbb{E}[u_\alpha^0(s', a') \,|\, s' \sim \mathcal{P}(\cdot \,|\, s, a), a' \sim \pi(\cdot \,|\, s')] - \mathbb{E}[u_{\alpha'}^0(s', a') \,|\, s' \sim \mathcal{P}(\cdot \,|\, s, a), a' \sim \pi(\cdot \,|\, s')])^2]$$
$$\leq \mu^2 \cdot \mathbb{E}_\rho[(u_\alpha^0(s, a) - u_{\alpha'}^0(s, a))^2],$$
we know that such fixed-point solution $u_{\alpha^*}^0$ uniquely exists.

Now, with a well-defined learning target $u_{\alpha^*}^0$, we are ready to prove the the global convergence of the meta-algorithm defined in (D.1) and (D.2) with two-layer neural network approximation.

**Theorem D.4.** *Suppose that we run $T \geq 64/(1 - \mu)^2$ iterations of the meta-algorithm defined in (D.1) and (D.2). Setting the stepsize $\eta = T^{-1/2}$, we have*
$$\mathbb{E}_{\text{init},\rho}[(u_{\overline{\alpha}}(s, a) - u_{\alpha^*}^0(s, a))^2] = O(R_u^2 T^{-1/2} + R_u^{5/2} m^{-1/4} + R_u^3 m^{-1/2}),$$
*where $\overline{\alpha} = 1/T \cdot \sum_{t=0}^{T-1} \alpha(t)$ and $\alpha^*$ is the approximate stationary point defined in (D.18).*

*Proof.* The proof of the theorem consists of two parts. We first analyze the progress of each step. Then based on such one-step analysis, we establish the error bound of the approximation via two-layer neural network $u_\alpha$.

**One-Step Analysis:** For any $t < T$, using the stationarity condition in (D.18) and the convexity of $\mathcal{B}^0(R_u)$, we obtain

$$\mathbb{E}_\rho[\|\alpha(t+1) - \alpha^*\|_2^2 \,|\, \alpha(t)] \tag{D.21}$$

$$= \mathbb{E}_\rho\big[\big\|\Pi_{\mathcal{B}^0(R_u)}(\alpha(t) - \eta \cdot g_{\alpha(t)}(s, a, s', a')) - \Pi_{\mathcal{B}^0(R_u)}(\alpha^* - \eta \bar{g}^0_{\alpha^*})\big\|_2^2 \,|\, \alpha(t)\big]$$

$$\leq \mathbb{E}_\rho\big[\big\|(\alpha(t) - \alpha^*) - \eta \cdot (g_{\alpha(t)}(s, a, s', a') - \bar{g}^0_{\alpha^*})\big\|_2^2 \,|\, \alpha(t)\big]$$

$$= \|\alpha(t) - \alpha^*\|_2^2 - 2\eta \cdot \langle \bar{g}_{\alpha(t)} - \bar{g}^0_{\alpha^*}, \alpha(t) - \alpha^* \rangle + \eta^2 \cdot \mathbb{E}_\rho[\|g_{\alpha(t)}(s, a, s', a') - \bar{g}^0_{\alpha^*}\|_2^2 \,|\, \alpha(t)].$$

In the following, we upper bound the last two terms in (D.21). First, to upper bound $\mathbb{E}_\rho[\|g_{\alpha(t)}(s, a, s', a') - \bar{g}^0_{\alpha^*}\|_2^2 \,|\, \alpha(t)]$, by the Cauchy-Schwarz inequality we have

$$\mathbb{E}_\rho[\|g_{\alpha(t)}(s, a, s', a') - \bar{g}^0_{\alpha^*}\|_2^2 \,|\, \alpha(t)]$$

$$\leq 2\mathbb{E}_\rho[\|g_{\alpha(t)}(s, a, s', a') - \bar{g}_{\alpha(t)}\|_2^2 \,|\, \alpha(t)] + 2\|\bar{g}_{\alpha(t)} - \bar{g}^0_{\alpha^*}\|_2^2$$

$$\leq 2\mathbb{E}_\rho[\|g_{\alpha(t)}(s, a, s', a') - \bar{g}_{\alpha(t)}\|_2^2 \,|\, \alpha(t)] + 4\|\bar{g}_{\alpha(t)} - \bar{g}^0_{\alpha(t)}\|_2^2 + 4\|\bar{g}^0_{\alpha(t)} - \bar{g}^0_{\alpha^*}\|_2^2, \tag{D.22}$$

where the total expectation on the first two terms on the right-hand side are characterized in Lemmas D.3 and D.2, respectively. To characterize $\|\bar{g}^0_{\alpha(t)} - \bar{g}^0_{\alpha^*}\|_2^2$, again using $\|(s, a)\|_2 \leq 1$, we have

$$\|\bar{g}^0_{\alpha(t)} - \bar{g}^0_{\alpha^*}\|_2^2 = \mathbb{E}_\rho\big[(\delta_{\alpha(t)}(s, a, s', a') - \delta_{\alpha^*}(s, a, s', a'))^2 \cdot \|\nabla_\alpha u^0_{\alpha(t)}(s, a)\|_2^2\big]$$

$$\leq \mathbb{E}_\rho\big[\big((u^0_{\alpha(t)}(s, a) - u^0_{\alpha^*}(s, a)) - \mu \cdot (u^0_{\alpha(t)}(s', a') - u^0_{\alpha^*}(s', a'))\big)^2\big]. \tag{D.23}$$

For the right-hand side of (D.23), we use the Cauchy-Schwarz inequality on the interaction term and obtain

$$\mathbb{E}_\rho\big[(u^0_{\alpha(t)}(s', a') - u^0_{\alpha^*}(s', a')) \cdot (u^0_{\alpha(t)}(s, a) - u^0_{\alpha^*}(s, a))\big]$$

$$\leq \mathbb{E}_\rho[(u^0_{\alpha(t)}(s', a') - u^0_{\alpha^*}(s', a'))^2]^{1/2} \cdot \mathbb{E}_\rho[(u^0_{\alpha(t)}(s, a) - u^0_{\alpha^*}(s, a))^2]^{1/2}$$

$$= \mathbb{E}_\rho[(u^0_{\alpha(t)}(s, a) - u^0_{\alpha^*}(s, a))^2], \tag{D.24}$$

where in the last line we use the fact that $(s, a)$ and $(s', a')$ have the same marginal distribution. Thus, we obtain

$$\|\bar{g}^0_{\alpha(t)} - \bar{g}^0_{\alpha^*}\|_2^2 \leq 4\mathbb{E}_\rho[(u^0_{\alpha(t)}(s, a) - u^0_{\alpha^*}(s, a))^2]. \tag{D.25}$$

Next, to upper bound $\langle \bar{g}_{\alpha(t)} - \bar{g}^0_{\alpha^*}, \alpha(t) - \alpha^* \rangle$, we use the Hölder's inequality to obtain

$$\langle \bar{g}_{\alpha(t)} - \bar{g}^0_{\alpha^*}, \alpha(t) - \alpha^* \rangle = \langle \bar{g}_{\alpha(t)} - \bar{g}^0_{\alpha(t)}, \alpha(t) - \alpha^* \rangle + \langle \bar{g}^0_{\alpha(t)} - \bar{g}^0_{\alpha^*}, \alpha(t) - \alpha^* \rangle$$

$$\geq -\|\bar{g}_{\alpha(t)} - \bar{g}^0_{\alpha(t)}\|_2 \cdot \|\alpha(t) - \alpha^*\|_2 + \langle \bar{g}^0_{\alpha(t)} - \bar{g}^0_{\alpha^*}, \alpha(t) - \alpha^* \rangle$$

$$\geq -R_u\|\bar{g}_{\alpha(t)} - \bar{g}^0_{\alpha(t)}\|_2 + \langle \bar{g}^0_{\alpha(t)} - \bar{g}^0_{\alpha^*}, \alpha(t) - \alpha^* \rangle, \tag{D.26}$$

where the second inequality follows from $\|\alpha(t) - \alpha^*\|_2 \leq R_u$. For the term $\langle \bar{g}^0_{\alpha(t)} - \bar{g}^0_{\alpha^*}, \alpha(t) - \alpha^* \rangle$ on the right-hand side of (D.26), we have

$$\langle \bar{g}^0_{\alpha(t)} - \bar{g}^0_{\alpha^*}, \alpha(t) - \alpha^* \rangle$$

$$= \mathbb{E}_\rho\Big[\big((u^0_{\alpha(t)}(s, a) - u^0_{\alpha^*}(s, a)) - \mu \cdot (u^0_{\alpha(t)}(s', a') - u^0_{\alpha^*}(s', a'))\big) \cdot \langle \nabla_\alpha u^0_{\alpha(t)}(s, a), \alpha(t) - \alpha^* \rangle\Big]$$

$$= \mathbb{E}_\rho\Big[\big((u^0_{\alpha(t)}(s, a) - u^0_{\alpha^*}(s, a)) - \mu \cdot (u^0_{\alpha(t)}(s', a') - u^0_{\alpha^*}(s', a'))\big) \cdot (u^0_{\alpha(t)}(s, a) - u^0_{\alpha^*}(s, a))\Big]$$

$$\geq \mathbb{E}_\rho[(u^0_{\alpha(t)}(s, a) - u^0_{\alpha^*}(s, a))^2] - \mu \cdot \mathbb{E}_\rho[(u^0_{\alpha(t)}(s, a) - u^0_{\alpha^*}(s, a))]^2$$

$$\geq (1 - \mu) \cdot \mathbb{E}_\rho[(u^0_{\alpha(t)}(s, a) - u^0_{\alpha^*}(s, a))^2], \tag{D.27}$$

where the second equality and the first inequality follow from (D.5) and (D.24), respectively.

Therefore, combining (D.21) with (D.22), (E.4), (D.26), and (D.27), we obtain

$$\mathbb{E}_\rho[\|\alpha(t+1) - \alpha^*\|_2^2 \,|\, \alpha(t)]$$

$$\leq \|\alpha(t) - \alpha^*\|_2^2 - \big(2\eta(1 - \gamma) - 8\eta^2\big) \cdot \mathbb{E}_\rho[(u^0_{\alpha(t)}(s, a) - u^0_{\alpha^*}(s, a))^2 \,|\, \alpha(t)] \tag{D.28}$$

$$+ 2\eta^2\|\bar{g}_{\alpha(t)} - \bar{g}^0_{\alpha(t)}\|_2^2 + 2\eta R_u\|\bar{g}_{\alpha(t)} - \bar{g}^0_{\alpha(t)}\|_2 + \eta^2 \cdot \mathbb{E}_\rho[\|g_{\alpha(t)}(s, a, s', a') - \bar{g}_{\alpha(t)}\|_2^2 \,|\, \alpha(t)].$$

**Error Bound:** Rearranging (D.28), we obtain

$$\mathbb{E}_\rho[(u_{\alpha(t)}(s,a) - u^0_{\alpha^*}(s,a))^2 \,|\, \alpha(t)]$$

$$\leq \mathbb{E}_\rho\big[2(u_{\alpha(t)}(s,a) - u^0_{\alpha(t)}(s,a))^2 + 2(u^0_{\alpha(t)}(s,a) - u^0_{\alpha^*}(s,a))^2 \,|\, \alpha(t)\big]$$

$$\leq \big(\eta(1-\gamma) - 4\eta^2\big)^{-1} \cdot \big(\|\alpha(t) - \alpha^*\|_2^2 - \mathbb{E}_\rho[\|\alpha(t+1) - \alpha^*\|_2^2 \,|\, \alpha(t)] + \xi_\alpha^2 \eta^2\big) \qquad \text{(D.29)}$$
$$+ O(R_u^{5/2} m^{-1/4} + R_u^3 m^{-1/2}).$$

Taking total expectation on both sides of (D.29) and telescoping for $t+1 \in [T]$, we further obtain

$$\mathbb{E}_{\text{init},\rho}[(u_{\overline{\alpha}}(s,a) - u^0_{\alpha^*}(s,a))^2] \leq \frac{1}{T} \sum_{t=0}^{T-1} \mathbb{E}_{\text{init},\rho}[(u_{\alpha(t)}(s,a) - u_{\alpha^*}(s,a))^2] \qquad \text{(D.30)}$$

$$\leq T^{-1} \cdot \big(\eta(1-\gamma) - 4\eta^2\big)^{-1} \cdot \big(\mathbb{E}_{\text{init}}[\|\alpha(0) - \alpha^*\|_2^2] + T\xi_\alpha^2 \eta^2\big)$$
$$+ O(R_u^{5/2} m^{-1/4} + R_u^3 m^{-1/2}).$$

Let $T \geq 64/(1-\mu)^2$ and $\eta = T^{-1/2}$, it holds that $T^{-1/2} \cdot (\eta(1-\gamma) - 4\eta^2)^{-1} \leq 16(1-\gamma)^{-1/2}$ and $T\eta^2 \leq 1$, which together with (D.30) implies

$$\mathbb{E}_{\text{init},\rho}[(u_{\alpha(t)}(s,a) - u^0_{\alpha^*}(s,a))^2 \,|\, \alpha(t)]$$

$$\leq \frac{16}{(1-\mu)^2\sqrt{T}} \cdot \big(\mathbb{E}_{\text{init}}[\|\alpha(0) - \alpha^*\|_2^2] + \xi_\alpha^2\big) + O(R_u^{5/2} m^{-1/4} + R_u^3 m^{-1/2})$$

$$\leq \frac{16(R_\alpha^2 + \xi_\alpha^2)}{(1-\mu)^2\sqrt{T}} + O(R_u^{5/2} m^{-1/4} + R_u^3 m^{-1/2}) = O(R_u^2 T^{-1/2} + R_u^{5/2} m^{-1/4} + R_u^3 m^{-1/2}),$$

where in the second inequality we use $\|\alpha(0) - \alpha^*\|_2 \leq R_u$ and in the equality we use Lemma D.3. Thus, we conclude the proof of Theorem D.4. $\qquad\square$

Following the definition of $u^0_\alpha$ in (D.4), we define the local linearization of $Q_\omega$ at the initialization as

$$Q^0_\omega(s,a) = \frac{1}{\sqrt{m_Q}} \sum_{i=1}^{m_Q} b_i \cdot \mathbb{1}\big\{[\omega(0)]_i^\top(s,a) \geq 0\big\} \cdot [\omega]_i^\top(s,a).$$

Similarly, for $f_\theta$ we define

$$f^0_\theta(s,a) = \frac{1}{\sqrt{m_f}} \sum_{i=1}^{m_f} b_i \cdot \mathbb{1}\big\{[\theta(0)]_i^\top(s,a) \geq 0\big\} \cdot [\theta]_i^\top(s,a).$$

In the sequel, we show that Theorem D.4 implies both Theorems 4.5 and 4.6.

To obtain Theorem 4.5, we set $\rho = \widetilde{\sigma}_k$, $u_\alpha = f_\theta$, $v = \tau_{k+1} \cdot (\beta_k^{-1} Q_{\omega_k} + \tau_k^{-1} f_{\theta_k})$, $\mu = 0$, and $R_u = R_f$. Using $\tau_{k+1}, \tau_k$, and $\beta_k$ specified in Algorithm 1, we have

$$\mathbb{E}_{\widetilde{\sigma}_k}[(v(s,a))^2] \leq 2\tau_{k+1}^2 \cdot \big(\beta_k^{-2} \cdot \mathbb{E}_{\widetilde{\sigma}_k}[(Q_{\omega_k}(s,a))^2] + \tau_k^{-2} \cdot \mathbb{E}_{\widetilde{\sigma}_k}[(f_{\theta_k}(s,a))^2]\big)$$

$$\leq 4\mathbb{E}_{\widetilde{\sigma}_k}[(f_{\theta(0)}(s,a))^2] + 4R_f^2,$$

where in the second inequality we use $\tau_{k+1}^2 \beta_k^{-2} + \tau_{k+1}^2 \tau_k^{-2} \leq 1$ and the fact that $(Q_{\omega_k}(s,a))^2 \leq 2(Q_{\omega(0)}(s,a))^2 + 2R_Q^2$ and $(f_{\theta_k}(s,a))^2 \leq 2(f_{\theta(0)}(s,a))^2 + 2R_f^2$, which is a consequence of the 1-Lipschitz continuity of the neural network with respect to the weights. Also note that $Q_{\omega(0)}(s,a) = f_{\theta(0)}(s,a)$ due to the fact that $Q_{\omega_k}$ and $f_{\theta_k}$ share the same initialization. Thus, we have $\overline{v}_1 = 4$, $\overline{v}_2 = 4$, and $\overline{v}_3 = 0$. Moreover, by $f^0_{\theta^*} = \Pi_{\mathcal{F}_{R_f,m_f}} \mathcal{T} f^0_{\theta^*} = \Pi_{\mathcal{F}_{R_f,m}}(\tau_{k+1} \cdot (\beta_k^{-1} Q_{\omega_k} + \tau_k^{-1} f_{\theta_k}))$, we have

$$f^0_{\theta^*} = \operatorname*{argmin}_{f \in \mathcal{F}_{R_f,m_f}} \big\{\big\|f - \tau_{k+1} \cdot (\beta_k^{-1} Q_{\omega_k} + \tau_k^{-1} f_{\theta_k})\big\|_{2,\widetilde{\sigma}_k}\big\},$$

which together with the fact that $\tau_{k+1} \cdot (\beta_k^{-1} Q^0_{\omega_k}(s,a) + \tau_k^{-1} f^0_{\theta_k}(s,a)) \in \mathcal{F}_{R_f,m_f}$ implies

$$\mathbb{E}_{\text{init},\widetilde{\sigma}_k}\big[\big(f^0_{\theta^*}(s,a) - \tau_{k+1} \cdot (\beta_k^{-1} Q_{\omega_k}(s,a) + \tau_k^{-1} f_{\theta_k}(s,a))\big)^2\big]$$

$$\leq \mathbb{E}_{\text{init},\widetilde{\sigma}_k}\big[\big(\tau_{k+1} \cdot (\beta_k^{-1} Q^0_{\omega_k}(s,a) + \tau_k^{-1} f^0_{\theta_k}(s,a)) - \tau_{k+1} \cdot (\beta_k^{-1} Q_{\omega_k}(s,a) + \tau_k^{-1} f_{\theta_k}(s,a))\big)^2\big]$$

$$\leq \tau_{k+1}^2 \beta_k^{-2} \cdot \mathbb{E}_{\text{init},\widetilde{\sigma}_k}[(Q^0_{\omega_k}(s,a) - Q_{\omega_k}(s,a))^2] + \tau_{k+1}^2 \tau_k^{-2} \cdot \mathbb{E}_{\text{init},\widetilde{\sigma}_k}[(f^0_{\theta_k}(s,a) - f_{\theta_k}(s,a))^2]$$

$$= O(R_f^3 m_f^{-1/2}). \qquad \text{(D.31)}$$

Finally, plugging (D.31) into Theorem D.4 for $f_\theta$, we obtain

$$\mathbb{E}_{\text{init},\widetilde{\sigma}_k}\left[\left(f_{\overline{\theta}}(s,a) - \tau_{k+1}\cdot(\beta_k^{-1}Q_{\omega_k}(s,a) + \tau_k^{-1}f_{\theta_k}(s,a))\right)^2\right]$$

$$\leq 2\mathbb{E}_{\text{init},\widetilde{\sigma}_k}[(f_{\overline{\theta}}(s,a) - f_{\theta^*}^0(s,a))^2] + 2\mathbb{E}_{\text{init},\widetilde{\sigma}_k}\left[\left(f_{\theta^*}^0(s,a) - \tau_{k+1}\cdot(\beta_k^{-1}Q_{\omega_k}(s,a) + \tau_k^{-1}f_{\theta_k}(s,a))\right)^2\right]$$

$$= O(R_f^2 T^{-1/2} + R_f^{5/2}m_f^{-1/4} + R_f^3 m_f^{-1/2})$$

which gives Theorem 4.5.

To obtain Theorem 4.6, we set $\rho = \sigma_k$, $u_\alpha = Q_\omega$, $v = (1-\gamma)\cdot r$, $\mu = \gamma$ and $R_u = R_Q$. Correspondingly, we have $\overline{v}_1 = 0, \overline{v}_2 = 0, \overline{v}_3 = R_{\max}^2$ and $u_{\alpha^*}^0 = Q_{\omega^*}^0$. Moreover, by the definition of the operator $\mathcal{T}$ in (D.20), we have $\mathcal{T} = \mathcal{T}^{\pi_{\theta_k}}$, which implies $Q^{\pi_{\theta_k}} = \mathcal{T}Q^{\pi_{\theta_k}}$. Meanwhile, by Assumption 4.3, we have $Q^{\pi_{\theta_k}} \in \mathcal{F}_{R_Q,m_Q}$, which implies $Q^{\pi_{\theta_k}} = \Pi_{\mathcal{F}_{R_Q,m_Q}} Q^{\pi_{\theta_k}} = \Pi_{\mathcal{F}_{R_Q,m_Q}} \mathcal{T}Q^{\pi_{\theta_k}}$. Since we already show that $Q_{\omega^*}^0$ is the unique solution to the equation $Q = \Pi_{\mathcal{F}_{R_Q,m_Q}} \mathcal{T}Q$, we obtain $Q_{\alpha^*}^0 = Q^{\pi_{\theta_k}}$. Therefore, we can substitute $Q_{\alpha^*}^0$ with $Q^{\pi_{\theta_k}}$ in Theorem D.4 to obtain Theorem 4.6.

# E   Proofs for Section 4.2

*Proof of Lemma 4.7.* We first have

$$\pi_{k+1}(a\,|\,s) = \exp\{\beta_k^{-1}Q^{\pi_{\theta_k}}(s,a) + \tau_k^{-1}f_{\theta_k}(s,a)\}/Z_{k+1}(s),$$

and

$$\pi_{\theta_{k+1}}(a\,|\,s) = \exp\{\tau_{k+1}^{-1}f_{\theta_{k+1}}(s,a)\}/Z_{\theta_{k+1}}(s).$$

Here $Z_{k+1}(s), Z_{\theta_{k+1}}(s) \in \mathbb{R}$ are normalization factors, which are defined as

$$Z_{k+1}(s) = \sum_{a'\in\mathcal{A}} \exp\{\beta_k^{-1}Q^{\pi_{\theta_k}}(s,a') + \tau_k^{-1}f_{\theta_k}(s,a')\},$$

$$Z_{\theta_{k+1}}(s) = \sum_{a'\in\mathcal{A}} \exp\{\tau_{k+1}^{-1}f_{\theta_{k+1}}(s,a')\}, \tag{E.1}$$

respectively. Thus, we reformulate the inner product in (4.5) as

$$\langle \log\pi_{\theta_{k+1}}(\cdot\,|\,s) - \log\pi_{k+1}(\cdot\,|\,s), \pi^*(\cdot\,|\,s) - \pi_{\theta_k}(\cdot\,|\,s)\rangle$$

$$= \langle \tau_{k+1}^{-1}f_{\theta_{k+1}}(s,\cdot) - (\beta_k^{-1}Q^{\pi_{\theta_k}}(s,\cdot) + \tau_k^{-1}f_{\theta_k}(s,\cdot)), \pi^*(\cdot\,|\,s) - \pi_{\theta_k}(\cdot\,|\,s)\rangle, \tag{E.2}$$

where we use the fact that

$$\langle \log Z_{k+1}(s) - \log Z_{\theta_{k+1}}(s), \pi^*(\cdot\,|\,s) - \pi_{\theta_k}(\cdot\,|\,s)\rangle$$

$$= (\log Z_{k+1}(s) - \log Z_{\theta_{k+1}}(s))\sum_{a'\in\mathcal{A}}(\pi^*(a'\,|\,s) - \pi_{\theta_k}(a'\,|\,s)) = 0.$$

Thus, it remains to upper bound the right-hand side of (E.2). We first decompose it to two terms, namely the error from learning the Q-function and the error from fitting the improved policy, that is,

$$\langle \tau_{k+1}^{-1}f_{\theta_{k+1}}(s,\cdot) - (\beta_k^{-1}Q^{\pi_{\theta_k}}(s,\cdot) + \tau_k^{-1}f_{\theta_k}(s,\cdot)), \pi^*(\cdot\,|\,s) - \pi_{\theta_k}(\cdot\,|\,s)\rangle$$

$$= \underbrace{\langle \tau_{k+1}^{-1}f_{\theta_{k+1}}(s,\cdot) - (\beta_k^{-1}Q_{\omega_k}(s,\cdot) + \tau_k^{-1}f_{\theta_k}(s,\cdot)), \pi^*(\cdot\,|\,s) - \pi_{\theta_k}(\cdot\,|\,s)\rangle}_{(i)}$$

$$+ \underbrace{\langle \beta_k^{-1}Q_{\omega_k}(s,\cdot) - \beta_k^{-1}Q^{\pi_{\theta_k}}(s,\cdot), \pi^*(\cdot\,|\,s) - \pi_{\theta_k}(\cdot\,|\,s)\rangle}_{(ii)}. \tag{E.3}$$

**Upper Bounding (i):** We have

$$\langle \tau_{k+1}^{-1}f_{\theta_{k+1}}(s,\cdot) - (\beta_k^{-1}Q_{\omega_k}(s,\cdot) + \tau_k^{-1}f_{\theta_k}(s,\cdot)), \pi^*(\cdot\,|\,s) - \pi_{\theta_k}(\cdot\,|\,s)\rangle \tag{E.4}$$

$$= \left\langle \tau_{k+1}^{-1}f_{\theta_{k+1}}(s,\cdot) - (\beta_k^{-1}Q_{\omega_k}(s,\cdot) + \tau_k^{-1}f_{\theta_k}(s,\cdot)), \pi_0(\cdot\,|\,s)\cdot\left(\frac{\pi^*(\cdot\,|\,s)}{\pi_0(\cdot\,|\,s)} - \frac{\pi_{\theta_k}(\cdot\,|\,s)}{\pi_0(\cdot\,|\,s)}\right)\right\rangle.$$

Taking expectation with respect to $s \sim \nu^*$ on the both sides of (E.4) and using the Cauchy-Schwarz inequality, we obatin

$$\left|\mathbb{E}_{\nu^*}\left[\langle \tau_{k+1}^{-1} f_{\theta_{k+1}}(s,\cdot) - (\beta_k^{-1} Q_{\omega_k}(s,\cdot) + \tau_k^{-1} f_{\theta_k}(s,\cdot)), \pi^*(\cdot \mid s) - \pi_{\theta_k}(\cdot \mid s)\rangle\right]\right|$$

$$= \left|\int_{\mathcal{S}} \left\langle \tau_{k+1}^{-1} f_{\theta_{k+1}}(s,\cdot) - (\beta_k^{-1} Q_{\omega_k}(s,\cdot) + \tau_k^{-1} f_{\theta_k}(s,\cdot)), \pi_0(\cdot \mid s) \cdot \nu_k(s) \cdot \left(\frac{\pi^*(\cdot \mid s)}{\pi_0(\cdot \mid s)} - \frac{\pi_{\theta_k}(\cdot \mid s)}{\pi_0(\cdot \mid s)}\right)\right\rangle \cdot \frac{\nu^*(s)}{\nu_k(s)} \mathrm{d}s\right|$$

$$= \left|\int_{\mathcal{S}\times\mathcal{A}} \left(\tau_{k+1}^{-1} f_{\theta_{k+1}}(s,a) - (\beta_k^{-1} Q_{\omega_k}(s,a) + \tau_k^{-1} f_{\theta_k}(s,a))\right) \cdot \left(\frac{\sigma^*(a \mid s)}{\widetilde{\sigma}_k(a \mid s)} - \frac{\pi_{\theta_k}(a \mid s) \cdot \nu^*(s)}{\widetilde{\sigma}_k(a \mid s)}\right) \mathrm{d}\widetilde{\sigma}_k(s,a)\right|$$

$$\leq \mathbb{E}_{\widetilde{\sigma}_k}\left[\left(\tau_{k+1}^{-1} f_{\theta_{k+1}}(s,a) - (\beta_k^{-1} Q_{\omega_k}(s,a) + \tau_k^{-1} f_{\theta_k}(s,a))\right)^2\right]^{1/2} \cdot \mathbb{E}_{\widetilde{\sigma}_k}\left[\left|\frac{\mathrm{d}\sigma^*}{\mathrm{d}\widetilde{\sigma}_k} - \frac{\mathrm{d}(\pi_{\theta_k}\nu^*)}{\mathrm{d}\widetilde{\sigma}_k}\right|^2\right]^{1/2}$$

$$\leq \tau_{k+1}^{-1}\epsilon_{k+1} \cdot \phi_k^*, \tag{E.5}$$

where in the last inequality we use the error bound in (4.3) and the definition of $\phi_k^*$ in (4.2).

**Upper Bounding (ii):** By the Cauchy-Schwartz inequality, we have

$$\left|\mathbb{E}_{\nu^*}[\langle \beta_k^{-1} Q_{\omega_k}(s,\cdot) - \beta_k^{-1} Q^{\pi_{\theta_k}}(s,\cdot), \pi^*(\cdot \mid s) - \pi_{\theta_k}(\cdot \mid s)\rangle]\right|$$

$$= \left|\int_{\mathcal{S}\times\mathcal{A}} (\beta_k^{-1} Q_{\omega_k}(s,a) - \beta_k^{-1} Q^{\pi_{\theta_k}}(s,a)) \cdot \left(\frac{\pi^*(a \mid s)}{\pi_{\theta_k}(a \mid s)} - \frac{\pi_{\theta_k}(a \mid s)}{\pi_{\theta_k}(a \mid s)}\right) \cdot \frac{\nu^*(s)}{\nu_k(s)} \mathrm{d}\sigma_k(s,a)\right|$$

$$\leq \mathbb{E}_{\sigma_k}[(\beta_k^{-1} Q_{\omega_k}(s,a) - \beta_k^{-1} Q^{\pi_{\theta_k}}(s,a))^2]^{1/2} \cdot \mathbb{E}_{\sigma_k}\left[\left|\frac{\mathrm{d}\sigma^*}{\mathrm{d}\sigma_k} - \frac{\mathrm{d}\nu^*}{\mathrm{d}\nu_k}\right|^2\right]^{1/2}$$

$$\leq \beta_k^{-1}\epsilon_k' \cdot \psi_k^*, \tag{E.6}$$

where in the last inequality we use the error bound in (4.4) and the definition of $\psi_k^*$ in (4.2). Finally, combining (E.2), (E.3), (E.5), and (E.6), we have

$$\left|\mathbb{E}_{\nu^*}[\langle \log \pi_{\theta_{k+1}}(\cdot \mid s) - \log \pi_{k+1}(\cdot \mid s), \pi^*(\cdot \mid s) - \pi_{\theta_k}(\cdot \mid s)\rangle]\right|$$

$$\leq \tau_{k+1}^{-1}\epsilon_{k+1} \cdot \phi_k^* + \beta_k^{-1}\epsilon_k' \cdot \psi_k^*,$$

which concludes the proof of Lemma 4.7. $\qquad\square$

*Proof of Lemma 4.8.* By the triangle inequality, we have

$$\|\tau_{k+1}^{-1} f_{\theta_{k+1}}(s,\cdot) - \tau_k^{-1} f_{\theta_k}(s,\cdot)\|_\infty^2$$

$$\leq 2\|\tau_{k+1}^{-1} f_{\theta_{k+1}}(s,\cdot) - \tau_k^{-1} f_{\theta_k}(s,\cdot) - \beta_k^{-1} Q_{\omega_k}(s,\cdot)\|_\infty^2 + 2\|\beta_k^{-1} Q_{\omega_k}(s,\cdot)\|_\infty^2. \tag{E.7}$$

For the first term on the right-hand side of (E.7), we have

$$\mathbb{E}_{\nu^*}[\|\tau_{k+1}^{-1} f_{\theta_{k+1}}(s,\cdot) - \tau_k^{-1} f_{\theta_k}(s,\cdot) - \beta_k^{-1} Q_{\omega_k}(s,\cdot)\|_\infty^2] \leq |\mathcal{A}| \cdot \tau_{k+1}^{-2}\epsilon_{k+1}^2. \tag{E.8}$$

For the second term on the right-hand side of (E.7), we have

$$\mathbb{E}_{\nu^*}[\|\beta_k^{-1} Q_{\omega_k}(s,\cdot)\|_\infty^2] \leq \beta_k^{-2} \cdot \mathbb{E}_{\nu^*}\left[\max_{a\in\mathcal{A}} 2(Q_{\omega_0}(s,a))^2 + 2R_f^2\right] = \beta_k^{-2}M, \tag{E.9}$$

where we use the 1-Lipschitz continuity of $Q_\omega$ in $\omega$ and the constraint $\|\omega_k - \omega_0\|_2 \leq R_\omega$. Then, taking expectation with respect to $s \sim \nu^*$ on the both sides of (E.7) and plugging in (E.8) and (E.9), we finish the proof of Lemma 4.8. $\qquad\square$

# F  Proof of Corollary 4.10

*Proof.* By Theorems 4.5 and 4.6, we have $\epsilon_{k+1} = O(R_f^2 T^{-1/2} + R_f^{5/2} m_f^{-1/4} + R_f^3 m_f^{-1/2})$ and $\epsilon_k' = O(R_Q^2 T^{-1/2} + R_Q^{5/2} m_Q^{-1/4} + R_Q^3 m_Q^{-1/2})$, which gives

$$\tau_{k+1}^{-1}\epsilon_{k+1} \cdot \phi_{k+1}^* = O\big(kK^{-1/2} \cdot \phi_k^* \cdot (R_f^2 T^{-1/2} + R_f^{5/2} m_f^{-1/4})\big),$$

$$|\mathcal{A}| \cdot \tau_{k+1}^{-2}\epsilon_{k+1}^2 = O\big(k^2 K^{-1} \cdot |\mathcal{A}| \cdot (R_f^2 T^{-1/2} + R_f^{5/2} m_f^{-1/4})^2\big),$$

$$\beta_k^{-1}\epsilon_k' \cdot \psi_k^* = O\big(K^{-1/2} \cdot \psi_k^* \cdot (R_Q^2 T^{-1/2} + R_Q^{5/2} m_Q^{-1/4})\big),$$

when $m_f = \Omega(R_f^2)$ and $m_Q = \Omega(R_Q^2)$.

Next, setting $m_f = R_f^{10} \cdot \Omega(K^6 \cdot \phi_k^{*\,4} + K^4 \cdot |\mathcal{A}|^2)$, $m_Q = \Omega(K^2 R_Q^{10} \cdot \psi_k^{*\,4})$ and $T = \Omega(K^3 R_f^4 \cdot \phi_k^{*\,2} + K R_Q^4 \cdot \psi_k^{*\,2})$, we further have

$$\varepsilon_k = \tau_{k+1}^{-1} \epsilon_{k+1} \cdot \phi_k^* + \beta_k^{-1} \epsilon_k' \cdot \psi_k^* = O(K^{-1}). \tag{F.1}$$

Meanwhile, setting $m_f = \Omega(K^4 R_f^{10} \cdot |\mathcal{A}|^2)$ and $T = \Omega(K^2 R_f^4 \cdot |\mathcal{A}|)$, we have

$$\varepsilon_k' = |\mathcal{A}| \cdot \tau_{k+1}^{-2} \epsilon_{k+1}^2 = O(K^{-1}). \tag{F.2}$$

Summing up (F.1) and (F.2) for $k + 1 \in [K]$ and plugging it into Theorem 4.9, we obtain

$$\min_{0 \le k \le K} \left\{ \mathcal{L}(\pi^*) - \mathcal{L}(\pi_{\theta_k}) \right\} \le \frac{\beta^2 \log |\mathcal{A}| + M + O(1)}{(1 - \gamma)\beta \cdot \sqrt{K}},$$

which completes the proof of Corollary 4.10. $\qquad\qquad\qquad\qquad\qquad\qquad\qquad\qquad\qquad\qquad\qquad\square$

# G  Proofs of Section 5

*Proof of Lemma 5.1.*  The proof follows that of Lemma 6.1 in [24]. By the definition of $V^\pi(s)$ in (2.1), we have

$$\mathbb{E}_{\nu^*}[V^{\pi^*}(s)] = \sum_{t=0}^{\infty} \gamma^t \cdot \mathbb{E}_{a_t \sim \pi^*(\cdot \,|\, s_t), s_t \sim (\mathcal{P}^{\pi^*})^t \nu^*} \left[ (1 - \gamma) \cdot r(s_t, a_t) \right] \tag{G.1}$$

$$= \sum_{t=0}^{\infty} \gamma^t \cdot \mathbb{E}_{a_t \sim \pi^*(\cdot \,|\, s_t), s_t \sim (\mathcal{P}^{\pi^*})^t \nu^*} \left[ (1 - \gamma) \cdot r(s_t, a_t) + V^\pi(s_t) - V^\pi(s_t) \right]$$

$$= \sum_{t=0}^{\infty} \gamma^t \cdot \mathbb{E}_{s_{t+1} \sim \mathcal{P}(\cdot \,|\, s_t, a_t), a_t \sim \pi^*(\cdot \,|\, s_t), s_t \sim (\mathcal{P}^{\pi^*})^t \nu^*} \left[ (1 - \gamma) \cdot r(s_t, a_t) + \gamma \cdot V^\pi(s_{t+1}) - V^\pi(s_t) \right]$$

$$+ \mathbb{E}_{\nu^*}[V^\pi(s)],$$

where the third inequality is obtained by taking $\mathbb{E}_{\nu^*}[V^\pi(s_0)] = \mathbb{E}_{\nu^*}[V^\pi(s)]$ out and, correspondingly, delaying $V^\pi(s_t)$ by one time step to $V^\pi(s_{t+1})$ in each term of the summation. Note that for the advantage function, by definition of the action-value function, we have

$$A^\pi(s, a) = Q^\pi(s, a) - V^\pi(s) = (1 - \gamma) \cdot r(s, a) + \gamma \cdot \mathbb{E}_{s' \sim \mathcal{P}(\cdot \,|\, s, a)}[V^\pi(s')] - V^\pi(s),$$

which together with (G.1) implies

$$\mathbb{E}_{\nu^*}[V^{\pi^*}(s)] = \sum_{t=0}^{\infty} \gamma^t \cdot \mathbb{E}_{a_t \sim \pi^*(\cdot \,|\, s_t), s_t \sim (\mathcal{P}^{\pi^*})^t \nu^*}[A^\pi(s_t, a_t)] + \mathbb{E}_{\nu^*}[V^\pi(s)]$$

$$= (1 - \gamma)^{-1} \cdot \mathbb{E}_{\sigma^*}[A^\pi(s, a)] + \mathbb{E}_{\nu^*}[V^\pi(s)]. \tag{G.2}$$

Here the second equality follows from $(\mathcal{P}^{\pi^*})^t \nu^* = \nu^*$ for any $t \ge 0$ and $\sigma^* = \pi^* \nu^*$. Finally, note that for any given $s \in \mathcal{S}$,

$$\mathbb{E}_{\pi^*}[A^\pi(s, a)] = \mathbb{E}_{\pi^*}[Q^\pi(s, a) - V^\pi(s)] = \langle Q^\pi(s, \cdot), \pi^*(\cdot \,|\, s) \rangle - \langle Q^\pi(s, \cdot), \pi(\cdot \,|\, s) \rangle$$

$$= \langle Q^\pi(s, \cdot), \pi^*(\cdot \,|\, s) - \pi(\cdot \,|\, s) \rangle. \tag{G.3}$$

Plugging (G.3) into (G.2) and recalling the definition of $\mathcal{L}(\pi)$ in (4.6), we finish the proof of Lemma 5.1. $\qquad\qquad\qquad\qquad\qquad\qquad\qquad\qquad\qquad\qquad\qquad\qquad\qquad\qquad\qquad\square$

*Proof of Lemma 5.2.*  First, we have

$$\mathrm{KL}(\pi^*(\cdot \,|\, s) \,\|\, \pi_{\theta_k}(\cdot \,|\, s)) - \mathrm{KL}(\pi^*(\cdot \,|\, s) \,\|\, \pi_{\theta_{k+1}}(\cdot \,|\, s))$$

$$= \langle \log(\pi_{\theta_{k+1}}(\cdot \,|\, s)/\pi_{\theta_k}(\cdot \,|\, s)), \pi^*(\cdot \,|\, s) \rangle$$

$$= \langle \log(\pi_{\theta_{k+1}}(\cdot \,|\, s)/\pi_{\theta_k}(\cdot \,|\, s)), \pi^*(\cdot \,|\, s) - \pi_{\theta_{k+1}}(\cdot \,|\, s) \rangle + \mathrm{KL}(\pi_{\theta_{k+1}}(\cdot \,|\, s) \,\|\, \pi_{\theta_k}(\cdot \,|\, s))$$

$$= \langle \log(\pi_{\theta_{k+1}}(\cdot \,|\, s)/\pi_{\theta_k}(\cdot \,|\, s)) - \beta_k^{-1} Q^{\pi_{\theta_k}}(s, \cdot), \pi^*(\cdot \,|\, s) - \pi_{\theta_k}(\cdot \,|\, s) \rangle$$

$$+ \beta_k^{-1} \cdot \langle Q^{\pi_{\theta_k}}(s, \cdot), \pi^*(\cdot \,|\, s) - \pi_{\theta_k}(\cdot \,|\, s) \rangle + \mathrm{KL}(\pi_{\theta_{k+1}}(\cdot \,|\, s) \,\|\, \pi_{\theta_k}(\cdot \,|\, s))$$

$$+ \langle \log(\pi_{\theta_{k+1}}(\cdot \,|\, s)/\pi_{\theta_k}(\cdot \,|\, s)), \pi_{\theta_k}(\cdot \,|\, s) - \pi_{\theta_{k+1}}(\cdot \,|\, s) \rangle. \tag{G.4}$$

Recall that $\pi_{k+1} \propto \exp\{\tau_k^{-1} f_{\theta_k} + \beta_k^{-1} Q^{\pi_{\theta_k}}\}$ and $Z_{k+1}(s)$ and $Z_{\theta_k}(s)$ are defined in (E.1). Also recall that we have $\langle \log Z_{\theta_k}(s), \pi(\cdot \,|\, s) - \pi'(\cdot \,|\, s) \rangle = \langle \log Z_k(s), \pi(\cdot \,|\, s) - \pi'(\cdot \,|\, s) \rangle = 0$ for all $k$,

$\pi$, and $\pi'$, which implies that, on the right-hand-side of (G.4),

$$\langle \log \pi_{\theta_k}(\cdot \,|\, s) + \beta_k^{-1} Q^{\pi_{\theta_k}}(s, \cdot), \pi^*(\cdot \,|\, s) - \pi_{\theta_k}(\cdot \,|\, s)\rangle$$

$$= \langle \tau_k^{-1} f_{\theta_k}(s, \cdot) + \beta_k^{-1} Q^{\pi_{\theta_k}}(s, \cdot), \pi^*(\cdot \,|\, s) - \pi_{\theta_k}(\cdot \,|\, s)\rangle - \langle \log Z_{\theta_k}(s), \pi^*(\cdot \,|\, s) - \pi_{\theta_k}(\cdot \,|\, s)\rangle$$

$$= \langle \tau_k^{-1} f_{\theta_k}(s, \cdot) + \beta_k^{-1} Q^{\pi_{\theta_k}}(s, \cdot), \pi^*(\cdot \,|\, s) - \pi_{\theta_k}(\cdot \,|\, s)\rangle - \langle \log Z_{k+1}(s), \pi^*(\cdot \,|\, s) - \pi_{\theta_k}(\cdot \,|\, s)\rangle$$

$$= \langle \log \pi_{k+1}(\cdot \,|\, s), \pi^*(\cdot \,|\, s) - \pi_{\theta_k}(\cdot \,|\, s)\rangle, \tag{G.5}$$

and

$$\big\langle \log(\pi_{\theta_{k+1}}(\cdot \,|\, s)/\pi_{\theta_k}(\cdot \,|\, s)), \pi_{\theta_k}(\cdot \,|\, s) - \pi_{\theta_{k+1}}(\cdot \,|\, s)\big\rangle$$

$$= \langle \tau_{k+1}^{-1} f_{\theta_{k+1}}(s, \cdot) - \tau_k^{-1} f_{\theta_k}(s, \cdot), \pi_{\theta_k}(\cdot \,|\, s) - \pi_{\theta_{k+1}}(\cdot \,|\, s)\rangle$$

$$\quad - \langle \log Z_{\theta_{k+1}}(s), \pi_{\theta_k}(\cdot \,|\, s) - \pi_{\theta_{k+1}}(\cdot \,|\, s)\rangle + \langle \log Z_{\theta_k}(s), \pi_{\theta_k}(\cdot \,|\, s) - \pi_{\theta_{k+1}}(\cdot \,|\, s)\rangle$$

$$= \langle \tau_{k+1}^{-1} f_{\theta_{k+1}}(s, \cdot) - \tau_k^{-1} f_{\theta_k}(s, \cdot), \pi_{\theta_k}(\cdot \,|\, s) - \pi_{\theta_{k+1}}(\cdot \,|\, s)\rangle. \tag{G.6}$$

Plugging (G.5) and (G.6) into (G.4), we obtain

$$\text{KL}(\pi^*(\cdot \,|\, s) \,\|\, \pi_{\theta_k}(\cdot \,|\, s)) - \text{KL}(\pi^*(\cdot \,|\, s) \,\|\, \pi_{\theta_{k+1}}(\cdot \,|\, s)) \tag{G.7}$$

$$= \langle \log(\pi_{\theta_{k+1}}(\cdot \,|\, s)/\pi_{k+1}(\cdot \,|\, s)), \pi^*(\cdot \,|\, s) - \pi_{\theta_k}(\cdot \,|\, s)\rangle + \beta_k^{-1} \cdot \langle Q^{\pi_{\theta_k}}(s, \cdot), \pi^*(\cdot \,|\, s) - \pi_{\theta_k}(\cdot \,|\, s)\rangle$$

$$\quad + \langle \tau_{k+1}^{-1} f_{\theta_{k+1}}(s, \cdot) - \tau_k^{-1} f_{\theta_k}(s, \cdot), \pi_{\theta_k}(\cdot \,|\, s) - \pi_{\theta_{k+1}}(\cdot \,|\, s)\rangle + \text{KL}(\pi_{\theta_{k+1}}(\cdot \,|\, s) \,\|\, \pi_{\theta_k}(\cdot \,|\, s))$$

$$\geq \langle \log(\pi_{\theta_{k+1}}(\cdot \,|\, s)/\pi_{k+1}(\cdot \,|\, s)), \pi^*(\cdot \,|\, s) - \pi_{\theta_k}(\cdot \,|\, s)\rangle + \beta_k^{-1} \cdot \langle Q^{\pi_{\theta_k}}(s, \cdot), \pi^*(\cdot \,|\, s) - \pi_{\theta_k}(\cdot \,|\, s)\rangle$$

$$\quad + \langle \tau_{k+1}^{-1} f_{\theta_{k+1}}(s, \cdot) - \tau_k^{-1} f_{\theta_k}(s, \cdot), \pi_{\theta_k}(\cdot \,|\, s) - \pi_{\theta_{k+1}}(\cdot \,|\, s)\rangle + 1/2 \cdot \|\pi_{\theta_{k+1}}(\cdot \,|\, s) - \pi_{\theta_k}(\cdot \,|\, s)\|_1^2,$$

where in the last inequality we use the Pinsker's inequality. Rearranging the terms in (G.7), we finish the proof of Lemma 5.2. $\qquad\square$