[Reviews · NeurIPS 2019]

Reviewer 1



This paper studied the proximal policy optimization (as well as TRPO), where the policy and action-value function are parameterized by two-layer neural networks. Due to the overparameterization of neural networks, the convergence rate for the policy improvement and policy evaluation can be analyzed. The overall convergence of the PPO algorithm then leverages the global convergence of infinite-dimensional mirror descent. Though I found the paper provides a valid of proof of convergence for PPO algorithm, I don't find the proof techniques to be novel enough. It appears that the entire proof lies in putting together several pieces of techniques/proofs that are already in existing work. More specifically, the proofs for policy improvement and policy evaluation are straightforward adaptations of the existing proofs for supervised learning over overparameterized neural networks. The overall convergence adapts the existing proof for mirror descent algorithm. The error propagation adapts the proof for the neural network Q-learning. Hence, the particular contribution in this paper is not very significant. More specific comments are as follows. - The proofs for the policy improvement and policy evaluation require taking average over the random initialization. However, the statements of the theorems (Theorems 4.5 and 4.6) do not include such an averaging. Please explain. - Typically, PPO even under linear function approximation may not convergence to global minimum. Is the convergence here due to the sufficiently large function class (i.e., Assumption 4.3)?

Reviewer 2



Originality: The authors apply the idea that overparametrization induces local linearization, which has been documented for supervised learning, and in another submission for TD learning. In particular, they decompose the error into two terms, one due to TD, and the other due to SGD, and incorporate them in the analysis of infinite-dimensional mirror descent. The insight that the previous previous analysis for TD could be generalised to a meta algorithm that includes both TD and SGD as particular cases is key. Related work is adequately cited, and differences with previous works are clearly stated, including differences with the sister submission [5]. Quality: The submission seems technically sound, and includes detailed proofs (I just skimmed through them). This is a complete piece of work. The main limitation of the analysis is the assumption on the neural network architecture, which according to (3.1) seems to be a multi-layer perceptron with a single hidden layer (as opposed to deep architectures with two hidden layers). No experimental results have been included. Clarity: The paper is well organised and clearly written. Significance: The submission addresses the convergence of PPO with a specific neural architecture, which is a very difficult task due to the nonconvexity of the objective, the parametrization of both value function and policy, and the stochastic updates. And they solve it with an intuitive approach. I believe the results are relevant, and will likely inspire further results on the nonasymptotic analysis of other deep-RL methods; especially if the local linearization effect induced by overparametrization can be identified in other neural network architectures. META REVIEW ============ The authors have addressed my comments. I appreciate their commitment to add simulation results to the paper. On the other hand, although I was very enthusiastic when reading this paper, I don't find the authors' response to Reviewer 1's concerns very satisfactory. In particular, the authors claim: "Our proof of the convergence of PPO/TRPO has several building blocks, which, to the best of our knowledge, is not covered by existing work", namely the "unified analysis of both TD and SGD", the "analysis of infinite-dimensional mirror descent under one-point monotonicity" and the "error propagation" of tracking a policy. It turns out that the first two points are very related to another submission: Paper 6047 'Neural Temporal-Difference Learning Converges to Global Optima'. Indeed, these are sentences copied from Paper 6047: - "overparametrization allows us to establish a notion of one-point monotonicity [25, 19] for the semigradients followed by neural TD, which ensures its evolution towards the global optimum of MSPBE along the solution path. Such a notion of monotonicity enables us to circumvent the first and second obstacles of bias and nonconvexity." - "In Section 4.2, we establish the nonasymptotic global rates of convergence of neural TD to the global optimum of the MSPBE when following the population semigradients in (3.3) and the stochastic semigradients in (3.1), respectively." - "The key to the proof of Theorem 4.4 is the one-point monotonicity of the population semigradient g(t), which is established through the local linearization..." Paper 6047 also covers Soft Q-learning and explicitly says that "our global convergence of neural soft Q-learning extends to a variant of the actor-critic algorithm." However, the relationship between that analysis and the error propagation of the current submission is not obvious. Therefore, even I still think the current paper is very relevant, it makes me think that the key idea of generalising previous neural-TD analysis is more incremental, so I reduce the score to 7.

Reviewer 3



The paper is a concrete work as theoretical work. The authors use overparameterized neural networks as Q functions, which should be inspired by the recent advance in this area. The convergence rate seems to be optimal to the problem, at least to my imagination. The paper, however, lacks an experimental section to validate the convergence rate is tight or not, which is a pity. Or there could be some analysis to compensate for the experiments in existing papers, which should convey insight to wider society. The overparameterization assumption seems to be not practical in actual experiments, at least to my recollection, although I understand this is probably not a job for an RL paper to accomplish. Besides, the paper seems to have no conclusion section? At line 37, the authors claim about solving the problem of infinite-dimensionality, nonconvexity comes from? could the authors elaborate a little more for the technical difficulty in the specific function? In algorithm 1, what is the criterion for line 5 and 6 to converge? or is it only one gradient descent step?

[Author Response · NeurIPS 2019]

We appreciate the valuable comments from the reviewers. We will revise accordingly.

**Reviewer #1.** (**Novelty in techniques.**) Our proof of the convergence of PPO/TRPO has several building blocks,
which, to the best of our knowledge, is not covered by existing work.

• **TD and SGD:** Our unified analysis covers both TD and SGD, where **TD is not covered by any existing supervised
learning analysis**. In particular, the semigradient used in the TD update is **not an unbiased stochastic gradient of
any supervised learning objective**. To see this, one may take the derivative of the semigradient, which gives an
asymmetric matrix. In contrast, in supervised learning, such a matrix is the Hessian, which must be symmetric.
A "straightforward adaptation" of existing supervised learning analysis does not yield the global convergence of TD. In
fact, most algorithms in supervised learning are known to at least converge (although they may converge to undesired
stationary points). In contrast, TD with nonlinear function approximator **is known to generally diverge** [1], which
eludes existing supervised learning analysis. Our unified analysis of both TD and SGD shows how overparametrization
allows bypassing the divergence of TD, which has not been observed in the context of supervised learning before.

• **Nonconvex mirror descent:** Most existing analysis of mirror descent's convergence to a **global optimum** builds
on the critical assumption that the objective is convex, which is not the case for the objective $J(\pi)$ in RL. Our **global
convergence** analysis of PPO/TRPO only builds on **one-point monotonicity** (Lemma 5.1), instead of convexity. In
fact, we are not aware of any existing analysis of **infinite-dimensional** mirror descent under one-point monotonicity.

• **Error propagation:** RL is divided into **policy-based** and **value-based** approaches. **PPO/TRPO falls into the
former**, while **Q-learning falls into the latter**. More specifically, PPO/TRPO explicitly tracks a policy, while Q-
learning does not. In particular, the Q-function tracked in Q-learning is not the action-value function of any policy. As
a result, the error propagation in Q-learning relies on the $\gamma$-**contraction of the Bellman optimality operator**, which
operates on the **Q-function**. In contrast, the error propagation in PPO/TRPO in our analysis relies on the **convergence
of mirror descent**, which updates the **policy**. Such two mechanisms are not comparable.

(**Expectation over initialization.**) Yes, the expectation is also taken over the initialization of the neural networks. Our
notation indeed lacks clarification. Thanks for pointing this out. We will revise accordingly.

(**Sufficiently large function class.**) Yes, the divergence is largely due to the limited representation power of **finite-
dimensional** linear function approximators. In contrast, overparametrized neural networks provide sufficient represen-
tation power as **infinite-dimensional** nonlinear function approximators, which enables the global convergence.

**Reviewer #2.** (**Shallow architecture.**) Thanks for pointing this out. The name "two-layer neural network" is standard
in recent work (see, e.g., [2]), counting the output layer as one of the layers. We will emphasize more on the shallow
architecture considered here.

(**Bounded reward.**) The state-value function and action-value function defined in Section 2 are normalized by a factor
of $1 - \gamma$. Hence, they have the same upper bound $R_{\max}$ as the reward function.

(**Simulation.**) Thanks for the suggestion. We will add illustrative simulation examples to the appendix.

**Reviewer #3.** (**Conclusion section.**) We will add a conclusion section to briefly summarize this paper as well as
potential future work.

(**Nonconvexity, infinite-dimensionality, and technical challenges.**) The nonconvexity arises from two aspects: (i)
The neural network parametrization of the energy-based policy and the action-value function makes the subproblems of
policy improvement and policy evaluation nonconvex. (ii) The RL objective $J(\pi)$ is also nonconvex in $\pi$. Meanwhile,
the continuous state space introduces infinite-dimensionality. The nonconvexity and infinite-dimensionality combined
makes the global convergence analysis challenging.

(**Algorithm 1.**) Yes, in Algorithm 1, lines 5 and 6 are just one projected TD step. Correspondingly, lines 5 and 6 in
Algorithm 2 are one projected SGD step.

(**Overparametrization.**) Our analysis does require the width of the neural network to be large enough. In practice, in
order to ensure the desired representation power of the neural network, a larger number of parameters is required.

[1] TSITSIKLIS, J. N. and VAN ROY, B. (1997). Analysis of temporal-diffference learning with function approximation.
In Advances in Neural Information Processing Systems.

[2] MUNOS, R. and SZEPESVARI, C. (2008). Finite-time bounds for fitted value iteration. Journal of Machine
Learning Research.

[3] ARORA, S., DU, S. S., HU, W., LI, Z. and WANG, R. (2019). Fine-grained analysis of optimization and
generalization for overparameterized two-layer neural networks. arXiv:1901.08584.


[Meta-Review · NeurIPS 2019]

The authors in aggregate put this paper above the acceptance threshold; it contains novel ideas that would be useful.